# TRex, a fast multi-animal tracking system with markerless identification, and 2D estimation of posture and visual fields

Tristan Walter[1,2,3]*, Iain D Couzin[1,2,3]*

[1]Max Planck Institute of Animal Behavior, Radolfzell, Germany; [2]Centre for the Advanced Study of Collective Behaviour, University of Konstanz, Konstanz, Germany; [3]Department of Biology, University of Konstanz, Konstanz, Germany

**Abstract** Automated visual tracking of animals is rapidly becoming an indispensable tool for the study of behavior. It offers a quantitative methodology by which organisms' sensing and decision-making can be studied in a wide range of ecological contexts. Despite this, existing solutions tend to be challenging to deploy in practice, especially when considering long and/or high-resolution video-streams. Here, we present TRex, a fast and easy-to-use solution for tracking a large number of individuals simultaneously using background-subtraction with real-time (60 Hz) tracking performance for up to approximately 256 individuals and estimates 2D visual-fields, outlines, and head/rear of bilateral animals, both in open and closed-loop contexts. Additionally, TRex offers highly accurate, deep-learning-based visual identification of up to approximately 100 unmarked individuals, where it is between 2.5 and 46.7 times faster, and requires 2–10 times less memory, than comparable software (with relative performance increasing for more organisms/longer videos) and provides interactive data-exploration within an intuitive, platform-independent graphical user-interface.

*For correspondence:
twalter@ab.mpg.de (TW);
icouzin@ab.mpg.de (IDC)

**Competing interests:** The authors declare that no competing interests exist.

## Introduction

Tracking multiple moving animals (and multiple objects, generally) is important in various fields of research such as behavioral studies, ecophysiology, biomechanics, and neuroscience (*Dell et al., 2014*). Many tracking algorithms have been proposed in recent years (*Ohayon et al., 2013*, *Fukunaga et al., 2015*, *Burgos-Artizzu et al., 2012*, *Rasch et al., 2016*), often limited to/only tested with a particular organism (*Hewitt et al., 2018*, *Branson et al., 2009*) or type of organism (e.g. protists, *Pennekamp et al., 2015*; fly larvae and worms, *Risse et al., 2017*). Relatively few have been tested with a range of organisms and scenarios (*Pérez-Escudero et al., 2014*, *Sridhar et al., 2019*, *Rodriguez et al., 2018*). Furthermore, many existing tools only have a specialized set of features, struggle with very long or high-resolution ($\geq$4 K) videos, or simply take too long to yield results. Existing fast algorithms are often severely limited with respect to the number of individuals that can be tracked simultaneously; for example, xyTracker (*Rasch et al., 2016*) allows for real-time tracking at 40 Hz while accurately maintaining identities, and thus is suitable for closed-loop experimentation (experiments where stimulus presentation can depend on the real-time behaviors of the individuals, for example *Bath et al., 2014*, *Brembs and Heisenberg, 2000*, *Bianco and Engert, 2015*), but has a limit of being able to track only five individuals simultaneously. ToxTrac (*Rodriguez et al., 2018*), a software comparable to xyTracker in it's set of features, is limited to 20 individuals and relatively low frame-rates ($\leq$25fps). Others, while implementing a wide range of features and offering high-performance tracking, are costly and thus limited in access (*Noldus et al., 2001*). Perhaps with the exception of proprietary software, one major problem at present is the severe fragmentation of features across the various software solutions. For example,

experimentalists must typically construct work-flows from many individual tools: One tool might be responsible for estimating the animal's positions, another for estimating their posture, another one for reconstructing visual fields (which in turn probably also estimates animal posture, but does not export it in any way) and one for keeping identities – correcting results of other tools post-hoc. It can take a very long time to make them all work effectively together, adding what is often considerable overhead to behavioral studies.

TRex, the software released with this publication (available at trex.run under an Open-Source license), has been designed to address these problems, and thus to provide a powerful, fast and easy to use tool that will be of use in a wide range of behavioral studies. It allows users to track moving objects/animals, as long as there is a way to separate them from the background (e.g. static backgrounds, custom masks, as discussed below). In addition to the positions of individuals, our software provides other per-individual metrics such as body shape and, if applicable, head-/tail-position. This is achieved using a basic posture analysis, which works out of the box for most organisms, and, if required, can be easily adapted for others. Posture information, which includes the body center-line, can be useful for detecting for example courtship displays and other behaviors that might not otherwise be obvious from mere positional data. Additionally, with the visual sense often being one of the most important modalities to consider in behavioral research, we include the capability for users to obtain a computational reconstruction of the visual fields of all individuals (*Strandburg-Peshkin et al., 2013*; *Rosenthal et al., 2015*). This not only reveals which individuals are visible from an individual's point-of-view, as well as the distance to them, but also which parts of others' bodies are visible.

Included in the software package is a task-specific tool, TGrabs, that is employed to pre-process existing video files and which allows users to record directly from cameras capable of live-streaming to a computer (with extensible support from generic webcams to high-end machine vision cameras). It supports most of the above-mentioned tracking features (positions, posture, visual field) and provides access to results immediately while continuing to record/process. This not only saves time, since tracking results are available immediately after the trial, but makes closed-loop support possible for large groups of individuals ($\leq$ 128 individuals). TRex and TGrabs are written in `C++` but, as part of our closed-loop support, we are providing a `Python`-based general scripting interface which can be fully customized by the user without the need to recompile or relaunch. This interface allows for compatibility with external programs (e.g. for closed-loop stimulus-presentation) and other custom extensions.

The fast tracking described above employs information about the kinematics of each organism in order to try to maintain their identities. This is very fast and useful in many scenarios, for example where general assessments about group properties (group centroid, alignment of individuals, density, etc.) are to be made. However, when making conclusions about *individuals* instead, maintaining identities perfectly throughout the video is a critical requirement. Every tracking method inevitably makes mistakes, which, for small groups of two or three individuals or short videos, can be corrected manually – at the expense of spending much more time on analysis, which rapidly becomes prohibitive as the number of individuals to be tracked increases. To make matters worse, when multiple individuals stay out of view of the camera for too long (such as if individuals move out of frame, under a shelter, or occlude one another) there is no way to know who is whom once they re-emerge. With no baseline truth available (e.g. using physical tags as in *Alarcón-Nieto et al., 2018*, *Nagy et al., 2013*; or marker-less methods as in *Pérez-Escudero et al., 2014*, *Romero-Ferrero et al., 2019*, *Rasch et al., 2016*), these mistakes cannot be corrected and accumulate over time, until eventually all identities are fully shuffled. To solve this problem (and without the need to mark, or add physical tags to individuals), TRex can, at the cost of spending more time on analysis (and thus not during live-tracking), automatically learn the identity of up to approximately 100 unmarked individuals based on their visual appearance. This machine-learning-based approach, herein termed *visual identification*, provides an independent source of information on the identity of individuals, which is used to detect and correct potential tracking mistakes without the need for human supervision.

In this paper, we evaluate the most important functions of our software in terms of speed and reliability using a wide range of experimental systems, including termites, fruit flies, locusts, and multiple species of schooling fish (although we stress that our software is not limited to such species).

Specifically regarding the visual identification of unmarked individuals in groups, `idtracker.ai` is currently state-of-the-art, yielding high-accuracy (> 99% in most cases) in maintaining consistent identity assignments across entire videos (*Romero-Ferrero et al., 2019*). Similarly to TRex, this is achieved by training an artificial neural network to visually differentiate between individuals, and using identity predictions from this network to avoid/correct tracking mistakes. Both approaches work without human supervision, and are limited to approximately 100 individuals. Given that `idtracker.ai` is the only currently available tool with visual identification for such large groups of individuals, and also because of the quality of results, we will use it as a benchmark for our visual identification system. Results will be compared in terms of both accuracy and computation speed, showing TRex' ability to achieve the same high level of accuracy but typically at far higher speeds, and with a much reduced memory requirement.

TRex is platform-independent and runs on all major operating systems (Linux, Windows, macOS) and offers complete batch processing support, allowing users to efficiently process entire sets of videos without requiring human intervention. All parameters can be accessed either through settings files, from within the graphical user interface (or *GUI*), or using the command-line. The user interface supports off-site access using a built-in web-server (although it is recommended to only use this from within a secure VPN environment). Available parameters are explained in the documentation directly as part of the GUI and on an external website (see below). Results can be exported to independent data-containers (NPZ, or CSV for plain-text type data) for further analyses in software of the user's choosing. We will not go into detail regarding the many GUI functions since albeit being of great utility to the researcher, they are only the means to easily apply the features presented herein. Some examples will be given in the main text and appendix, but a comprehensive collection of all of them, as well as detailed documentation, is available in the up-to-date online-documentation which can be found at trex.run/docs.

## Results

Our software package consists of two task-specific tools, TGrabs and TRex, with different specializations. TGrabs is primarily designed to connect to cameras and to be very fast. It employs the same program code as TRex to achieve real-time online tracking, such as could be employed for closed-loop experiments (the user can launch TGrabs from the opening dialog of TRex). However, its focus on speed comes at the cost of not having access to the rich GUI or more sophisticated (and thus slower) processing steps, such as deep-learning-based identification, that TRex provides. TRex focusses on the more time-consuming tasks, as well as visual data exploration, re-tracking existing results – but sometimes it simply functions as an easier-to-use graphical interface for tracking and adjusting parameters. Together they provide a wide range of capabilities to the user and are often used in sequence as part of the same work-flow. Typically, such a sequence can be summarized in four stages (see also *Figure 1* for a flow diagram):

1. Segmentation in TGrabs. When recording a video or converting a previously recorded file (e.g. MP4, .AVI, etc.), it is segmented into background and foreground-objects (`blobs`), the latter typically being the entities to be tracked. Results are saved to a custom, non-proprietary video format (`PV`) (*Figure 2a*).
2. Tracking the video, either directly in TGrabs, or in TRex after pre-processing, with access to customizable visualizations and the ability to change tracking parameters on-the-fly. Here, we will describe two types of data available within TRex, 2D posture- and visual-field estimation, as well as real-time applications of such data (*Figure 2b*).
3. Automatic identity correction (*Figure 2c*), a way of utilizing the power of a trained neural network to perform visual identification of individuals, is available in TRex only. This step may not be necessary in many cases, but it is the only way to guarantee consistent identities throughout the video. It is also the most processing-heavy (and thus usually the most time-consuming) step, as well as the only one involving machine learning. All previously collected posture- and other tracking-related data are utilized in this step, placing it late in a typical workflow.
4. Data visualization is a critical component of any research project, especially for unfamiliar datasets, but manually crafting one for every new experiment can be very time-consuming. Thus, TRex offers a universal, highly customizable, way to make all collected data available for interactive exploration (*Figure 2d*) – allowing users to change many display options and recording video clips for external playback. Tracking parameters can be adjusted on the fly (many with

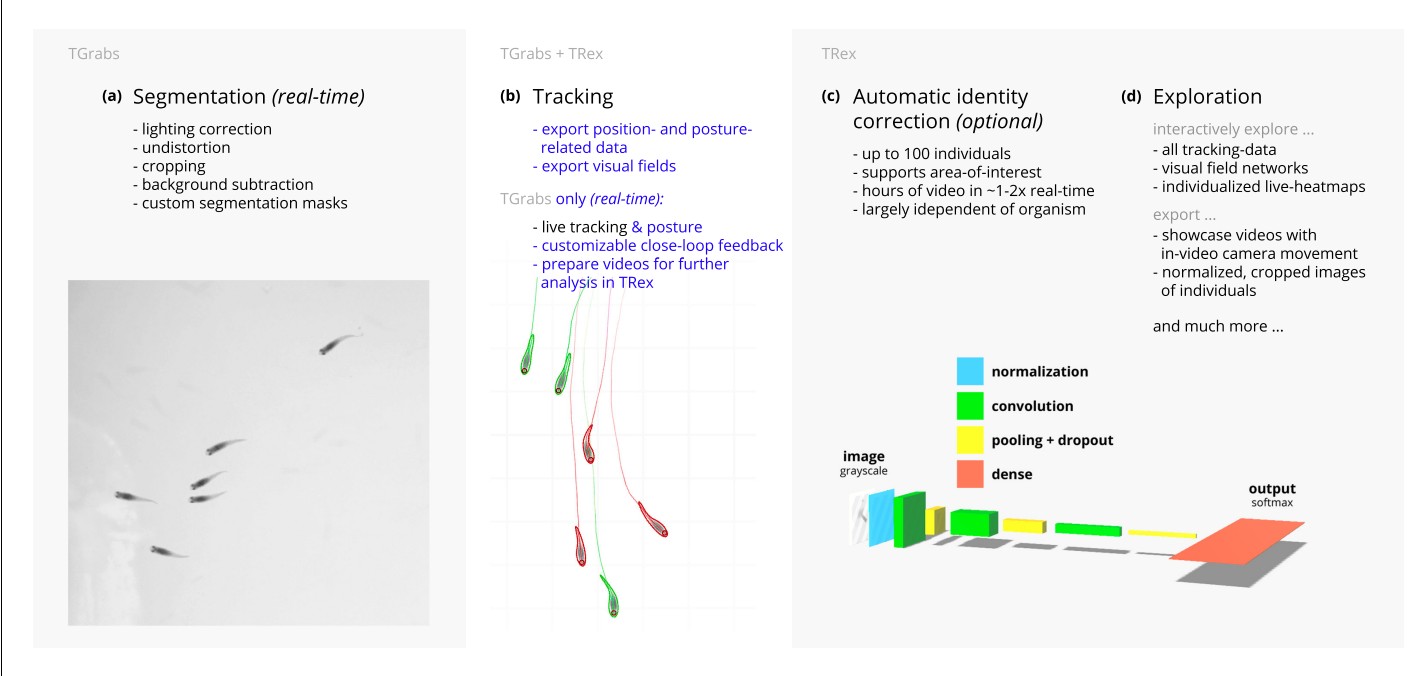

**Figure 1.** Videos are typically processed in four main stages, illustrated here each with a list of prominent features. Some of them are accessible from both TRex and TGrabs, while others are software specific (as shown at the very top). (a) The video is either recorded directly with our software (TGrabs), or converted from a pre-recorded video file. Live-tracking enables users to perform closed-loop experiments, for which a virtual testing environment is provided. (b) Videos can be tracked and parameters adjusted with visual feedback. Various exploration and data presentation features are provided and customized data streams can be exported for use in external software. (c) After successful tracking, automatic visual identification can, optionally, be used to refine results. An artificial neural network is trained to recognize individuals, helping to automatically correct potential tracking mistakes. In the last stage, many graphical tools are available to users of TRex, a selection of which is listed in (d).

visual feedback) – important for example when preparing a closed-loop feedback with a new species or setup.

Below we assess the performance of our software regarding three properties that are most important when using it (or in fact any tracking software) in practice: (i) The time it takes to perform tracking (ii) the time it takes to perform automatic identity correction and (iii) the peak memory consumption when correcting identities (since this is where memory consumption is maximal), as well as (iv) the accuracy of the produced trajectories after visual identification.

While accuracy is an important metric and specific to identification tasks, time and memory are typically of considerable practical importance for all tasks. For example, tracking-speed may be the difference between only being able to run a few trials or producing more reliable results with a much larger number of trials. In addition, tracking speed can make a major difference as the number of individuals increases. Furthermore, memory constraints can be extremely prohibitive making tracking over long video sequences and/or for a large number of individuals extremely time-consuming, or impossible, for the user.

In all of our tests, we used a relatively modest computer system, which could be described as a mid-range consumer or gaming PC:

- Intel Core i9-7900X CPU
- NVIDIA Geforce 1080 Ti
- 64 GB RAM
- NVMe PCIe x4 hard-drive
- Debian bullseye (debian.org)

As can be seen in the following sections (memory consumption, processing speeds, etc.) using a high-end system is not necessary to run TRex and, anecdotally, we did not observe noticeable improvements when using a solid state drive versus a normal hard drive. A video card (presently an

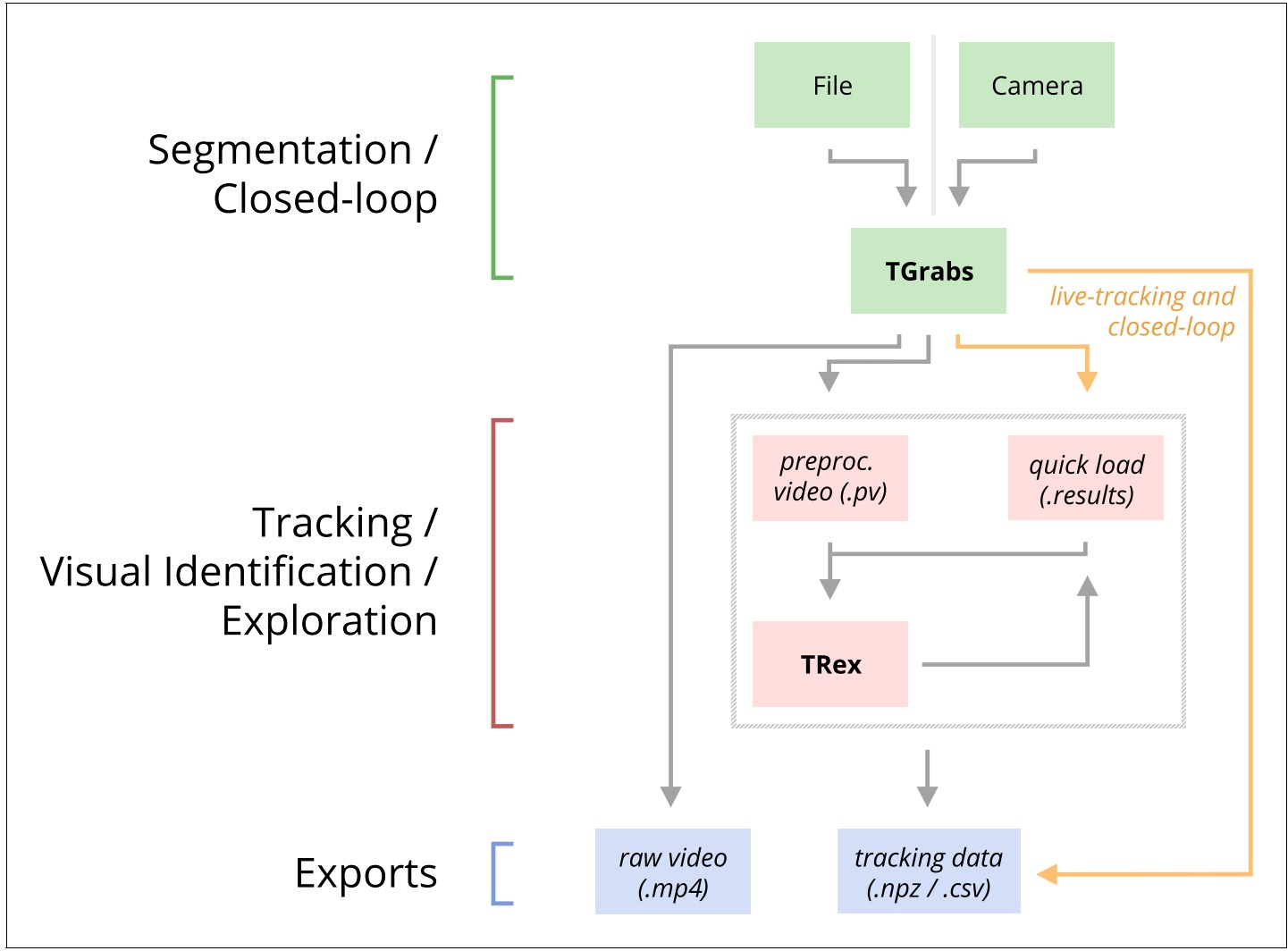

**Figure 2.** An overview of the interconnection between TRex, TGrabs and their data in- and output formats, with titles on the left corresponding to the stages in 1. Starting at the top of the figure, video is either streamed to TGrabs from a file or directly from a compatible camera. At this stage, preprocessed data are saved to a .pv file which can be read by TRex later on. Thanks to its integration with parts of the TRex code, TGrabs can also perform online tracking for limited numbers of individuals, and save results to a .results file (that can be opened by TRex) along with individual tracking data saved to numpy data-containers (.npz) or standard CSV files, which can be used for analysis in third-party applications. If required, videos recorded directly using TGrabs can also be streamed to a .mp4 video file which can be viewed in commonly available video players like VLC.

The online version of this article includes the following video for figure 2:

**Figure 2—video 1.** This video shows an overview of the typical chronology of operations when using our software.

https://elifesciences.org/articles/64000#fig2video1

NVIDIA card due to the requirements of TensorFlow) is recommended for tasks involving visual identification as such computations will take much longer without it – however, it is not required. We decided to employ this system due to having a relatively cheap, compatible graphics card, as well as to ensure that we have an easy way to produce direct comparisons with `idtracker.ai` – which according to their website requires large amounts of RAM (32 – 128 GB, idtrackerai online documentation) and a fast solid-state drive.

*Table 1* shows the entire set of videos used in this paper, which have been obtained from multiple sources (credited under the table) and span a wide range of different organisms, demonstrating TRex' ability to track anything as long as it moves occasionally. Videos involving a large number (> 100) of individuals are all the same species of fish since these were the only organisms we had available in such quantities. However, this is not to say that only fish could be tracked efficiently in these

**Table 1.** A list of the videos used in this paper as part of the evaluation of TRex, along with the species of animals in the videos and their common names, as well as other video-specific properties.

Videos are given an incremental ID, to make references more efficient in the following text, which are sorted by the number of individuals in the video. Individual quantities are given accurately, except for the videos with more than 100 where the exact number may be slightly more or less. These videos have been analyzed using TRex' dynamic analysis mode that supports unknown quantities of animals. Videos 7 and 8, as well as 13–11, are available as part of the original idtracker paper (*Pérez-Escudero et al., 2014*). Many of the videos are part of yet unpublished data: Guppy videos have been recorded by A. Albi, videos with sunbleak (Leucaspius delineatus) have been recorded by D. Bath. The termite video has been kindly provided by H. Hugo and the locust video by F. Oberhauser. Due to the size of some of these videos (>150 GB per video), they have to be made available upon specific request. Raw versions of these videos (some trimmed), as well as full preprocessed versions, are available as part of the dataset published alongside this paper (*Walter et al., 2020*).

| ID | Species | Common | # ind. | Fps (Hz) | Duration | Size (Px$^2$) (px$^2$) |
|----|---------|--------|--------|----------|----------|------------------------|
| 0 | *Leucaspius delineatus* | Sunbleak | 1024 | 40 | 8 min 20 s | 3866 × 4048 |
| 1 | *Leucaspius delineatus* | Sunbleak | 512 | 50 | 6 min 40 s | 3866 × 4140 |
| 2 | *Leucaspius delineatus* | Sunbleak | 512 | 60 | 5 min 59 s | 3866 × 4048 |
| 3 | *Leucaspius delineatus* | Sunbleak | 256 | 50 | 6 min 40 s | 3866 × 4140 |
| 4 | *Leucaspius delineatus* | Sunbleak | 256 | 60 | 5 min 59 s | 3866 × 4048 |
| 5 | *Leucaspius delineatus* | Sunbleak | 128 | 60 | 6 min | 3866 × 4048 |
| 6 | *Leucaspius delineatus* | Sunbleak | 128 | 60 | 5 min 59 s | 3866 × 4048 |
| 7 | *Danio rerio* | Zebrafish | 100 | 32 | 1 min | 3584 × 3500 |
| 8 | *Drosophila melanogaster* | Fruit-fly | 59 | 51 | 10 min | 2306 × 2306 |
| 9 | *Schistocerca gregaria* | Locust | 15 | 25 | 1hr 0 min | 1880 × 1881 |
| 10 | *Constrictotermes cyphergaster* | Termite | 10 | 100 | 10 min 5 s | 1920 × 1080 |
| 11 | *Danio rerio* | Zebrafish | 10 | 32 | 10 min 10 s | 3712 × 3712 |
| 12 | *Danio rerio* | Zebrafish | 10 | 32 | 10 min 3 s | 3712 × 3712 |
| 13 | *Danio rerio* | zebrafish | 10 | 32 | 10 min 3 s | 3712 × 3712 |
| 14 | *Poecilia reticulata* | Guppy | 8 | 30 | 3 hr 15 min 22 s | 3008 × 3008 |
| 15 | *Poecilia reticulata* | Guppy | 8 | 25 | 1 hr 12 min | 3008 × 300 |
| 16 | *Poecilia reticulata* | Guppy | 8 | 35 | 3 hr 18 min 13 s | 3008 × 3008 |
| 17 | *Poecilia reticulata* | Guppy | 1 | 140 | 1 hr 9 min 32 s | 1312 × 1312 |

quantities. We used the full dataset with up to 1024 individuals in one video (Video 0) to evaluate raw tracking speed without visual identification and identity corrections (next sub-section). However, since such numbers of individuals exceed the capacity of the neural network used for automatic identity corrections (compare also *Romero-Ferrero et al., 2019* who used a similar network), we only used a subset of these videos (videos 7 through 16) to look specifically into the quality of our visual identification in terms of keeping identities and its memory consumption.

## Tracking: speed and accuracy

In evaluating the 4.2 Tracking portion of TRex, the main focus lies with processing speed, while accuracy in terms of keeping identities is of secondary importance. Tracking is required in all other parts of the software, making it an attractive target for extensive optimization. Especially with regard to closed-loop, and live-tracking situations, there may be no room even to lose a millisecond between frames and thus risk dropping frames. We therefore designed TRex to support the simultaneous tracking of many ($\geq$256) individuals *quickly* and achieve reasonable *accuracy* for up to 100 individuals – which are the two suppositions we will investigate in the following.

Trials were run without posture/visual-field estimation enabled, where tracking generally, and consistently, reaches speeds faster than real-time (processing times of 1.5 – 40 % of the video duration, 25 – 100 Hz) even for a relatively large number of individuals (77 – 94.77 % for up to 256 individuals, see *Appendix 4—table 1*). Videos with more individuals (> 500) were still tracked within

reasonable time of 235– 358 % of the video duration. As would be expected from these results, we found that combining tracking and recording in a single step generally leads to higher processing speeds. The only situation where this was not the case was a video with 1024 individuals, which suggests that live-tracking (in TGrabs) handles cases with many individuals slightly worse than offline tracking (in TRex). Otherwise, 5– 35 % shorter total processing times were measured (14.55 % on average, see *Appendix 4—table 4*), compared to running TGrabs separately and then tracking in TRex. These percentage differences, in most cases, reflect the ratio between the video duration and the time it takes to track it, suggesting that most time is spent – by far – on the conversion of videos. This additional cost can be avoided in practice when using TGrabs to record videos, by directly writing to a custom format recognized by TRex, and/or using its live-tracking ability to export tracking data immediately after the recording is stopped.

We also investigated trials that were run with posture estimation *enabled* and we found that real-time speed could be achieved for videos with ≤128 individuals (see column 'tracking' in *Appendix 4—table 4*). Tracking speed, when posture estimation is enabled, depends more strongly on the size of individuals in the image.

Generally, tracking software becomes slower as the number of individuals to be tracked increases, as a result of an exponentially growing number of combinations to consider during matching. TRex uses a novel tree-based algorithm by default (see Tracking), but circumvents problematic situations by falling back on using the *Hungarian method* (also known as the *Kuhn-Munkres algorithm*, *Kuhn, 1955*) when necessary. Comparing our mixed approach (see Tracking) to purely using the Hungarian method shows that, while both perform similarly for few individuals, the Hungarian method is easily outperformed by our algorithm for larger groups of individuals (as can be seen in *Appendix 4—figure 3*). This might be due to custom optimizations regarding local cliques of individuals, whereby we ignore objects that are too far away, and also as a result of our optimized presorting. The Hungarian method has the advantage of not leading to combinatorical explosions in some situations – and thus has a lower *maximum* complexity while proving to be less optimal in the *average* case. For further details, see the appendix: Appendix D Matching an object to an object in the next frame.

In addition to speed, we also tested the accuracy of our tracking method, with regard to the consistency of identity assignments, comparing its results to the manually reviewed data (the methodology of which is described in the next section). In order to avoid counting follow-up errors as 'new' errors, we divided each trajectory in the uncorrected data into 'uninterrupted' segments of frames, instead of simply comparing whole trajectories. A segment is interrupted when an individual is lost

**Table 2.** Results of the human validation for a subset of videos.

Validation was performed by going through all problematic situations (e.g. individuals lost) and correcting mistakes manually, creating a fully corrected dataset for the given videos. This dataset may still have missing frames for some individuals, if they could not be detected in certain frames (as indicated by 'of that interpolated'). This was usually a very low percentage of all frames, except for Video 9, where individuals tended to rest on top of each other – and were thus not tracked – for extended periods of time. This baseline dataset was compared to all other results obtained using the automatic visual identification by TRex ($N = 5$) and `idtracker.ai` ($N = 3$) to estimate correctness. We were not able to track Videos 9 and 10 with `idtracker.ai`, which is why correctness values are not available.

| Video metrics | | Review stats | | % correct | |
|---|---|---|---|---|---|
| Video | # ind. | Reviewed (%) | Of that interpolated (%) | TRex | `idtracker.ai` |
| 7 | 100 | 100.0 | 0.23 | 99.07 ± 0.013 | 98.95 ± 0.146 |
| 8 | 59 | 100.0 | 0.15 | 99.68 ± 0.533 | 99.94 ± 0.0 |
| 9 | 15 | 22.2 | 8.44 | 95.12 ± 6.077 | N/A |
| 10 | 10 | 100.0 | 1.21 | 99.7 ± 0.088 | N/A |
| 13 | 10 | 100.0 | 0.27 | 99.98 ± 0.0 | 99.96 ± 0.0 |
| 12 | 10 | 100.0 | 0.59 | 99.94 ± 0.006 | 99.63 ± 0.0 |
| 11 | 10 | 100.0 | 0.5 | 99.89 ± 0.009 | 99.34 ± 0.002 |

The online version of this article includes the following source data for Table 2:

Source data 1. A table of positions for each individual of each manually approved and corrected trial.

(for any of the reasons given in 4.3.1 Preparing Tracking-Data) and starts again when it is reassigned to another object later on. We term these (re-)assignments *decisions* here. Each segment of every individual can be uniquely assigned to a similar/identical segment in the baseline data and its identity. Following one trajectory in the uncorrected data, we can detect these wrong decisions by checking whether the baseline identity associated with one segment of that trajectory changes in the next. We found that roughly 80 % of such decisions made by the tree-based matching were correct, even with relatively high numbers of individuals (100). For trajectories where no manually reviewed data were available, we used automatically corrected trajectories as a base for our comparison – we evaluate the accuracy of these automatically corrected trajectories in the following section. Even though we did not investigate accuracy in situations with more than 100 individuals, we suspect similar results since the property with the strongest influence on tracking accuracy – individual density – is limited physically and most of the investigated species school tightly in either case.

## Visual identification: accuracy

Since the goal of using visual identification is to generate consistent identity assignments, we evaluated the accuracy of our method in this regard. As a benchmark, we compare it to manually reviewed datasets as well as results from `idtracker.ai` for the same set of videos (where possible). In order to validate trajectories exported by either software, we manually reviewed multiple videos with the help from a tool within TRex that allows to view each crossing and correct possible mistakes in-place. Assignments were deemed incorrect, and subsequently corrected by the reviewer, if the centroid of a given individual was not contained within the object it was assigned to (e.g. the individual was not part of the correct object). Double assignments per object are impossible due to the nature of the tracking method. Individuals were also forcibly assigned to the correct objects in case they were visible but not detected by the tracking algorithm. After manual corrections had

**Table 3.** Evaluating comparability of the automatic visual identification between `idtracker.ai` and TRex.

Columns show various video properties, as well as the associated uniqueness score (see Guiding the training process) and a similarity metric. Similarity (*% similar individuals*) is calculated based on comparing the positions for each identity exported by both tools, choosing the closest matches overall and counting the ones that are differently assigned per frame. An individual is classified as 'wrong' in that frame, if the euclidean distance between the matched solutions from `idtracker.ai` and TRex exceeds 1 % of the video width. The column '% similar individuals' shows percentage values, where a value of 99% would indicate that, on average, 1 % of the individuals are assigned differently. To demonstrate how uniqueness corresponds to the quality of results, the last column shows the average uniqueness achieved across trials. A file containing all X and Y positions for each trial and each software combined into one very large table is available from *Walter et al., 2020*, along with the data in different formats.

| Video | # ind. | N TRex | % similar individuals | Final uniqueness |
|---|---|---|---|---|
| 7 | 100 | 5 | 99.8346 ± 0.5265 | 0.9758 ± 0.0018 |
| 8 | 59 | 5 | 98.6885 2.1145 | 0.9356 ± 0.0358 |
| 13 | 10 | 5 | 99.9902 0.3737 | 0.9812 ± 0.0013 |
| 11 | 10 | 5 | 99.9212 ± 1.1208 | 0.9461 ± 0.0039 |
| 12 | 10 | 5 | 99.9546 ± 0.8573 | 0.9698 ± 0.0024 |
| 14 | 8 | 5 | 98.8356 ± 5.8136 | 0.9192 ± 0.0077 |
| 15 | 8 | 5 | 99.2246 ± 4.4486 | 0.9576 ± 0.0023 |
| 162 | 8 | 5 | 99.7704 ± 2.1994 | 0.9481 ± 0.0025 |

The online version of this article includes the following source data for  Table 3:

Source data 1. Assignments between identities from multiple solutions, as calculated by a bipartite-graph matching algorithm.For each permutation of trials from TRex and idtracker.ai for the same video, the algorithm sought to match the trajectories of the same physical individuals in both trials with each other by finding the ones with the smallest mean euclidean distance per frame between them. Available from *Walter et al., 2020* as T2_Source_data.zip.

been applied, 'clean' trajectories were exported – providing a per-frame baseline truth for the respective videos. A complete table of reviewed videos, and the percentage of reviewed frames per video, can be found in *Table 2*. For longer videos (> 1 hr), we relied entirely on a comparison between results from `idtracker.ai` and TRex. Their paper (*Romero-Ferrero et al., 2019*) suggests a very high accuracy of over 99.9 % correctly identified individual images for most videos, which should suffice for most relevant applications and provide a good baseline truth. As long as both tools produce sufficiently similar trajectories, we therefore know they have found the correct solution.

A direct comparison between TRex and `idtracker.ai` was not possible for Videos 9 and 10, where idtracker.ai frequently exceeded hardware memory-limits and caused the application to be terminated, or did not produce usable results within multiple days of run-time. However, we were able to successfully analyze these videos with TRex and evaluate its performance by comparing to manually reviewed trajectories (see below in Visual identification: accuracy). Due to the stochastic nature of machine learning, and thus the inherent possibility of obtaining different results in each run, as well as other potential factors influencing processing time and memory consumption, both TRex and `idtracker.ai` have been executed repeatedly (5x TRex, 3x idtracker.ai).

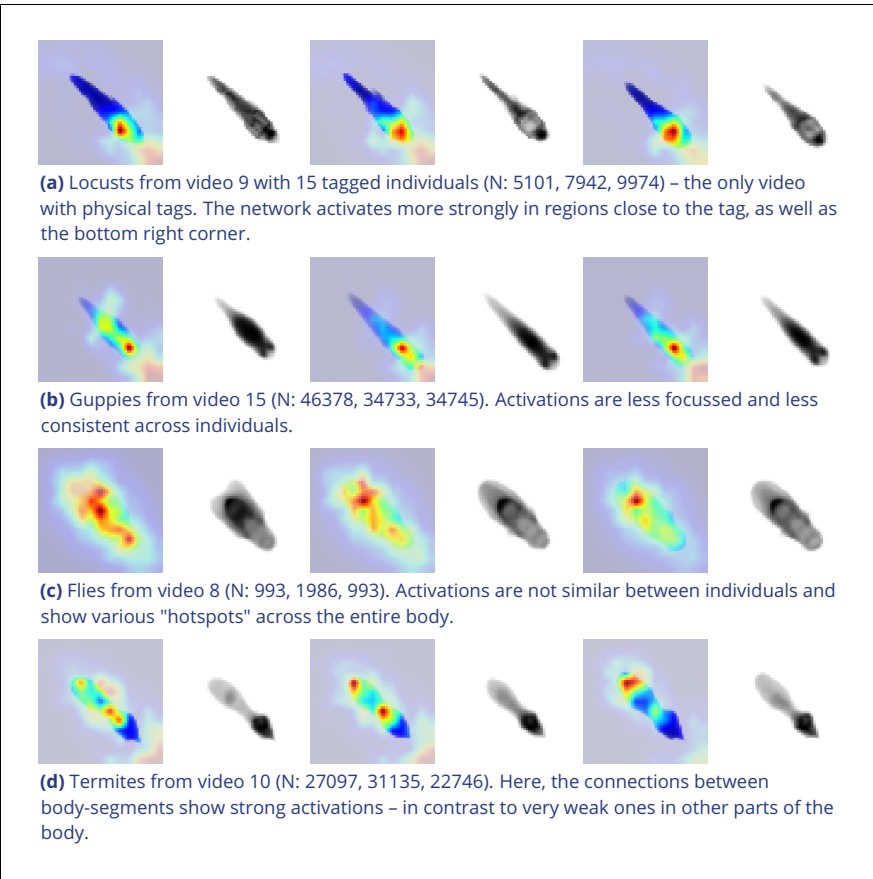

**(a)** Locusts from video 9 with 15 tagged individuals (N: 5101, 7942, 9974) – the only video with physical tags. The network activates more strongly in regions close to the tag, as well as the bottom right corner.

**(b)** Guppies from video 15 (N: 46378, 34733, 34745). Activations are less focussed and less consistent across individuals.

**(c)** Flies from video 8 (N: 993, 1986, 993). Activations are not similar between individuals and show various "hotspots" across the entire body.

**(d)** Termites from video 10 (N: 27097, 31135, 22746). Here, the connections between body-segments show strong activations – in contrast to very weak ones in other parts of the body.

**Figure 3.** Activation differences for images of randomly selected individuals from four videos, next to a median image of the respective individual – which hides thin extremities, such as legs in (a) and (c). The captions in (a-d) detail the species per group and number of samples per individual. Colors represent the relative activation differences, with hotter colors suggesting bigger magnitudes, which are computed by performing a forward-pass through the network up to the last convolutional layer (using keract). The outputs for each identity are averaged and stretched back to the original image size by cropping and scaling according to the network architecture. Differences shown here are calculated per cluster of pixels corresponding to each filter, comparing average activations for images from the individual's class to activations for images from other classes.
The online version of this article includes the following source data for figure 3:

**Source data 1.** Code, as well as images/weights needed to produce this figure (see README).

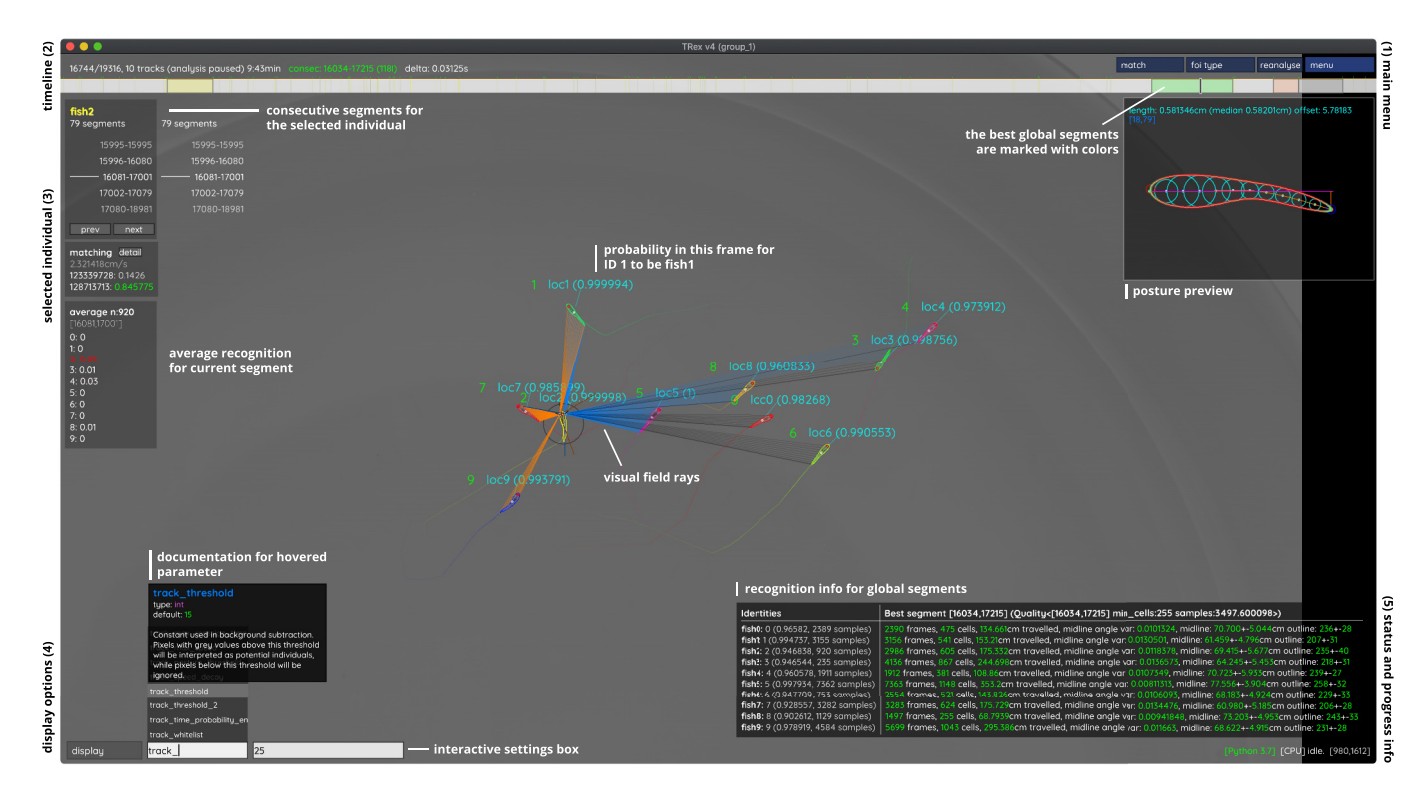

**Figure 4.** An overview of TRex' the main interface, which is part of the documentation at trex.run/docs. Interface elements are sorted into categories in the four corners of the screen (labelled here in black). The omni-box on the bottom left corner allows users to change parameters on-the-fly, helped by a live auto-completion and documentation for all settings. Only some of the many available features are displayed here. Generally, interface elements can be toggled on or off using the bottom-left display options or moved out of the way with the cursor. Users can customize the tinting of objects (e.g. sourcing it from their speed) to generate interesting effect and can be recorded for use in presentations. Additionally, all exportable metrics (such as border-distance, size, x/y, etc.) can also be shown as an animated graph for a number of selected objects. Keyboard shortcuts are available for select features such as loading, saving, and terminating the program. Remote access is supported and offers the same graphical user interface, for example in case the software is executed without an application window (for batch processing purposes).

The trajectories exported by both `idtracker.ai` and `TRex` were very similar throughout (see *Table 3*). While occasional disagreements happened, similarity scores were higher than 98 % in all and higher than 99 % in most cases (i.e. less than 1 % of individuals have been differently assigned in each frame on average). Most difficulties that *did* occur were, after manual review, attributable to situations where multiple individuals cross over excessively within a short time-span. In each case that has been manually reviewed, identities switched back to the correct individuals – even after temporary disagreement. We found that both solutions occasionally experienced these same problems, which often occur when individuals repeatedly come in and out of view in quick succession (e.g. overlapping with other individuals). Disagreements were expected for videos with many such situations due to the way both algorithms deal differently with them: `idtracker.ai` assigns identities only based on the network output. In many cases, individuals continue to partly overlap even while already being tracked, which results in visual artifacts and can lead to unstable predictions by the network and causing `idtracker.ai`'s approach to fail. Comparing results from both `idtracker.ai` and `TRex` to manually reviewed data (see *Table 2*) shows that both solutions consistently provide high-accuracy results of above 99.5 % for most videos, but that TRex is slightly improved in all cases while also having a better overall frame coverage per individual (99.65 % versus `idtracker.ai`'s 97.93 %, where 100 % would mean that all individuals are tracked in every frame; not shown). This suggests that the splitting algorithm (see appendix, Appendix K Algorithm for splitting touching individuals) is working to TRex' advantage here.

**Table 4.** Both TRex and `idtracker.ai` analyzed the same set of videos, while continuously logging their memory consumption using an external tool.

Rows have been sorted by video_length ∗ #individuals, which seems to be a good predictor for the memory consumption of both solutions. `idtracker.ai` has mixed mean values, which, at low individual densities are similar to TRex' results. Mean values can be misleading here, since more time spent in low-memory states skews results. The maximum, however, is more reliable since it marks the memory that is necessary to run the system. Here, `idtracker.ai` clocks in at significantly higher values (almost always more than double) than TRex.

| Video | #ind. | Length | Max.consec. | TRex memory (GB) | Idtracker.ai memory (GB) |
|---|---|---|---|---|---|
| 12 | 10 | 10 min | 26.03s | 4.88 ± 0.23, max 6.31 | 8.23 ± 0.99, max 28.85 |
| 13 | 10 | 10 min | 36.94s | 4.27 ± 0.12, max 4.79 | 7.83 ± 1.05, max 29.43 |
| 11 | 10 | 10 min | 28.75s | 4.37 ± 0.32, max 5.49 | 6.53 ± 4.29, max 29.32 |
| 7 | 100 | 1 min | 5.97s | 9.4 ± 0.47, max13.45 | 15.27 ± 1.05, max 24.39 |
| 15 | 8 | 72 min | 79.4s | 5.6 ± 0.22, max 8.41 | 35.2 ± 4.51, max 91.26 |
| 10 | 10 | 10 min | 1391s | 6.94 ± 0.27, max 10.71 | N/A |
| 9 | 15 | 60 min | 7.64s | 13.81 ± 0.53, max 16.99 | N/A |
| 8 | 59 | 10 min | 102.35s | 12.4 ± 0.56, max 17.41 | 35.3 ± 0.92, max 50.26 |
| 14 | 8 | 195 min | 145.77s | 12.44 ± 0.8, max 21.99 | 35.08 ± 4.08, max 98.04 |
| 16 | 8 | 198 min | 322.57s | 16.15 ± 1.6, max 28.62 | 49.24 ± 8.21, max 115.37 |

The online version of this article includes the following source data for Table 4:

**Source data 1.** Data from log files for all trials as a single table, where each row is one sample.The total memory of each sample is calculated as $SWAP + PRIVATE + SHARED$. Each row indicates at which exact time, by which software, and as part of which trial it was taken.

Additionally, while TRex could successfully track individuals in all videos without tags, we were interested to see the effect of tags (in this case QR tags attached to locusts, see *Figure 3a*) on network training. In *Figure 3*, we visualize differences in network activation, depending on the visual features available for the network to learn from, which are different between species (or due to physically added tags, as mentioned above). The 'hot' regions indicate larger between-class differences for that specific pixel (values are the result of activation in the last convolutional layer of the trained network, see figure legend). Differences are computed separately within each group and are not directly comparable between trials/species in value. However, the distribution of values – reflecting the network's reactivity to specific parts of the image – is. Results show that the most apparent differences are found for the stationary parts of the body (not in absolute terms, but following normalization, as shown in *Figure 4c*), which makes sense seeing as this part (i) is the easiest to learn due to it being in exactly the same position every time, (ii) larger individuals stretch further into the corners of a cropped image, making the bottom right of each image a source of valuable information (especially in *Figure 3a and b*) and (iii) details that often occur in the head-region (like distance between the eyes) which can also play a role here. 'Hot' regions in the bottom right corner of the activation images (e.g. in *Figure 3d*) suggest that also pixels are reacted to which are explicitly *not* part of the individual itself but of other individuals – likely this corresponds to the network making use of size/shape differences between them.

As would be expected, distinct patterns can be recognized in the resulting activations after training as soon as physical tags are attached to individuals (as in *Figure 3a*). While other parts of the image are still heavily activated (probably to benefit from size/shape differences between individuals), tags are always at least a large part of where activations concentrate. The network seemingly makes use of the additional information provided by the experimenter, where that has occurred. This suggests that, while definitely not being necessary, adding tags probably does not worsen, and likely may even improve, training accuracy, for difficult cases allowing networks to exploit any source of inter-individual variation.

## Visual identification: memory consumption

In order to generate comparable results between both tested software solutions, the same external script has been used to measure shared, private and swap memory of `idtracker.ai` and TRex,

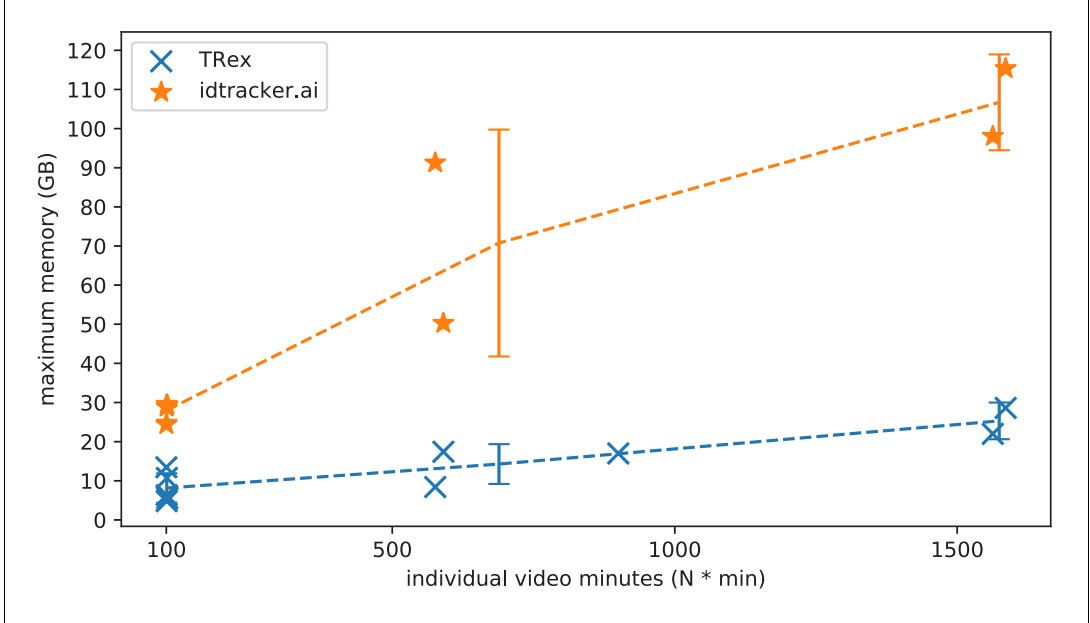

**Figure 5.** The maximum memory by TRex and `idtracker.ai` when tracking videos from a subset of all videos (the same videos as in **Table 3**). Results are plotted as a function of video length (min) multiplied by the number of individuals. We have to emphasize here that, for the videos in the upper length regions of multiple hours (*2*, *2*), we had to set `idtracker.ai` to store segmentation information on disk – as opposed to in RAM. This uses less memory, but is also slower. For the video with flies we tried out both and also settled for on-disk, since otherwise the system ran out of memory. Even then, the curve still accelerates much faster for `idtracker.ai`, ultimately leading to problems with most computer systems. To minimize the impact that hardware compatibility has on research, we implemented switches limiting memory usage while always trying to maximize performance given the available data. TRex can be used on modern laptops and normal consumer hardware at slightly lower speeds, but without any *fatal* issues.

The online version of this article includes the following source data for figure 5:

**Source data 1.** Each data-point from *Figure 5* as plotted, indexed by video and software used.

respectively. There are a number of ways with which to determine the memory usage of a process. For automation purposes, we decided to use a tool called syrupy, which can start and save information about a specified command automatically. We modified it slightly, so we could obtain more accurate measurements for Swap, Shared and Private separately, using ps_mem.

As expected, differences in memory consumption are especially prominent for long videos (4-7x lower maximum memory), and for videos with many individuals (2-3x lower). Since we already experienced significant problems tracking a long video (> 3 hr) of only eight individuals with `idtracker.ai`, we did not attempt to further study its behavior in long videos with many individuals. However, we would expect `idtracker.ai` memory usage to increase even more rapidly than is visible in *Figure 5* since it retains a lot of image data (segmentation/pixels) in memory and we already had to 'allow' it to relay to hard-disk in our efforts to make it work for Videos 8, 14, and 16 (which slows down analysis). The maximum memory consumption across all trials was on average 5.01±2.54 times higher in `idtracker.ai`, ranging from 1.81 to 10.85 times the maximum memory consumption of TRex for the same video (see *Table 4*).

Overall memory consumption for TRex also contains posture data, which contributes a lot to RAM usage. Especially with longer videos, disabling posture can lower the hardware needs for running our software. If posture is to be retained, the user can still (more slightly) reduce memory requirements by changing the outline re-sampling scale (one by default), which adjusts the outline resolution between sub- and super-pixel accuracy. While analysis will be faster – and memory consumption lower – when posture is disabled (only limited by the matching algorithm, see *Appendix 4—figure 3*), users of the visual identification might experience a decrease in training accuracy or speed (see *Figure 6*).

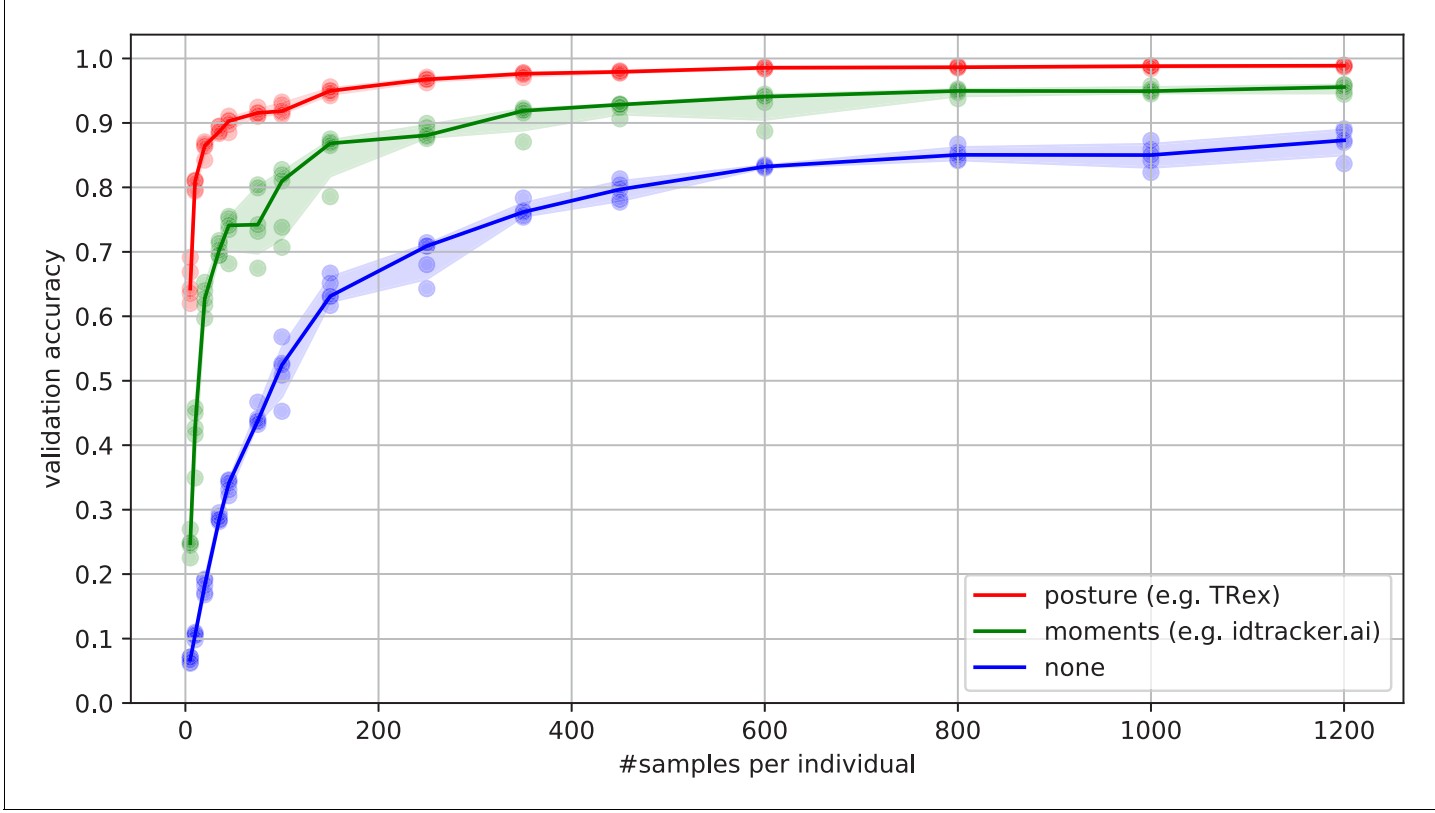

**Figure 6.** Convergence behavior of the network training for three different normalization methods. This shows the maximum achievable validation accuracy after 100 epochs for 100 individuals (Video 7), when sub-sampling the number of examples per individual. Tests were performed using a manually corrected training dataset to generate the images in three different ways, using the same, independent script (see *Figure 8*): Using no normalization (blue), using normalization based on image moments (green, similar to `idtracker.ai`), and using posture information (red, as in TRex). Higher numbers of samples per individual result in higher maximum accuracy overall, but – unlike the other methods – posture-normalized runs already reach an accuracy above the 90 % mark for ≥75 samples. This property can help significantly in situations with more crossings, when longer global segments are harder to find.

The online version of this article includes the following source data for figure 6:

**Source data 1.** Raw data-points as plotted in *Figure 6*.

## Visual identification: processing time

Automatically correcting the trajectories (to produce consistent identity assignments) means that additional time is spent on the training and application of a network, specifically for the video in question. Visual identification builds on some of the other methods described in this paper (tracking and posture estimation), naturally making it by far the most complex and time-consuming process in TRex – we thus evaluated how much time is spent on the entire sequence of all required processes. For each run of TRex and `idtracker.ai`, we saved precise timing information from start to finish. Since `idtracker.ai` reads videos *directly* and preprocesses them again each run, we used the same starting conditions with our software for a direct comparison:

A trial starts by converting/preprocessing a video in TGrabs and then immediately opening it in TRex, where automatic identity corrections were applied. TRex terminated automatically after satisfying a correctness criterion (high uniqueness value) according to equation (Accumulation of additional segments and stopping-criteria). It then exported trajectories, as well as validation data (similar to `idtracker.ai`), concluding the trial. The sum of time spent within TGrabs and TRex gives the total amount of time for that trial. For the purpose of this test it would not have been fair to compare only TRex processing times to `idtracker.ai`, but it is important to emphasize that conversion could be skipped entirely by using TGrabs to record videos directly from a camera instead of opening an existing video file.

**Table 5.** Evaluating time-cost for automatic identity correction – comparing to results from `idtracker.ai`.

Timings consist of preprocessing time in TGrabs plus network training in TRex, which are shown separately as well as combined (*ours (min)*, $N = 5$). The time it takes to analyze videos strongly depends on the number of individuals and how many usable samples per individual the initial segment provides. The length of the video factors in as well, as does the stochasticity of the gradient descent (training). `idtracker.ai` timings ($N = 3$) contain the whole tracking and training process from start to finish, using its terminal_mode (v3). Parameters have been manually adjusted per video and setting, to the best of our abilities, spending at most one hour per configuration. For videos 16 and 14, we had to set `idtracker.ai` to storing segmentation information on disk (as compared to in RAM) to prevent the program from being terminated for running out of memory.

| Video | # ind. | Length | Sample | TGrabs (min) | TRex (min) | Ours (min) | idtracker.ai (min) |
|-------|--------|--------|--------|--------------|------------|------------|--------------------|
| 7 | 100 | 1min | 1.61s | 2.03 ± 0.02 | 74.62 ± 6.75 | 76.65 | 392.22 ± 119.43 |
| 8 | 59 | 10min | 19.46s | 9.28 ± 0.08 | 96.7 ± 4.45 | 105.98 | 495.82 ± 115.92 |
| 9 | 15 | 60min | 33.81s | 13.17 ± 0.12 | 101.5 ± 1.85 | 114.67 | N/A |
| 11 | 10 | 10min | 12.31s | 8.8 ± 0.12 | 21.42 ± 2.45 | 30.22 | 127.43 ± 57.02 |
| 12 | 10 | 10min | 10.0s | 8.65 ± 0.07 | 23.37 ± 3.83 | 32.02 | 82.28 ± 3.83 |
| 13 | 10 | 10min | 36.91s | 8.65 ± 0.047 | 12.47 ± 1.27 | 21.12 | 79.42 ± 4.52 |
| 10 | 10 | 10min | 16.22s | 4.43 ± 0.05 | 35.05 ± 1.45 | 39.48 | N/A |
| 14 | 8 | 195min | 67.97s | 109.97 ± 0.05 | 70.48 ± 3.67 | 180.45 | 707.0 ± 27.55 |
| 15 | 8 | 72min | 79.36s | 32.1 ± 0.42 | 30.77 ± 6.28 | 62.87 | 291.42 ± 16.83 |
| 16 | 8 | 198min | 134.07s | 133.1 ± 2.28 | 68.85 ± 13.12 | 201.95 | 1493.83 ± 27.75 |

The online version of this article includes the following source data for Table 5:

Source data 1. Preprocessed log files (see also notebooks.zip in *Walter et al., 2020*) in a table format. The total processing time (s) of each trial is indexed by video and software used – TGrabs for conversion and TRex and `idtracker.ai` for visual identification. This data is also used in *Appendix 4—table 4*.

In *Table 5*, we can see that video length and processing times (in TRex) did not correlate directly. Indeed, a 1 min video (Video 8) took significantly longer than one that was 60 min long (Video 15). The reason for this, initially counterintuitive, result is that the process of learning identities requires sufficiently long video sequences: longer samples have a higher likelihood of capturing more of the total possible intra-individual variance which helps the algorithm to more comprehensively represent each individual's appearance. Longer videos naturally provide more material for the algorithm to choose from and, simply due to their length, have a higher probability of containing at least one higher quality segment that allows higher uniqueness-regimes to be reached more quickly (see Guiding the training process and H.2 Stopping-criteria). Thus, it is important to use sufficiently long video sequences for visual identification, and longer sequences can lead to better results – both in terms of quality and processing time.

Compared to `idtracker.ai`, TRex (conversion + visual identification) shows both considerably lower computation times (2.57× to 46.74× faster for the same video), as well as lower variance in the timings (79% lower for the same video on average).

## Discussion

We have designed TRex to be a versatile and fast program that can enable researches to track animals (and other mobile objects) in a wide range of situations. It maintains identities of up to 100 untagged individuals and produces corrected tracks, along with posture estimation, visual-field reconstruction, and other features that enable the quantitative study of animal behavior. Even videos that cannot be tracked by other solutions, such as videos with over 500 animals, can now be tracked within the same day of recording.

While all options are available from the command-line and a screen is not required, TRex offers a rich, yet straight-foward to use, interface to local as well as remote users. Accompanied by the integrated documentation for all parameters, each stating purpose, type and value ranges, as well as a comprehensive online documentation, new users are provided with all the information required for a quick adoption of our software. Especially to the benefit of new users, we evaluated the parameter

space using videos of diverse species (fish, termites, locusts) and determined which parameters work best in most use-cases to set their default values.

The interface is structured into groups (see *Figure 5*), categorized by the typical use-case:

1. The main menu, containing options for loading/saving, options for the timeline and reanalysis of parts of the video
2. Timeline and current video playback information
3. Information about the selected individual
4. Display options and an interactive 'omni-box' for viewing and changing parameters
5. General status information about TRex and the Python integration.

The tracking accuracy of TRex is at the state-of-the-art while typically being $2.57\times$ to $46.74\times$ faster than comparable software and having lower hardware requirements – especially RAM. In addition to visual identification and tracking, it provides a rich assortment of additional data, including body posture, visual fields, and other kinematic as well as group-related information (such as derivatives of position, border and mean neighbor distance, group compactness, etc); even in live-tracking and closed-loop situations.

Raw tracking speeds (without visual identification) still achieved roughly 80 % accuracy per decision (as compared to > 99% with visual identification). We have found that real-time performance can be achieved, even on relatively modest hardware, for all numbers of individuals $\leq 256$ without posture estimation ($\leq 128$ with posture estimation). More than 256 individuals can be tracked as well, remarkably still delivering frame-rates at about $10 - 25$ frames per second using the same settings.

Not only does the increased processing-speeds benefit researchers, but the contributions we provide to data exploration should not be underestimated as well – merely making data more easily accessible right out-of-the-box, such as visual fields and live-heatmaps (see *Appendix 1—figure 1*), has the potential to reveal features of group- and individual behavior which have not been visible before. TRex makes information on multiple timescales of events available simultaneously, and sometimes this is the only way to detect interesting properties (e.g. trail formation in termites).

Since the software is already actively used within the Max Planck Institute of Animal Behavior, reported issues have been taken into consideration during development. However, certain theoretical, as well as practically observed, limitations remain:

- Posture: While almost all shapes can be detected correctly (by adjusting parameters), some shapes – especially round shapes – are hard to interpret in terms of 'tail' or 'head'. This means that only the other image alignment method (moments) can be used. However, it does introduce some limitations for example calculating visual fields is impossible.
- Tracking: Predictions, if the wrong direction is assumed, might go really far away from where the object is. Objects are then 'lost' for a fixed amount of time (parameter). This can be 'fixed' by shortening this time-period, though this leads to different problems when the software does not wait long enough for individuals to reappear.
- General: Barely visible individuals have to be tracked with the help of deep learning (e.g. using *Caelles et al., 2017*) and a custom-made mask per video frame, prepared in an external program of the users choosing.
- Visual identification: All individuals have to be *visible* and *separate* at the same time, at least once, for identification to work at all. Visual identification, for example with very high densities of individuals, can thus be very difficult. This is a hard restriction to any software since finding consecutive global segments is the underlying principle for the successful recognition of individuals.

We will continue updating the software, increasingly addressing the above issues (and likely others), as well as potentially adding new features. During development, we noticed a couple of areas where improvements could be made, both theoretical and practical in nature. Specifically, incremental improvements in analysis speed could be made regarding visual identification by using the trained network more sporadically – for example it is not necessary to predict every image of very long consecutive segments, since, even with fewer samples, prediction values are likely to converge to a certain value early on. A likely more potent change would be an improved 'uniqueness' algorithm, which, during the accumulation phase, is better at predicting which consecutive segment

will improve training results the most. This could be done, for example, by taking into account the variation between images of the same individual. Other planned extensions include:

- (Feature): We want to have a more general interface available to users, so they can create their own plugins. Working with the data in live-mode, while applying their own filters. As well as specifically being able to write a plugin that can detect different species/annotate them in the video.
- (Crossing solver): Additional method optimized for splitting overlapping, solid-color objects. The current method, simply using a threshold, is effective for many species but often produces large holes when splitting objects consisting of largely the same color.

To obtain the most up-to-date version of TRex, please download it at trex.run or update your existing installation according to our instructions listed on trex.run/docs/install.html.

## Materials and methods

In the following sections, we describe the methods implemented in TRex and TGrabs, as well as their most important features in a typical order of operations (see *Figure 1* for a flow diagram), starting out with a raw video. We will then describe how trajectories are obtained and end with the most technically involved features.

### Segmentation

When an image is first received from a camera (or a video file), the objects of interest potentially present in the frame must be found and cropped out. Several technologies are available to separate the foreground from the background (segmentation). Various machine learning algorithms are frequently used to great effect, even for the most complex environments (*Hughey et al., 2018*; *Robie et al., 2017*; *Francisco et al., 2019*). These more advanced approaches are typically beneficial for the analysis of field-data or organisms that are very hard to see in video (e.g. very transparent or low contrast objects/animals in the scene). In these situations, where integrated methods might not suffice, it is possible to segment objects from the background using external, for example deep-learning based, tools (see next paragraph). However, for most laboratory experiments, simpler (and also much faster), classical image-processing methods yield satisfactory results. Thus, we provide as a generically useful capability *background-subtraction*, which is the default method by which objects are segmented. This can be used immediately in experiments where the background is relatively static. Backgrounds are generated automatically by uniformly sampling images from the source video(s) – different modes are available (min/max, mode and mean) for the user to choose from. More advanced image-processing techniques like luminance equalization (which is useful when lighting varies between images), image undistortion, and brightness/contrast adjustments are available in TGrabs and can enhance segmentation results – but come at the cost of slightly increased processing time. Importantly, since many behavioral studies rely on $\geq 4$ K resolution videos, we heavily utilize the GPU (if available) to speed up most of the image-processing, allowing TRex to scale well with increasing image resolution.

TGrabs can generally find any object in the video stream, and subsequently pass it on to the tracking algorithm (next section), as long as either (i) the background is relatively static while the objects move at least occasionally, (ii) the objects/animals of interest have enough contrast to the background, or (iii) the user provides an additional binary mask per frame which is used to separate the objects of interest from the background, the typical means of doing this being by deep-learning based segmentation (e.g. *Caelles et al., 2017*). These masks are expected to be in a video-format themselves and correspond 1:1 in length and dimensions to the video that is to be analyzed. They are expected to be binary, marking individuals in white and background in black. Of course, these binary videos could be used on their own, but would not retain grey-scale information of the objects. There are a lot of possible applications where this could be useful; but generally, whenever individuals are really hard to detect visually and need to be recognized by a different software (e.g. a machine-learning-based segmentation like *Maninis et al., 2018*). Individual frames can then be connected using our software as a second step.

The detected objects are saved to a custom non-proprietary compressed file format (Preprocessed Video or PV, see appendix Appendix G The PV file format), that stores only the most

essential information from the original video stream: the objects and their pixel positions and values. This format is optimized for quick random index access by the tracking algorithm (see next section) and stores other meta-information (like frame timings) utilized during playback or analysis. When recording videos directly from a camera, they can also be streamed to an additional and independent MP4 container format (plus information establishing the mapping between PV and MP4 video frames).

## Tracking

Once animals (or, more generally, termed 'objects' henceforth) have been successfully segmented from the background, we can either use the live-tracking feature in TGrabs or open a pre-processed file in TRex, to generate the trajectories of these objects. This process uses information regarding an object's movement (i.e. its kinematics) to follow it across frames, estimating future positions based on previous velocity and angular speed. It will be referred to as 'tracking' in the following text, and is a required step in all workflows.

Note that this approach alone is very fast, but, as will be shown, is subject to error with respect to maintaining individual identities. If that is required, there is a further step, outlined in Automatic visual identification based on machine learning below, which can be applied at the cost of processing speed. First, however, we will discuss the general basis of tracking, which is common to approaches that do, and do not, require identities to be maintained with high-fidelity. Tracking can occur for two distinct categories, which are handled slightly differently by our software:

1. There is a known number of objects
2. There is an unknown number of objects

The first case assumes that the number of tracked objects in a frame cannot exceed a certain expected number of objects (calculated automatically, or set by the user). This allows the algorithm to make stronger assumptions, for example regarding noise, where otherwise 'valid' objects (conforming to size expectations) are ignored due to their positioning in the scene (e.g. too far away from previously lost individuals). In the second case, new objects may be generated until all viable objects in a frame are assigned. While being more susceptible to noise, this is useful for tracking a large number of objects, where counting objects may not be possible, or where there is a highly variable number of objects to be tracked.

For a given video, our algorithm processes every frame sequentially, extending existing trajectories (if possible) for each of the objects found in the current frame. Every object can only be assigned to one trajectory, but some objects may not be assigned to any trajectory (e.g. in case the number of objects exceeds the allowed number of individuals) and some trajectories might not be assigned to any object (e.g. while objects are out of view). To estimate object identities across frames, we use an approach akin to the popular Kalman filter (*Kalman, 1960*) which makes predictions based on multiple noisy data streams (here, positional history and posture information). In the initial frame, objects are simply assigned from top-left to bottom-right. In all other frames, assignments are made based on probabilities (see appendix Appendix D Matching an object to an object in the next frame) calculated for every combination of object and trajectory. These probabilities represent the degree to which the program believes that 'it makes sense' to extend an existing trajectory with an object in the current frame, given its position and speed. Our tracking algorithm only considers assignments with probabilities larger than a certain threshold, generally constrained to a certain proximity around an object assigned in the previous frame.

Matching a set of objects in one frame with a set of objects in the next frame is representative of a typical assignment problem, which can be solved in polynomial time (e.g. using the Hungarian method *Kuhn, 1955*). However, we found that, in practice, the computational complexity of the Hungarian method can constrain analysis speed to such a degree that we decided to implement a custom algorithm, which we term tree-based matching, which has a better *average-case* performance (see evaluation), even while having a comparatively bad *worst-case* complexity. Our algorithm constructs a tree of all possible object/trajectory combinations in the frame and tries to find a compatible (such that no objects/trajectories are assigned twice) set of choices, maximizing the sum of probabilities amongst these choices (described in detail in the appendix Appendix D Matching an object to an object in the next frame). Problematic are situations where a large number of objects are in close proximity of one another, since then the number of possible sets of choices grows

exponentially. These situations are avoided by using a mixed approach: tree-based matching is used most of the time, but as soon as the combinatorical complexity of a certain situation becomes too great, our software falls back on using the Hungarian method. If videos are known to be problematic throughout (e.g. with > 100 individuals consistently very close to each other), the user may choose to use an approximate method instead (described in the appendix Appendix D), which simply iterates through all objects and assigns each to the trajectory for which it has the highest probability and subsequently does not consider whether another object has an even higher probability for that trajectory. While the approximate method scales better with an increasing number of individuals, it is 'wrong' (seeing as it does not consider all possible combinations) – which is why it is not recommended unless strictly necessary. However, since it does not consider all combinations, making it more sensitive to parameter choice, it scales better for very large numbers of objects and produces results good enough for it to be useful in very large groups (see *Appendix 4—table 2*).

Situations where objects/individuals are touching, partly overlapping, or even completely overlapping, is an issue that all tracking solutions have to deal with in some way. The first problem is the *detection* of such an overlap/crossing, the second is its *resolution*. `idtracker.ai`, for example, deals only with the first problem: It trains a neural network to detect crossings and essentially ignores the involved individuals until the problem is resolved by movement of the individuals themselves. However, using such an image-based approach can never be fully independent of the species or even video (it has to be retrained for each specific experiment) while also being time-costly to use. In some cases the size of objects might indicate that they contain multiple overlapping objects, while other cases might not allow for such an easy distinction – for example when sexually dimorphic animals (or multiple species) are present at the same time. We propose a method, similar to `xyTracker` in that it uses the object's movement history to detect overlaps. If there are fewer objects in a region than would be expected by looking at previous frames, an attempt is made to split the biggest ones in that area. The size of that area is estimated using the maximal speed objects are allowed to travel per frame (parameter, see documentation track_max_speed). This, of course, requires relatively good predictions or, alternatively, high frame-rates relative to the object's movement speeds (which are likely necessary anyway to observe behavior at the appropriate time-scales).

By default, objects suspected to contain overlapping individuals are split by thresholding their background-difference image (see appendix Appendix K), continuously increasing the threshold until the expected number (or more) similarly sized objects are found. Grayscale values and, more generally, the shading of three-dimensional objects and animals often produces a natural gradient (see for example *Figure 4*) making this process surprisingly effective for many of the species we tested with. Even when there is almost no visible gradient and thresholding produces holes inside objects, objects are still successfully separated with this approach. Missing pixels from inside the objects can even be regenerated afterwards. The algorithm fails, however, if the remaining objects are too small or are too different in size, in which case the overlapping objects will not be assigned to any trajectory until all involved objects are found again separately in a later frame.

After an object is assigned to a specific trajectory, two kinds of data (posture and visual-fields) are calculated and made available to the user, which will each be described in one of the following subsections. In the last subsection, we outline how these can be utilized in real-time tracking situations.

## Posture analysis

Groups of animals are often modeled as systems of simple particles (*Inada and Kawachi, 2002*; *Cavagna et al., 2010*; *Perez-Escudero and de Polavieja, 2011*), a reasonable simplification which helps to formalize/predict behavior. However, intricate behaviors, like courtship displays, can only be fully observed once the body shape and orientation are considered (e.g. using tools such as DeepPoseKit, *Graving et al., 2019*, LEAP *Pereira et al., 2019*/SLEAP *Pereira et al., 2020*, and DeepLabCut, *Mathis et al., 2018*). TRex does not track individual body parts apart from the head and tail (where applicable), but even the included simple and fast 2D posture estimator already allows for deductions to be made about how an animal is positioned in space, bent and oriented – crucial for example when trying to estimate the position of eyes/antennae as part of an analysis, where this is required (e.g. *Strandburg-Peshkin et al., 2013*; *Rosenthal et al., 2015*). When detailed

tracking of all extremities is required, TRex offers an option that allows it to interface with third-party software like DeepPoseKit (*Graving et al., 2019*), SLEAP (*Pereira et al., 2020*), or DeepLabCut (*Mathis et al., 2018*). This option (output_image_per_tracklet), when set to true, exports cropped and (optionally) normalized videos per individual that can be imported directly into these tools – where they might perform better than the raw video. Normalization, for example, can make it easier for machine-learning algorithms in these tools to learn where body-parts are likely to be (see *Figure 6*) and may even reduce the number of clicks required during annotation.

In TRex, the 2D posture of an animal consists of (i) an outline around the outer edge of a blob, (ii) a center-line (or midline for short) that curves with the body and (iii) positions on the outline that represent the front and rear of the animal (typically head and tail). Our only assumptions here are that the animal is bilateral with a mirror-axis through its center and that it has a beginning and an end, and that the camera-view is roughly perpendicular to this axis. This is true for most animals, but may not hold for example for jellyfish (with radial symmetry) or animals with different symmetries (e.g. radiolaria (protozoa) with spherical symmetry). Still, as long as the animal is not exactly circular from the perspective of the camera, the midline will follow its longest axis and a posture can be estimated successfully. The algorithm implemented in our software is run for every (cropped out) image of an individual and processes it as follows:

i. A tree-based approach follows edge pixels around an object in a clock-wise manner. Drawing the line *around* pixels, as implemented here, instead of through their centers, as done in comparable approaches, helps with very small objects (e.g. one single pixel would still be represented as a valid outline, instead of a single point).

ii. The pointiest end of the outline is assumed, by default, to be either the tail or the head (based on curvature and area between the outline points in question). Assignment of head vs. tail can be set by the user, seeing as some animals might have 'pointier' heads than tails (e.g. termite workers, one of the examples we employ). Posture data coming directly from an image can be very noisy, which is why the program offers options to simplify outline shapes using an Elliptical Fourier Transform (EFT, see *Iwata et al., 2015*; *Kuhl and Giardina, 1982*) or smoothing via a simple weighted average across points of the curve (inspired by common subdivision techniques, see *Warren and Weimer, 2001*). The EFT allows for the user to set the desired level of approximation detail (via the number of elliptic fourier descriptors, EFDs) and thus make it 'rounder' and less jittery. Using an EFT with just two descriptors is equivalent to fitting an ellipse to the animal's shape (as, for example, xyTracker does), which is the simplest supported representation of an animal's body.

iii. The reference-point chosen in (ii) marks the start for the midline-algorithm. It walks both left and right from this point, always trying to move approximately the same distance on the outline (with limited wiggle-room), while at the same time minimizing the distance from the left to the right point. This works well for most shapes and also automatically yields distances between a midline point and its corresponding two points on the outline, estimating thickness of this object's body at this point.

Compared to the tracking itself, posture estimation is a time-consuming process and can be disabled. It is, however, required to estimate – and subsequently normalize – an animal's orientation in space (e.g. required later in Automatic visual identification based on machine learning), or to reconstruct their visual field as described in the following sub-section.

## Reconstructing 2D visual fields

Visual input is an important modality for many species (e.g. fish *Strandburg-Peshkin et al., 2013*, *Bilotta and Saszik, 2001* and humans *Colavita, 1974*). Due to its importance in widely used model organisms like zebrafish (*Danio rerio*), we decided to include the capability to conduct a two-dimensional reconstruction of each individual's visual field as part of the software. The requirements for this are successful posture estimation and that individuals are viewed from above, as is usually the case in laboratory studies.

The algorithm makes use of the fact that outlines have already been calculated during posture estimation. Eye positions are estimated to be evenly distanced from the 'snout' and will be spaced apart depending on the thickness of the body at that point (the distance is based on a ratio, relative to body-size, which can be adjusted by the user). Eye orientation is also adjustable, which influences the size of the stereoscopic part of the visual field. We then use ray-casting to intersect rays from

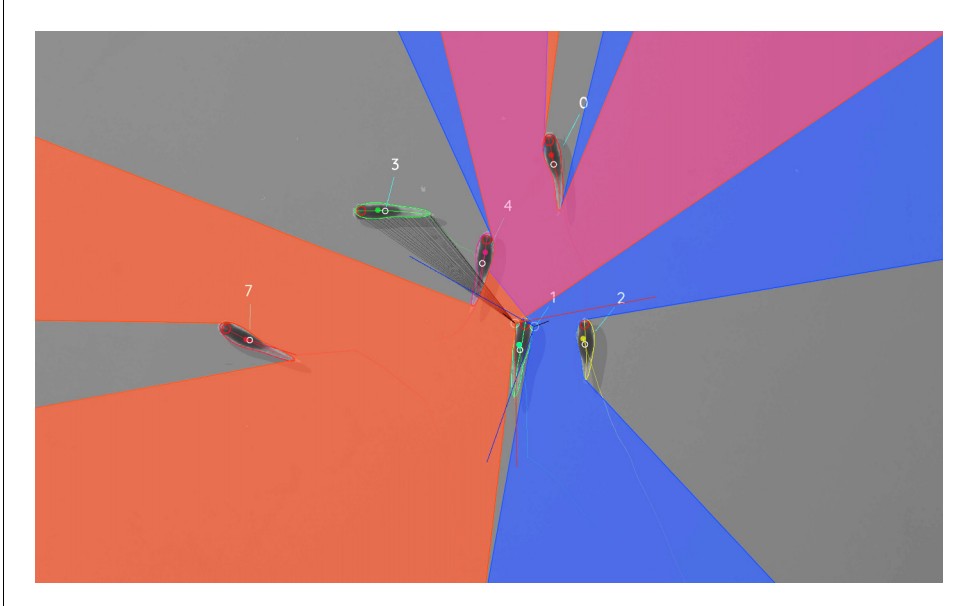

**Figure 7.** Visual field estimate of the individual in the center (zoomed in, the individuals are approximately 2 – 3 cm long, Video 15). Right (blue) and left (orange) fields of view intersect in the binocular region (pink). Most individuals can be seen directly by the focal individual (1, green), which has a wide field of view of 260° per eye. Individual three on the top-left is not detected by the focal individual directly and not part of its first-order visual field. However, second-order intersections (visualized by gray lines here) are also saved and accessible through a separate layer in the exported data.

The online version of this article includes the following video for figure 7:

**Figure 7—video 1.** A clip from Video 15, showing TRex' visual-field estimation for Individual 1 https://youtu.be/yEO_3lpZlzU.

https://elifesciences.org/articles/64000#fig7video1

each of the eyes with all other individuals as well as the focal individual itself (self-occlusion). Individuals not detected in the current frame are approximated using the last available posture. Data are organized as a multi-layered 1D-image of fixed size for each frame, with each image prepresenting angles from −180° to 180° for the given frame. Simulating a limited field-of-view would thus be as simple as cropping parts of these images off the left and right sides. The different layers per pixel encode:

1. identity of the occluder
2. distance to the occluder
3. body-part that was hit (distance from the head on the outline in percent).

While the individuals viewed from above on a computer screen look two-dimensional, one major disadvantage of any 2D approach is, of course, that it is merely a projection of the 3D scene. Any visual field estimator has to assume that, from an individual's perspective, other individuals act as an occluder in all instances (see *Figure 7*). This may only be partly true in the real world, depending on the experimental design, as other individuals may be able to move slightly below, or above, the focal individuals line-of-sight, revealing otherwise occluded conspecifics behind them. We therefore support multiple occlusion-layers, allowing second-order and *N*th-order occlusions to be calculated for each individual.

## Realtime tracking option for closed-loop experiments

Live tracking is supported, as an option to the user, during the recording, or conversion, of a video in TGrabs. When closed-loop feedback is enabled, TGrabs focusses on maintaining stable recording frame-rates and may not track recorded frames if tracking takes too long. This is done to ensure that

the recorded file can later be tracked again in full/with higher accuracy (thus no information is lost) if required, and to help the closed-loop feedback to stay synchronized with real-world events.

During development we worked with a mid-range gaming computer and Basler cameras at 90fps and $2048^2$px resolution, where drawbacks did not occur. Running the program on hardware with specifications below our recommendations (see 2 Results), however, may affect frame-rates as described below.

TRex loads a prepared `Python` script, handing down an array of data per individual in every frame. Which data fields are being generated and sent to the script is selected by the script. Available fields are:

- Position
- Midline information
- Visual field.

If the script (or any other part of the recording process) takes too long to execute in one frame, consecutive frames may be dropped until a stable frame-rate can be achieved. This scales well for all computer-systems, but results in fragmented tracking data, causing worse identity assignment, and reduces the number of frames and quality of data available for closed-loop feedback. However, since even untracked frames are saved to disk, these inaccuracies can be fixed in TRex later. Alternatively, if live-tracking is enabled but closed-loop feedback is disabled, the program maintains detected objects in memory and tracks them in an asynchronous thread (potentially introducing wait time after the recording stops). When the program terminates, the tracked individual's data are exported – along with a `results` file that can be loaded by the `tracker` at a later time.

In order to make this interface easy to use for prototyping and to debug experiments, the script may be changed during its run-time and will be reloaded if necessary. Errors in the `Python` code lead to a temporary pause of the closed-loop part of the program (not the recording) until all errors have been fixed.

Additionally, thanks to `Python` being a fully-featured scripting language, it is also possible to call and send information to other programs during real-time tracking. Communication with other external programs may be necessary whenever easy-to-use `Python` interfaces are not available for for example hardware being used by the experimenter.

## Automatic visual identification based on machine learning

Tracking, when it is only based on individual's positional history, can be very accurate under good circumstances and is currently the fastest way to analyze video recordings or to perform closed-loop experiments. However, such tracking methods simply do not have access to enough information to allow them to ensure identities are maintained for the duration of most entire trials – small mistakes can and will happen. There are cases, for example when studying polarity (only based on short trajectory segments), or other general group-level assessments, where this is acceptable and identities do not have to be maintained perfectly. However, consistent identities are required in many individual-level assessments, and with no baseline truth available to correct mistakes, errors start accumulating until eventually all identities are fully shuffled. Even a hypothetical, *perfect* tracking algorithm will not be able to yield correct results in all situations as multiple individuals might go out of view at the same time (e.g. hiding under cover or just occluded by other animals). There is no way to tell who is whom, once they re-emerge.

The only way to solve this problem is by providing an independent source of information from which to infer identity of individuals, which is of course a principle we make use of all the time in our everyday lives: Facial identification of con-specifics is something that is easy for most humans, to an extent where we sometimes recognize face-like features where there aren't any. Our natural tendency to find patterns enables us to train experts on recognizing differences between animals, even when they belong to a completely different taxonomic order. Tracking individuals is a demanding task, especially with large numbers of moving animals (*Liu et al., 2009* shows humans to be effective for up to four objects). Human observers are able to solve simple memory recall tasks for 39 objects at only 92 % correct (see *Humphrey and Khan, 1992*), where the presented objects do not even have to be identified individually (just classified as old/new) and contain more inherent variation than most con-specific animals would. Even with this being true, human observers are still the most efficient solution in some cases (e.g. for long-lived animals in complex habitats). Enhancing visual inter-

individual differences by attaching physical tags is an effective way to make the task easier and more straight-forward to automate. RFID tags are useful in many situations, but are also limited since individuals have to be in very close proximity to a sensor in order to be detected (**Bonter and Bridge, 2011**). Attaching fiducial markers (such as QR codes) to animals allows for a very large number (thousands) of individuals to be uniquely identified at the same time (see **Gernat et al., 2018**; **Wild et al., 2020**; **Mersch et al., 2013**; **Crall et al., 2015**) – and over a much greater distance than RFID tags. Generating codes can also be automated, generating tags with optimal visual inter-marker distances (**Garrido-Jurado et al., 2016**), making it feasible to identify a large number of individuals with minimal tracking mistakes.

While physical tagging is often an effective method by which to identify individuals, it requires animals to be caught and manipulated, which can be difficult (**Mersch et al., 2013**) and is subject to the physical limitations of the respective system. Tags have to be large enough so a program can recognize it in a video stream. Even worse, especially with increased relative tag-size, the animal's behavior may be affected by the presence of the tag or during its application (**Dennis et al., 2008**; **Pankiw and Page, 2003**; **Sockman and Schwabl, 2001**), and there might be no way for experimenters to necessarily know that it did (unless with considerable effort, see **Switzer and Combes, 2016**). In addition, for some animals, like fish and termites, attachment of tags that are effective for discriminating among a large number of individuals can be problematic, or impossible.

Recognizing such issues, (**Pérez-Escudero et al., 2014**) first proposed an algorithm termed *idtracker*, generalizing the process of pattern recognition for a range of different species. Training an expert program to tell individuals apart, by detecting slight differences in patterning on their bodies, allows the correction of identities without any human involvement. Even while being limited to about 15 individuals per group, this was a very promising approach. It became much improved upon only a few years later by the same group in their software `idtracker.ai` (**Romero-Ferrero et al., 2019**), implementing a paradigm shift from explicit, hard-coded, color-difference detection to using more general machine learning methods instead – increasing the supported group size by an order of magnitude.

We employ a method for visual identification in TRex that is similar to the one used in `idtracker.ai`, where a neural network is trained to visually recognize individuals and is used to correct tracking mistakes automatically, without human intervention – the network layout (see *1* c) is almost the same as well (differing only by the addition of a pre-processing layer and using 2D-instead of 1D-dropout layers). However, in TRex, processing speed and chances of success are improved (the former being greatly improved) by (i) minimizing the variance landscape of the problem and (ii) exploring the landscape to our best ability, optimally covering all poses and lighting-conditions an individual can be in, as well as (iii) shortening the training duration by significantly altering the training process – for example choosing new samples more adaptively and using different stopping-criteria (accuracy, as well as speed, are part of the later evaluation).

While 4.2 Tracking already *tries* to (within each trajectory) consistently follow the same individual, there is no way to ensure/check the validity of this process without providing independent identity information. Generating this source of information, based on the visual appearance of individuals, is what the algorithm for visual identification, described in the following subsections, aims to achieve. Re-stated simply, the goal of using automatic visual identification is to obtain reliable predictions of the identities of all (or most) objects in each frame. Assuming these predictions are of sufficient quality, they can be used to detect and correct potential mistakes made during 4.2 Tracking by looking for identity switches within trajectories. Ensuring that predicted identities within trajectories are consistent, by proxy, also ensures that each trajectory is consistently associated with a single, real individual. In the following, before describing the four stages of that algorithm, we will point out key aspects of how tracking/image data are processed and how we addressed the points (i)-(iii) above and especially highlight the features that ultimately improved performance compared to other solutions.

## Preparing tracking-data

Visual identification starts out only with the trajectories that the 4.2 Tracking provides. Tracking, on its own, is already an improvement over other solutions, especially since (unlike e.g. `idtracker.ai`) TRex makes an effort to separate overlapping objects (see the Appendix K Algorithm for splitting

touching individuals) and thus is able to keep track of individuals for longer (see *Appendix 4—figure 2*). Here, we – quite conservatively – assume that, after every problematic situation (defined in the list below), the assignments made by our tracking algorithm are wrong. Whenever a problematic situation is encountered as part of a trajectory, we split the trajectory at that point. This way, all trajectories of all individuals in a video become an assortment of trajectory snippets (termed 'segments' from here on), which are clear of problematic situations, and for each of which the goal is to find the correct identity ('correct' meaning that identities are consistently assigned to the same *real* individual throughout the video). Situations are considered 'problematic', and cause the trajectory to be split, when:

- The individual has been lost for at least one frame. For example when individuals are moving unexpectedly fast, are occluded by other individuals/the environment, or simply not present anymore (e.g. eaten).
- Uncertainty of assignment was too high (>50%) for example due to very high movement speeds or extreme variation in size between frames. With simpler tracking tasks in mind, these segments are kept as *connected* tracks, but regarded as separate ones here.
- Timestamps suggest skipped frames. Missing frames in the video may cause wrong assignments and are thus treated as if the individuals have been lost. This distinction can only be made if accurate frame timings are available (when recording using TGrabs or provided alongside the video files in separate npz files).

Unless one of the above conditions becomes true, a segment is assumed to be consecutive and connected; that is, throughout the whole segment, no mistakes have been made that lead to identities being switched. Frames where all individuals are currently within one such segment at the same time will henceforth be termed *global segments*.

Since we know that there are no problematic situations inside each per-individual segment, and thus also not across individuals within the range of a global segment, we can choose any global segment as a basis for an initial, arbitrary assignment of identities to trajectories. One of the most important steps of the identification algorithm then becomes deciding which global segment is the best starting point for the training. If a mistake is made here, consecutive predictions for other segments will fail and/or produce unreliable results in general.

Only a limited set of global segments is kept – striking a balance between respecting user-given constraints and capturing as much of the variance as possible. In many of the videos used for evaluation, we found that only few segments had to be considered – however, computation time is ultimately bounded by reducing the number of qualifying segments. While this is true, it is also beneficial to avoid auto-correlation by incorporating samples from all sections of the video instead of only sourcing them from a small portion – to help achieve a balance, global segments are binned by their middle frame into four bins (each quarter of the video being a bin) and then reducing the number of segments inside each bin. With that goal in mind, we sort the segments within bins by their 'quality' – a combination of two factors:

1. To capture as much as possible the variation due to an individual's own movement, as well as within the background that it moves across, a 'good' segment should be a segment where all individuals move as much as possible and also travel as large a distance as possible. Thus, we derive a per-individual *spatial coverage descriptor* for the given segment by dissecting the arena (virtually) into a grid of equally sized, rectangular 'cells' (depending on the aspect ratio of the video). Each time an individual's center-point moves from one cell to the next, a counter is incremented for that individual. To avoid situations where, for example, all individuals but one are moving, we only use the lowest per-individual spatial coverage value to represent a given segment.
2. It is beneficial to have more examples for the network to learn from. Thus, as a second sorting criterion, we use the average number of samples per individual.

After being sorted according to these two metrics, the list of segments per bin is reduced, according to a user-defined variable (four by default), leaving only the most viable options per quarter of video.

The number of visited cells may, at first, appear to be essentially equivalent to a spatially normalized *distance travelled* (as used in `idtracker.ai`). In edge cases, where individuals never stop or always stop, both metrics can be very similar. However, one can imagine an individual continuously

**(a)** No normalization.          **(b)** Using the main body-axis (moments).          **(c)** Using posture information.

**Figure 8.** Comparison of different normalization methods. Images all stem from the same video and belong to the same identity. The video has previously been automatically corrected using the visual identification. Each object visible here consists of $N$ images $M_i, i \in [0, N]$ that have been accumulated into a single image using $\min_{i \in [0,N]} M_i$, with *min* being the element-wise minimum across images. The columns represent same samples from the same frames, but normalized in three different ways: In (a), images have not been normalized at all. Images in (b) have been normalized by aligning the objects along their main axis (calculated using *image-moments*), which only gives the axis within 0– 180 degrees. In (c), all images have been aligned using posture information generated during the tracking process. As the images become more and more recognizable to *us* from left to right, the same applies to a network trying to tell identities apart: Reducing noise in the data speeds up the learning process.

moving around in the same corner of the arena, which would be counted as an equally good segment for that individual as if it had traversed the whole arena (and thus capturing all variable environmental factors). In most cases, using highly restricted movement for training is problematic, and worse than using a shorter segment of the individual moving diagonally through the entire space, since the latter captures more of the variation within background, lighting conditions and the animals movement in the process.

## Minimizing the variance landscape by normalizing samples

A big strength of machine learning approaches is their resistance to noise in the data. Generally, any machine learning method will likely still converge – even with noisy data. Eliminating unnecessary noise and degrees of freedom in the dataset, however, will typically help the network to converge much more quickly: Tasks that are easier to solve will of course also be solved more accurately within similar or smaller timescales. This is due to the optimizer not having to consider various parts of the possible parameter-space during training, or, put differently, shrinking the overall parameter-space to the smallest possible size without losing important information. The simplest such optimization included in most tracking and visual identification approaches is to segment out the objects and centering the individuals in the cropped out images. This means that (i) the network does not have to consider the whole image, (ii) needs only to consider one individual at a time and (iii) the corners of the image can most likely be neglected.

Further improving on this, approaches like `idtracker.ai` align all objects along their most-elongated axis, essentially removing global orientation as a degree of freedom. The orientation of an arbitrary object can be calculated for example using an approach often referred to as image-moments (*Hu, 1962*), yielding an angle within $[0 - 180]°$. Of course, this means that:

1. circular objects have a random (noisy) orientation
2. elongated objects (e.g. fish) can be either head-first or flipped by $180°$ and there is no way to discriminate between those two cases (see second row, *Figure 8*)
3. a C-shaped body deformation, for example, results in a slightly bent axis, meaning that the head will not be in exactly the same position as with a straight posture of the animal.

Each of these issues adds to the things the network has to learn to account for, widening the parameter-space to be searched and increasing computation time. However, barring the first point, each problem can be tackled using the already available posture information. Knowing head and tail positions and points along the individual's center-line, the individual's heads can be locked roughly into a single position. This leaves room only for their rear end to move, reducing variation in the data to a minimum (see *Figure 8*). In addition to faster convergence, this also results in better

generalization right from the start and even with a smaller number of samples per individual (see *Figure 6*). For further discussion of highly deformable bodies, such as of rodents, please see Appendix (Appendix L Posture and Visual Identification of Highly-Deformable Bodies).

## Guiding the training process

Per batch, the stochastic gradient descent is directed by the local accuracy (a fraction of correct/ total predictions), which is a simple and commonly used metric that has no prior knowledge of where the samples within a batch come from. This has the desirable consequence that no knowledge about the temporal arrangement of images is necessary in order to train and, more importantly, to apply the network later on.

In order to achieve accurate results quickly across batches, while at the same time making it possible to indicate to the user potentially problematic sequences within the video, we devised a metric that can be used to estimate local as well as global training quality: We term this uniqueness and it combines information about objects within a frame, following the principle of non-duplication; images of individuals within the same frame are required to be assigned different identities by the networks predictions.

The program generates image data for evenly spaced frames across the entire video. All images of tracked individuals within the selected frames are, after every epoch of the training, passed on to the network. It returns a vector of probabilities $p_{ij}$ for each image $i$ to be identity $j \in [0, N]$, with $N$ being the number of individuals. Based on these probabilities, uniqueness can be calculated as in Box 1, evenly covering the entire video. The magnitude of this probability vector per image is taken

## Box 1. Calculating uniqueness for a frame.

Algorithm 1: The algorithm used to calculate the uniqueness score for an individual frame. Probabilities $\hat{p}i|b$ are predictions by the pre-trained network. During the accumulation these predictions will gradually improve proportional to the global training quality. Multiplying the unique percentage $|\text{uids}|^{-1}|f(x)|$ by the (scaled) mean probability deals with cases of low accuracy, where individuals switch every frame (but uniquely).

**Data**: frame x

**Result**: Uniqueness score for frame x

uids = map{}

$\hat{p}(i|b)$ is the probability of blobb to be identity i

f(x) returns a list of the tracked objects in frame

x $E(v) = (1 + \exp(-\pi))/(1 + \exp(-\pi v))$ is a shift of roughly $+0.5$ and non-linear scaling of values $0 \le v \le 1$.

**for each** object $b \in f(x)$ **do**

 $\text{maxid} = \arg\max \hat{p}(i|b)$ with $i \in$ identities

 **if** maxid $\in$ uids **then**

 $\text{uids[maxid]} = \max(\text{uids[maxid]}, \hat{p}(\text{maxid}, b))$

 **else**

 $\text{uids[maxid]} = \hat{p}(\text{maxid}, b)$

 **end**

**end**

**return** $|\text{uids}|^{-1}|f(x)| * E\left(|\text{uids}|^{-1}\left(\sum_{i \in \text{uids}} \text{uids}[i]\right)\right)$.

into account, rewarding strong predictions of $\max_j\{p_{ij}\} = 1$ and punishing weak predictions of $\max_j\{p_{ij}\} < 1$.

Uniqueness is not integrated as part of the loss function, but it is used as a global gradient before and after each training unit in order to detect global improvements. Based on the average uniqueness calculated before and after a training unit, we can determine whether to stop the training, or whether training on the current segment made our results worse (faulty data). If uniqueness is consistently high throughout the video, then training has been successful and we may terminate early. Otherwise, valleys in the uniqueness curve indicate bad generalization and thus currently missing information regarding some of the individuals. In order to detect problematic sections of the video we search for values below $1 - \frac{0.5}{N}$, meaning that the section potentially contains new information we should be adding to our training data. Using accuracy per-batch and then using uniqueness to determine global progress, we get the best of both worlds: A context-free prediction method that is trained on global segments that are strategically selected by utilizing local context information.

The closest example of such a procedure in `idtracker.ai` is the termination criterion after *protocol 1*, which states that individual segments have to be consistent and certain enough in all global segments in order to stop iterating. While this seems to be similar at first, the way accuracy is calculated and the terminology here are quite different: (i) Every metric in `idtracker.ai`'s final assessment after *protocol one* is calculated at segment-level, not utilizing per-frame information. *Uniqueness* works per-frame, not per segment, and considers individual frames to be entirely independent from each other. It can be considered a much stronger constraint set upon the network's predictive ability, seeing as it basically counts the number of times mistakes are estimated to have happened within single frames. Averaging only happens *afterwards*. (ii) The terminology of identities being unique is only used in `idtracker.ai` once after *procotol one* and essentially as a binary value, not recognizing its potential as a descendable gradient. Images are simply added until a certain percentage of images has been reached, at which point accumulation is terminated. (iii) Testing uniqueness is much faster than testing network accuracy across segments, seeing as the same images are tested over and over again (meaning they can be cached) and the testing dataset can be much smaller due to its locality. *Uniqueness* thus provides a stronger gradient estimation, while at the same time being more local (meaning it can be used independently of whether images are part of global segments), as well as more manageable in terms of speed and memory size.

In the next four sections, we describe the training phases of our algorithm (1-3), and how the successfully trained network can be used to automatically correct trajectories based on its predictions (4).

## The initial training unit

All global segments are considered and sorted by the criteria listed below in Accumulation of additional segments and stopping-criteria. The best suitable segment from the beginning of that set of segments is used as the initial dataset for the network. Images are split into a training and a validation set (4:1 ratio). Efforts are made to equalize the sample sizes per class/identity beforehand, but there has to always be a trade-off between similar sample sizes (encouraging unbiased priors) and having as many samples as possible available for the network to learn from. Thus, in order to alleviate some of the severity of dealing with imbalanced datasets, the performance during training iterations is evaluated using a categorical focal loss function (*Lin et al., 2020*). Focal loss down-weighs classes that are already reliably predicted by the network and in turn emphasizes neglected classes. An Adam optimizer (*Bengio et al., 2015*) is used to traverse the loss landscape towards the global (or to at least a local) minimum.

The network layout used for the classification in TRex (see *Figure 1c*) is a typical Convolutional Neural Network (CNN). The concepts of 'convolutional' and 'downsampling' layers, as well as the back-propagation used during training, are not new. They were introduced in *Fukushima, 1988*, inspired originally by the work of Hubel and Wiesel on cats and rhesus monkeys (*Hubel and Wiesel, 1959*; *Hubel and Wiesel, 1963*; *Wiesel and Hubel, 1966*), describing receptive fields and their hierarchical structure in the visual cortex. Soon afterward, in *LeCun et al., 1989*, CNNs, in combination with back-propagation, were already successfully used to recognize handwritten ZIP codes – for the first time, the learning process was fully automated. A critical step towards making their application practical, and the reason they are popular today.

The network architecture used in our software is similar to the identification module of the network in *Romero-Ferrero et al., 2019*, and is, as in most typical CNNs, (reverse-)pyramid-like. However, key differences between TRex' and `idtracker.ai's` procedure lie with the way that training data is prepared (see previous sections) and how further segments are accumulated/evaluated (see next section). Furthermore, contrary to `idtracker.ai's` approach, images in TRex are augmented (during training) before being passed on to the network. While this augmentation is relatively simple (random shift of the image in x-direction), it can help to account for positional noise introduced by for example the posture estimation or the video itself when the network is used for predictions later on *Perez and Wang, 2017*. We do not flip the image in this step, or rotate it, since this would defeat the purpose of using orientation normalization in the first place (as in Minimizing the variance landscape by normalizing samples, see *Figure 8*). Here, in fact, normalization of object orientation (during training and predictions) could be seen as a superior alternative to data augmentation.

The input data for TRex' network is a single, cropped grayscale image of an individual. This image is first passed through a 'lambda' layer (blue) that normalizes the pixel values, dividing them by half the value limit of $255/2 = 127.5$ and subtracting 1 – this moves them into the range of $[-1, 1]$. From then on, sections are a combination of convolutional layers (kernel sizes of 16, 64, and 100 pixels), each followed by a 2D ($2 \times 2$) max-pooling and a 2D spatial dropout layer (with a rate of 0.25). Within each of these blocks the input data is reduced further, focussing it down to information that is deemed important. Toward the end, the data are flattened and flow into a densely connected layer (100 units) with exactly as many outputs as the number of classes. The output is a vector with values between 0 and 1 for all elements of the vector, which, due to softmax-activation, sum to 1.

Training commences by performing a stochastic gradient descent (using the Adam optimizer, see *Bengio et al., 2015*), which iteratively minimizes the error between network predictions and previously known associations of images with identities – the original assignments within the initial frame segment. The optimizer's behavior in the last five epochs is continuously observed and training is terminated immediately if one of the following criteria is met:

- the maximum number of iterations is reached (150 by default, but can be set by the user)
- a plateau is achieved at a high per-class accuracy
- overfitting/overly optimizing for the training data at the loss of generality
- no further improvements can be made (due to the accuracy within the current training data already being 1).

The initial training unit is also by far the most important as it determines the predicted identities within further segments that are to be added. It is thus less risky to overfit than it is important to get high-quality training results, and the algorithm has to be relatively conservative regarding termination criteria. Later iterations, however, are only meant to extend an already existing dataset and thus (with computation speed in mind) allow for additional termination criteria to be added:

- plateauing at/circling around a certain val_loss level
- plateauing around a certain uniqueness level.

## Accumulation of additional segments and stopping-criteria

If necessary, initial training results can be improved by adding more samples to the active dataset. This could be done manually by the user, always trying to select the most promising segment next, but requiring such manual work is not acceptable for high-throughput processing. Instead, in order to translate this idea into features that can be calculated automatically, the following set of metrics is re-generated per (yet inactive) segment after each successful step:

1. Average uniqueness index (rounded to an integer percentage in 5 % steps)
2. Minimal distance to regions that have previously been trained on (rounded to the next power of two), larger is better as it potentially includes samples more different from the already known ones
3. Minimum *cells visited* per individual (larger is better for the same reason as 2)
4. Minimum average samples per individual (larger is better)
5. Whether its image data has already been generated before (mostly for saving memory)
6. The uniqueness value is smaller than $U_{prev}^2$ after five steps, with $U_{prev}$ being the best uniqueness value previous to the current accumulation step.

With the help of these values, the segment list is sorted and the best segment selected to be considered next. Adding a segment to a set of already active samples requires us to correct the identities inside it, potentially switching temporary identities to represent the same *real* identities as in our previous data. This is done by predicting identities for the new samples using the network that has been trained on the old samples. Making mistakes here can lead to significant subsequent problems, so merely plausible segments will be added – meaning only those samples are accepted for which the predicted IDs are *unique* within each unobstructed sequence of frames for every temporary identity. If multiple temporary individuals are predicted to be the same real identity, the segment is saved for later and the search continues.

If multiple additional segments are found, the program tries to actively improve local uniqueness valleys by adding samples first from regions with comparatively *low* accuracy predictions. Seeing as low accuracy regions will also most likely fail to predict unique identities, it is important to emphasize here that this is generally not a problem for the algorithm: Failed segments are simply ignored and can be inserted back into the queue later. Smoothing the curve also makes sure to prefer regions close to valleys, making the algorithm follow the valley walls upwards in both directions.

Finishing a training unit does not necessarily mean that it was successful. Only the network states improving upon results from previous units are considered and saved. Any training result – except the initial one – may be rejected after training in case the uniqueness score has not improved globally, or at least remained within 99 % of the previous best value. This ensures stability of the process, even with tracking errors present (which can be corrected for later on, see next section). If a segment is rejected, the network is restored to the best recorded state.

Each new segment is always combined with regularly sampled data from previous steps, ensuring that identities don't switch back and forth between steps due to uncertain predictions. If switching did occur, then the uniqueness and accuracy values can never reach high value regimes – leading to the training unit being discarded as a result. The contribution of each previously added segment $R$ is limited to $\lceil |R_S|/(\mathrm{samples\_max} * |R|/N) \rceil$ samples, with $N$ as the total number of frames in global segments for this individual and $\mathrm{samples\_max}$ a constant that is calculated using image size and memory constraints (or 1 GB by default). $R_S$ is the actual *usable* number of images in segment $R$. This limitation is an attempt to not bias the priors of the network by sub-sampling segments according to their contribution to the total number of frames in global segments.

Training is considered to be successful globally, as soon as either (i) accumulative individual gaps between sampled regions is less than 25 % of the video length for all individuals, or (ii) uniqueness has reached a value higher than $1 - \frac{0.5}{N_{\mathrm{id}}}$ (1) so that almost all detected identities are present exactly once per frame. Otherwise, training will be continued as described above with additional segments – each time extending the percentage of images seen by the network further.

Training accuracy/consistency could potentially be further improved by letting the program add an arbitrary amount of segments, however we found this not to be necessary in any of our test-cases. Users are allowed to set a custom limit if required in their specific cases.

## The final training unit

After the accumulation phase, one last training step is performed. In previous steps, validation data has been kept strictly separate from the training set to get a better gauge on how generalizable the results are to unseen parts of the video. This is especially important during early training units, since 'overfitting' is much more likely to occur in smaller datasets and we still potentially need to add samples from different parts of the video. Now that we are not going to extend our training dataset anymore, maintaining generalizability is no longer the main objective – so why not use *all* of the available data? The entire dataset is simply merged and sub-sampled again, according to the memory strategy used. Network training is started, with a maximum of $\max\{3; \mathrm{max\_epochs} * 0.25\}$ iterations (max_epochs is 150 by default). During this training, the same stopping-criteria apply as during the initial step.

Even if we tolerate the risk of potentially overfitting on the training data, there is still a way to detect overfitting if it occurs: Only training steps that lead to improvements in mean uniqueness across the video are saved. Often, if prediction results become worse (e.g. due to overfitting), multiple individuals in a single frame are predicted to be the same identity – precisely the problem which our uniqueness metric was designed to detect.

For some videos, this is the step where most progress is made (e.g. Video 9). The reason being that this is the first time when all the training data from all segments is considered at once (instead of mostly the current segment plus fewer samples from previously accepted segments), and samples from all parts of the video have an equal likelihood of being used in training after possible reduction due to memory-constraints.

## Assigning identities based on network predictions

After the network has been successfully trained, all parts of the video which were not part of the training are packaged together and the network calculates predictive probabilities for each image of each individual to be any of the available identities. The vectors returned by the network are then averaged per consecutive segment per individual. The average probability vectors for all overlapping segments are weighed against each other – usually forcing assignment to the most likely identity (ID) for each segment, given that no other segments have similar probabilities. When referring to segments here, meant is simply a number of consecutive frames of one individual that the tracker is fairly sure does *not* contain any mix-ups. We implemented a way to detect tracking mistakes, which is mentioned later.

If an assignment is ambiguous, meaning that multiple segments $S_{j...M}$ overlapping in time have the same maximum probability index $\underset{i \in [0,N]}{arg\,max} \left\{ P(i|S_j) \right\}$ for the segment to belong to a certain identity (i), a decision has to be made. Assignments are deferred if the ratio

$$R_{\max} = \max\left\{ \frac{P(i|S_j)}{P(i|S_k)}, \forall S_{j \neq k} \in \text{overlapping segments} \right\}$$

between any two maximal probabilities is *larger than* 0.6 for said $i$ ($R_{\max}$ is inverted if it is greater than 1). In such a case, we rely on the general purpose tracking algorithm to pick a sensible option – other identities might even be successfully assigned (using network predictions) in following frames, which is a complexity we do not have to deal with here. In case all ratios are *below* 0.6, when the best choices per identity are not too ambiguous, the following steps are performed to resolve remaining conflicts:

1. Count the number of samples $N_{me}$ in the current segment, and the number of samples $N_{he}$ in the other segment that this segment is compared to
2. Calculate average probability vectors $P_{me}$ and $P_{he}$
3. If $S(P_{me}, N_{me}) \geq S(P_{he}, N_{he})$, then assign the current segment with the ID in question. Otherwise assign the ID to the other segment. Where:

$$\text{norm}(x) = \frac{x}{N_{me} + N_{he}}, \text{sig}(x) = \left( 1 + e^{2\pi(0.5-x)} \right)^{-1}$$
$$S(p,x) = \text{sig}(p) + \text{sig}(\text{norm}(x)). \tag{2}$$

This procedure prefers segments with larger numbers of samples over segments with fewer samples, ensuring that identities are not switched around randomly whenever a short segment (e.g. of noisy data) is predicted to be the given identity for a few frames – at least as long as a better alternative is available. The non-linearity in $S(p,x)$ exaggerates differences between lower values and dampens differences between higher values: For example, the quality of a segment with 4000 samples is barely different from a segment with 5000 samples; however, there is likely to be a significant quality difference between segments with 10 and 100 samples.

In case something goes wrong during the tracking, for example an individual is switched with another individual without the program knowing that it might have happened, the training might still be successful (for example if that particular segment has not been used for training). In such cases, the program tries to correct for identity switches mid-segment by calculating a running-window median identity throughout the whole segment. If the identity switches for a significant length of time, before identities are assigned to segments, the segment is split up at the point of the first change within the window and the two parts are handled as separate segments from then on.

## Software and licenses

TRex is published under the GNU GPLv3 license (see here for permissions granted by GPLv3). All the codes have been written by the first author of this paper (a few individual lines of code from other sources have been marked inside the code). While none of these libraries are distributed alongside TRex (they have to be provided separately), the following libraries are used: OpenCV (opencv.org) is a core library, used for all kinds of image manipulation. GLFW (glfw.org) helps with opening application windows and maintaining graphics contexts, while DearImGui (github.com/ocornut/imgui) helps with some more abstractions regarding graphics. pybind11 (*Jakob et al., 2017*) for Python integration within a C++ environment. miniLZO (oberhumer.com/opensource/lzo) is used for compression of PV frames. Optional bindings are available to FFMPEG (ffmpeg.org) and libpng libraries, if available. (optional) GNU Libmicrohttpd (gnu.org/software/libmicrohttpd), if available, can be used for an HTTP interface of the software, but is non-essential.

## Acknowledgements

We thank A Albi, F Nowak, H Hugo, D E Bath, F Oberhauser, H Naik, J Graving, I Etheredge for helping with their insights, by providing videos, for comments on the manuscript, testing the software and for frequent coffee breaks during development. The development of this software would not have been possible without them. We thank D Mink and M Groettrup providing additional video material of mice. We thank the reviewers and editors for their constructive and useful comments and suggestions. IDC acknowledges support from the NSF (IOS-1355061), the Office of Naval Research grant (ONR, N00014-19-1-2556), the Struktur- und Innovationsfunds für die Forschung of the State of Baden-Württemberg, the Deutsche Forschungsgemeinschaft (DFG, German Research Foundation) under Germany's Excellence Strategy–EXC 2117 – 422037984, and the Max Planck Society.

## Additional information

### Funding

| Funder | Grant reference number | Author |
| --- | --- | --- |
| Division of Integrative Organismal Systems | IOS-1355061 | Iain D Couzin |
| Office of Naval Research | N00014-19-1-2556 | Iain D Couzin |
| Deutsche Forschungsgemeinschaft | EXC 2117-422037984 | Iain D Couzin |
| Max-Planck-Gesellschaft | | Iain D Couzin |
| Struktur- und Innovationsfunds fuer die Forschung of the State of Baden-Wuerttemberg | | Iain D Couzin |

The funders had no role in study design, data collection and interpretation, or the decision to submit the work for publication.

### Author contributions

Tristan Walter, Conceptualization, Data curation, Software, Formal analysis, Investigation, Visualization, Methodology, Writing - original draft, Project administration, Writing - review and editing; Iain D Couzin, Conceptualization, Resources, Supervision, Funding acquisition, Project administration, Writing - review and editing

### Author ORCIDs

Tristan Walter https://orcid.org/0000-0001-8604-7229
Iain D Couzin https://orcid.org/0000-0001-8556-4558

### Ethics

Animal experimentation: We herewith confirm that the care and use of animals described in this work is covered by the protocols 35-9185.81/G-17/162, 35-9185.81/G-17/88 and 35-9185.81/G-16/116 granted by the Regional Council of the State of Baden-Württemberg, Freiburg, Germany, to the Max Planck Institute of Animal Behavior in accordance with the German Animal Welfare Act (TierSchG) and the Regulation for the Protection of Animals Used for Experimental or Other Scientific Purposes (Animal Welfare Regulation Governing Experimental Animals - TierSchVersV).

### Decision letter and Author response
Decision letter https://doi.org/10.7554/eLife.64000.sa1
Author response https://doi.org/10.7554/eLife.64000.sa2

## Additional files

### Supplementary files
• Transparent reporting form

### Data availability

Video data that has been used in the evaluation of TRex has been deposited in MPG Open Access Data Repository (Edmond), under the Creative Commons BY 4.0 license, at https://dx.doi.org/10.17617/3.4y Most raw videos have been trimmed, since original files are each up to 200GB in size. Pre-processed versions (in PV format) are included, so that all steps after conversion can be reproduced directly (conversion speeds do not change with video length, so proportional results are reproducible as well). Full raw videos are made available upon reasonable request. All analysis scripts, scripts used to process the original videos, and the source code/pre-compiled binaries (linux-64) that have been used, are archived in this repository. Most intermediate data (PV videos, log files, tracking data, etc.) are included, and the binaries along with the scripts can be used to automatically generate all intermediate steps. The application source code is available for free under https://github.com/mooch443/trex. Videos 11, 12 and 13 are part of idtracker.ai's example videos: URL https://drive.google.com/file/d/1pAR6oJjrEn7jf_OU2yMdyT2UJZMTNoKC/view?usp=sharing (10_zebrafish.tar.gz) [Francisco Romero, 2018, Examples for idtracker.ai, Online, Accessed 23-Oct-2020]; Video 7 (video_example_100fish_1min.avi): URL https://drive.google.com/file/d/1Tl64CHrQoc05PDElHvYGzjqtybQc4g37/view?usp=sharing [Francisco Romero, 2018, Examples for idtracker.ai, Online, Accessed 23-Oct-2020]; V1 from Appendix 12: https://drive.google.com/drive/folders/1Nir2fzgxofz-fcojEiG_JCNXsGQXj_9k [Francisco Romero, 2018, Examples for idtracker.ai, Online, Accessed 09-Feb-2021].

The following dataset was generated:

| Author(s) | Year | Dataset title | Dataset URL | Database and Identifier |
|---|---|---|---|---|
| Walter T, Albi A, Bath D, Hugo H, Oberhauser F, Mink D, Groettrup M | 2020 | Reproduction Data for: TRex, a fast multi-animal tracking system with markerless identification, and 2D estimation of posture and visual fields | https://edmond.mpdl.mpg.de/imeji/collection/eVbVH0_57TwQsAe8 | MPDL Edmond, eVbVH0_57TwQsAe8 |

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

## Appendix 1

### Installation requirements and usage

Compiled, ready-to-use binaries are available for all major operating systems (Windows, Linux, MacOS). However, it should be possible to compile the software yourself for any Unix- or Windows-based system ($\geq 8$), possibly with minor adjustments. Tested setups include:

- Windows, Linux, MacOS
- A computer with $\geq 16\text{GB}$ RAM is recommended
- OpenCV(opencv.org) libraries $\geq v3.3$
- Python libraries $\geq v3.6$, as well as additional packages such as:
- Keras $\approx v2.2$ with one of the following backends installed
    - Tensorflow $<v2$ (tensorflow.org) (either CPU-based, or GPU-based)
    - Theano (deeplearning.net)
- GPU-based recognition requires an NVIDIA graphics-card and drivers (see Tensorflow documentation).

For detailed download/installation instructions and up-to-date requirements, please refer to the documentation at trex.run/install.

### Workflow

TRex can be opened in one of two ways: (i) Simply starting the application (e.g. using the operating systems' file-browser), (ii) using the command-line. If the user simply opens the application, a file opening dialog displays a list of compatible files as well as information on a selected files content. Certain startup parameters can be adjusted from within the graphical user-interface (see *Figure 8*, "interactive settings box"), before confirming and loading up the file (see *Appendix 1—figure 2*). Users with more command-line experience, or the intent of running TRex in batch-mode, can append necessary parameter values without adding them to a settings file.

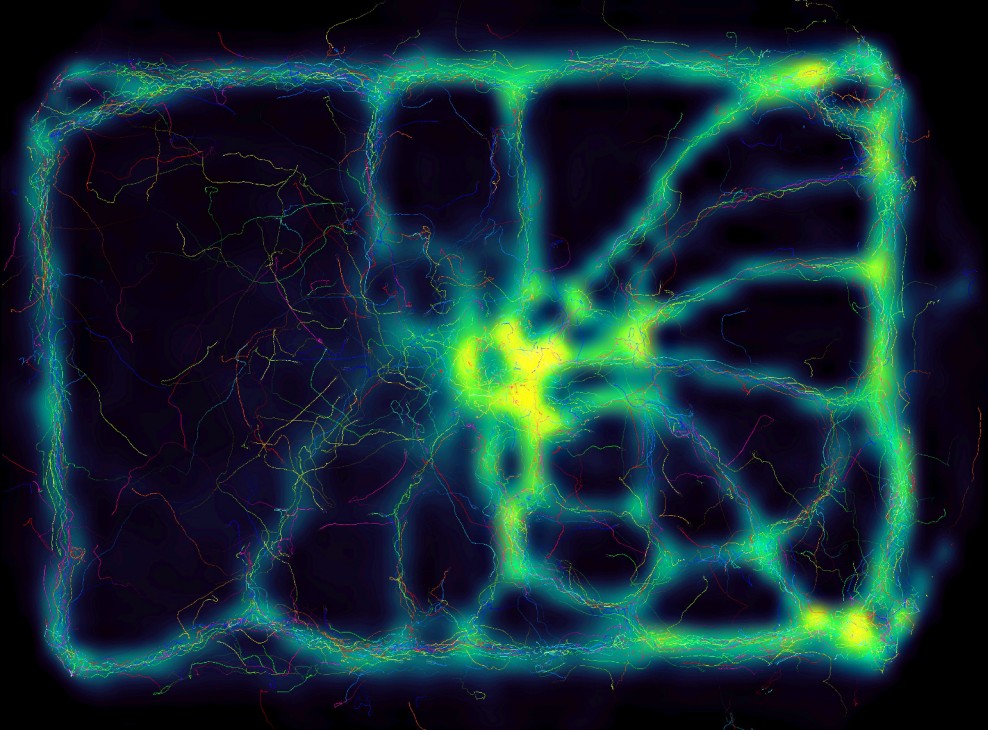

**Appendix 1—figure 1.** Using the interactive heatmap generator within TRex, the foraging trail formation of *Constrictotermes cyphergaster* (termites) can be visualized during analysis, as well as other potentially interesting metrics (based on posture- as well basic positional data). This is generalizable

*Appendix 1—figure 1 continued on next page*

*Appendix 1—figure 1 continued*

to all output data fields available in TRex, for example also making it possible to visualize 'time' as a heatmap and showing where individuals were more likely to be located during the beginning or towards end of the video. *Video: H. Hugo.*

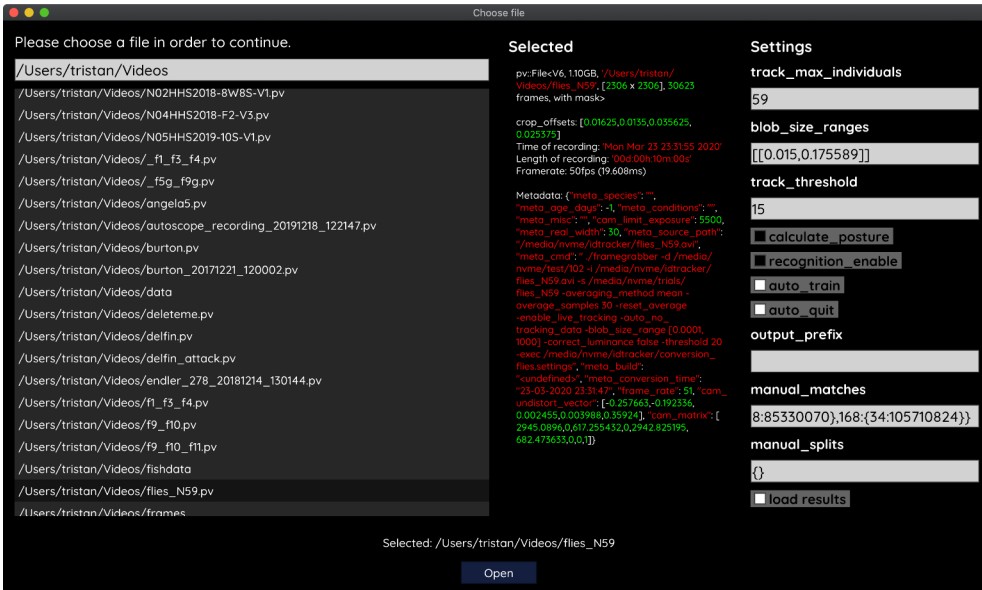

**Appendix 1—figure 2.** The file opening dialog. On the left is a list of compatible files in the current folder. The center column shows meta-information provided by the video file, including its frame-rate and resolution – or some of the settings used during conversion and the timestamp of conversion. The column on the right provides an easy interface for adjusting the most important parameters before starting up the software. Most parameters can be changed later on from within TRex as well.

To acquire video-files that can be opened using TRex, one needs to first run TGrabs in one way or another. It is possible to use a webcam (generic USB camera) for recording, but TGrabs can also be compiled with Basler Pylon5 support The baslerweb.com Pylon SDK is required to be installed to support Basler USB cameras. TGrabs can also convert existing videos and write to a more suitable format for TRex to interact with (a static background with moving objects clearly separated in front of it). It can be started just like TRex, although most options are either set via the command-line, or a web-interface. TGrabs can perform basic tracking tasks on the fly, offering closed-loop support as well.

For automatic visual recognition, one might need to adjust some parameters. Mostly, these adjustments consist of changing the following parameters:

- blob_size_ranges: Setting one (or multiple) size thresholds for individuals, by giving lower and upper limit value pairs.
- track_max_individuals: Sets the number of individuals expected in a trial. This number needs to be known for recognition tasks (and will be guessed if not provided), but can be set to 0 for unknown numbers of individuals.
- track_max_speed: Sets the maximum speed (cm/s) that individuals are expected to travel at. This is influenced by meta information provided to TGrabs by the user (e.g. the width of the tank), as well as frame timings.
- track_threshold: Even TRex can threshold images of individuals, so it is beneficial to not threshold away too many pixels during conversion/recording and do finer-grade adjustments in the tracker itself.
- outline_resample: A factor that is $gt$ $0$, by which the number of points in the outline is essentially 'divided'. Smaller resample rates lead to more points on the outline (good for very small shapes).

Training can be started once the user is satisfied with the basic tracking results. Consecutive segments are highlighted in the time-line and suggest better or worse tracking, based on their quantity and length. Problematic segments of the video are highlighted using yellow bars in that same timeline, giving another hint to the user as to the tracking quality. To start the training, the user just clicks on 'train network' in the main menu – triggering the accumulation process immediately. After training, the user can click on 'auto correct' in the menu and let TRex correct the tracks automatically (this will re-track the video). The entire process can be automated by adding the 'auto_train' parameter to the command-line, or selecting it in the interface.

## Output

Once finished, the user may export the data in the desired format. Which parts of the data are exported is up to the user as well. By default, almost all the data is exported and saved in NPZ files in the output folder.

Output folders are structured in this way:

output folder:
- Settings files
- Training weights
- Saved program states
- data folder:
    - Statistics
    * All exported NPZ files (named [video_name]_fish[number].npz – the prefix 'fish' can be changed).
    * . . .
- frames folder (contains video clips recorded in the GUI, for example for presentations):
    * [video name] folder
        · clip[index].avi
        · . . .
    * . . .

At any point in time (except during training), the user can save the current program state and return to it at a later time (e.g. after a computer restart).

### Export options

After individuals have been assigned by the matching algorithm, various metrics are calculated (depending on settings):

- Angle: The angle of an individual can be calculated without any context using image moments (*Hu, 1962*). However, this angle is only reliable within 0 to 180 degrees – not the full 360. Within these 180 degrees it is probably more accurate than is movement direction.
- Position: Centroid information on the current, as well as the previous position of the individual are maintained. Based on previous positions, velocity as well as acceleration are calculated. This process is based on information sourced from the respective video file or camera on the time passed between frames. The centroid of an individual is calculated based on the mass center of the pixels that the object comprises. Angles calculated in the previous steps are corrected (flipped by 180 degrees) if the angle difference between movement direction and angle + 180 degrees is smaller than with the raw angle.
- Posture: A large part of the computational complexity comes from calculating the posture of individuals. While this process is relatively fast in TRex, it is still the main factor (except with many individuals, where the matching process takes longest). We dedicated a subsection to it below.
- Visual Field: Based on posture, rays can be cast to detect which animal is visible from the position of another individual. We also dedicated a subsection to visual field further down.
- Other features can be computed, such as inter-individual distances or distance to the tank border. These are optional and will only be computed if necessary when exporting the data. A (non-comprehensive) list of metrics that can be exported follows:
    - Time: The time of the current frame (relative to the start of the video) in seconds.
    - Frame: Index of the frame in the PV video file.

–Individual components of position its derivatives (as well as their magnitudes, e.g. speed)
–Midline offset: The center-line, for example of a beating fish-tail, is normalized to be roughly parallel to the x-axis (from its head to a user-defined percentage of a body). The y-offset of its last point is exported as a 'midline offset'. This is useful, for example to detect burst-and-glide events.
–Midline variance: Variance in midline offset, for example for detection of irregular postures or increased activity.
–Border distance
–Average neighbor distance: Could be used to detect individuals who prefer to be located far away from the others or are avoided by them.

Additionally, tracks of individuals can be exported as a series of cropped-out images – a very useful tool if they are to be used with an external posture estimator or tag-recognition. This series of images can be either every single image, or the median of multiple images (the time-series is downsampled).

## Appendix 2

### From video frame to blobs

Video frames can originate either from a camera, or from a pre-recorded video file saved on disk. TGrabs treats both sources equally, the only exception being some minor details and that pre-recorded videos have a well-defined end (which only has an impact on MP4 encoding). Multiple formats are supported, but the full list of supported codecs depends on the specific system and OpenCV version installed. TGrabs saves images in RAW quality, but does not store complete images. Merely the objects of interest, defined by common tracking parameters such as size, will actually be written to a file. Since TGrabs is mostly meant for use with stable backgrounds (except when contrast is good or a video-mask is provided), the rest of the area can be approximated by a static background image generated in the beginning of the process (or previously).

Generally, every image goes through a number of steps before it can be tracked in TRex:

1. Images are decoded by either (i) a camera driver, or (ii) OpenCV. They consist of an array of values between 0 and 255 (grayscale). Color images will be converted to grayscale images (color channel or 'hue' can be chosen).
2. Timing information is saved and images are appended to a queue of images to be processed
3. All operations from now on are performed on the GPU if available. Once images are in the queue, they are picked one-by-one by the processing thread, which performs operations on them based on user-defined parameters:
   - Cropping
   - –Inverting
   - Contrast/brightness and lighting corrections
   - –Undistortion (see OpenCV Tutorial)
4. (optional) Background subtraction ($d(x) = b(x) - f(x)$, with $f$ being the image and $b$ the background image), leaving a difference image containing only the objects. This can be an absolute difference $|b(x) - f(x)|$ or a signed one, which has different effects on the following step. Otherwise $d(x) = f(x)$
5. Thresholding to obtain a binary image, with all pixels either being 1 or 0:

$$t(x) = \begin{cases} 0 & d(x) < T \\ 1 & d(x) \geq T \end{cases}$$

   where $0 \leq T \leq 255$ is the threshold constant.
6. Options are available for further adjustment of the binary image: Dilation, Erosion and Closing are used to close gaps in the shapes, which are filled up by successive dilation and erosion operations (see *Appendix 2—figure 1*). If there is an imbalance of dilation and erosion commands, noise can be removed or shapes made more inclusive.
7. The original image is multiplied by the thresholded image, obtaining a masked grayscale image: $t(x) \cdot f(x)$, where $\cdot$ is the element-wise multiplication operator.

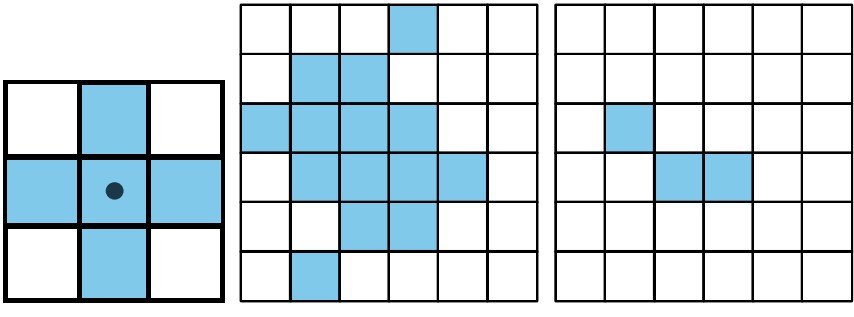

(a) The structure element.  (b) The original image.  (c) Image modified by the structure element.

*Appendix 2—figure 1 continued on next page*

**Appendix 2—figure 1.** Example of morphological operations on images: 'Erosion'. Blue pixels denote on-pixels with color values greater than zero, white pixels are 'off-pixels' with a value equal to zero. A mask is moved across the original image, with its center (dot) being the focal pixel. A focal pixel is *retained* if all the on-pixels within the structure element/mask are on top of on-pixels in the original image. Otherwise the focal pixel is set to 0. The type of operation performed is entirely determined by the structure element.

At this point, the masked image is returned to the CPU, where connected components (objects) are detected. A connected component is a number of adjacent pixels with color values greater than zero. Algorithms for connected-component labeling either use a 4-neighborhood or an 8-neighborhood, which considers diagonal neighbors to be adjacent as well. Many such algorithms are available (*AbuBaker et al., 2007*, *Chang and Chen, 2003*, and many others), even capable of real-time speeds (*Suzuki et al., 2003*, *He et al., 2009*). However, since we want to use a compressed representation throughout our solution, as well as transfer over valuable information to integrate it with posture analysis, we needed to implement our own (see Appendix C Connected components algorithm).

MP4 encoding has some special properties, since its speed is mainly determined by the external encoding software. Encoding at high-speed frame-rates can be challenging, since we are also encoding to a PV-file simultaneously. Videos are encoded in a separate thread, without muxing, and will be remuxed after the recording is stopped. For very high frame-rates or resolutions, it may be necessary to limit the duration of videos since all of the images have to be kept in RAM until they have been encoded. RAW images in RAM can take up a lot of space ($1024*1024*1000 = 1,048,576,000$ bytes for 1000 images quite low in resolution). If there a recording length is defined prior to starting the program, or a video is converted to PV and streamed to MP4 at the same time (though it is unclear why that would be necessary), TGrabs is able to automatically determine which frame-rate can be maintained reliably and without filling the memory.

## Appendix 3

### Connected components algorithm

Pixels are not represented individually in TRex. Instead, they are saved as connected horizontal line segments. For each of these lines, only y- as well as start- and end-position are saved ($y, x_0$ and $x_1$). This representation is especially suited for objects stretching out along the x-axis, but of course its worst-case is a straight, vertical line – in which case space requirements are $O(2 * N)$ for $N$ pixels. Especially for big objects, however, only a fraction of coordinates has to be kept in memory (with a space requirement of $O(2 * H)$ instead of $O(W * H)$, with $W, H$ being width and height of the object).

Extracting these connected horizontal line segments from an image can be parallelized easily by cutting the image into full-width pieces and running the following algorithm repeatedly for each row:

1. From 0 to $W$, iterate all pixels. Always maintain the previous value (binary), as well as the current value. We start out with our previous value of $\bar{p} = 0$ (the border is considered not to be an object).
2. Now repeat for every pixel $p_i$ in the current row:
   a. If $\bar{p}$ is one and $p_i$ is 0, set $\bar{p} := 0$ and save the position as the end of a line segment $x_1 = i - 1$.
   b. If $\bar{p}$ is 0 and $p_i$ is 1, we did not have a previous line segment and a new one starts. We save it as our current line segment with $x_0$ and $y$ equal to the current row. Set $\bar{p} := 1$.
3. After each row, if we have a valid current line, we save it in our array of lines. If $\bar{p} = 1$ was set, and the line segment ended at the border $W$ of the image, we first set its end position to $x_1 := W - 1$.

We keep the array of extracted lines sorted by their y-coordinate, as well as their x-coordinates in the order we encountered them. To extract connected components, we now just need to walk through all extracted rows and detect changes in the y-coordinate. The only information needed are the current row and the previous row, as well as a list of active preliminary 'blobs' (or connected components). A blob is simply a collection of ordered horizontal line segments belonging to a single connected component. These blobs are preliminary until the whole image has been processed, since they might be merged into a single blob further down despite currently being separate (see *Appendix 3—figure 1*).

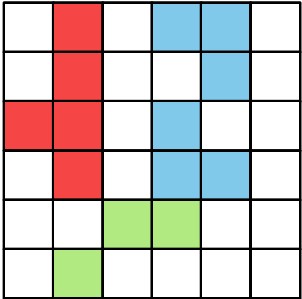

**Appendix 3—figure 1.** An example array of pixels, or image, to be processed by the connected components algorithm. This figure should be read from top to bottom, just as the connected components algorithm would do. When this image is analyzed, the red and blue objects will temporarily stay separate within different 'blobs'. When the green pixels are reached, both objects are combined into one identity.

'Rows' are an array of horizontal lines with the same y-coordinate, ordered by their x-coordinates (increasing). The following algorithm only considers pairs of previous row $R_{i-1}$ and current row $R_i$. We start by inserting all separate horizontal line segments of the very first row into the pool of active blobs, each assigned their own blob. Lines within row $R_i$ are $L_{i,j}$. Coordinates of $L_{i,j}$ will be denoted as $x_0(i,j)$, $x_1(i,j)$ and $y(i,j)$. Our current index in row $R_{i-1}$ is $j$ and our index in row $R_i$ is $k$. We initialize $j := 0, k := 1$. Now for each pair of rows, three different actions may be required depending on the case at hand. All three actions are hierarchically ordered and mutually exclusive (like a typical if/else

structure would be), meaning that cases $0 - 2$ can be true at the same time while no other combination can be simultaneously true:

1. Case 0,1 and 2: We have to create a new blob. This is the case if (0) the line in $R_i$ ends before the line in $R_{i-1}$ starts ($x_1(i, k) + 1 < x_0(i, j)$), or (1) y-coordinates of $R_i$ and $R_{i-1}$ are farther apart than 1 ($y(i-1, j) > y(i, k) + 1$), or (2) there are no lines left in $R_{i-1}$ to match the current line in $R_i$ to ($j \geq |R_{i-1}|$). $L_{i,k}$ is assigned with a new blob.
2. Case 3: Segment in the previous row ends before the segment in the current row starts. If $x_0(i, k) > x_1(i-1, j) + 1$, then we just have to $j := j + 1$.
3. Case 4: Segment in the previous row and segment in the current row intersect in x-coordinates. If $L_{i,k}$ is no yet assigned with a blob, assign it with the one from $L_{i-1,j}$. Otherwise, both blobs have to be merged. This is done in a sub-routine, which guarantees that lines within blobs stay properly sorted during merging. This means that (i) y-coordinates increase or stay the same and (ii) x-coordinates increase monotonically. Afterwards, we increase either $k$ or $j$ based on which one associated line ends earlier:
   If $x_1(i, k) \leq x_1(i-1, j)$, then we increase $k := k + 1$; otherwise $j := j + 1$.

After the previous algorithm has been executed on a pair of $R_{i-1}$ and $R_i$, we increase $i$ by one $i := i + 1$. This process is continued until $i = H$, at which point all connected components are contained within the active blob array.

Retaining information about pixel values adds slightly more complexity to the algorithm, but is straight-forward to implement. In TRex, horizontal line segments comprise $y$, $x_0$ and $x_1$ values plus an additional pointer. It points to the start of a line within array of all pixels (or an image matrix), adding only little computational complexity overall.

Based on the horizontal line segments and their order, posture analysis can be sped up when properly integrated. Another advantage is that detection of connected components within arrays of horizontal line segments is supported due to the way the algorithm functions – we can just get rid of the extraction phase.

## Appendix 4

### Matching an object to an object in the next frame

#### Terminology

A graph is a mathematical structure commonly used in many fields of research, such as computer science, biology and linguistics. Graphs are made up of vertices, which in turn are connected by edges. Below we define relevant terms that we are going to use in the following section:

- Directed graph: Edges have a direction assigned to them
- Weighted edges: Edges have a weight (or cost) assigned to them
- Adjacent nodes: Nodes which are connected immediately by an edge
- Path: A path is a sequence of edges, where each edges starting vertex is the end vertex of the previous edge
- Acyclic graph: The graph contains no path in which the same vertex appears more than once
- Connected graph: There are no vertices without edges, there is a path from any vertex to any other vertex in the graph
- Bipartite graph: Vertices can be sorted into two distinct groups, without an edge from any vertex to elements of its own group – only to the other group
- Tree: A tree is a connected, undirected, acyclic graph, in which any two vertices are only connected by exactly one path
- Rooted, directed out-tree: A tree where one vertex has been defined to be the root and directed edges, with all edges flowing away from the root
- Visited vertex: A vertex that is already part of the current path
- Leaf: A vertex which has only one edge arriving, but none going out (in a tree this are the bottom-most vertices)
- Depth-first/breadth-first and best-first search: Different strategies to pick the next vertex to explore for a set of paths with traversable edges. Depth-first prefers to first go deeper inside a graph/tree, before going on to explore other edges of the same vertex. Breadth-first is the opposite of depth-search. Best-first search uses strategies to explore the most promising path first.

#### Background

The transportation problem is one of the fundamental problems in computer science. It solves the problem of transporting a finite number of *goods* to a finite number of *factories*, where each possible transport route is associated with a *cost* (or weight). Every factory has a *demand* for goods and every good has a limited *supply*. The sum of this cost has to be minimized (or benefits maximized), while remaining within the constraints given by supply and demand. In the special case where demand by each factory and supply for each good are exactly equal to 1, this problem reduces to the *assignment problem*.

The assignment problem can be further separated into two distinct cases: the *balanced* and the *unbalanced* assignment problem. In the balanced case, net-supply and demand are the same – meaning that the number of factories matches exactly the number of suppliers. While the balanced case can be solved slightly more efficiently, most practical problems are usually unbalanced (*Ramshaw and Tarjan, 2012b*). Thankfully, unbalanced assignments can be reduced to balanced assignments, for example using graph-duplication methods or by adding nodes (*Ramshaw and Tarjan, 2012a*, *Ramshaw and Tarjan, 2012a*). This makes the widely used Hungarian method (*Kuhn, 1955*; *Munkres, 1957*) a viable solution to both, with a computational complexity of $O(n^3)$. It can be further improved using Fibonacci heaps (not implemented in TRex), resulting in $O(ms + s^2 \log n)$ time-complexity (*Fredman and Tarjan, 1987*), with $m$ being the number of possible connections/edges, $s \leq n$ the number of factories to be supplied and $n$ the number of factories. Re-balancing, by adding nodes or other structures, also adds computational cost – especially when $s \ll n$ (*Ramshaw and Tarjan, 2012b*).

## Adaptation for our matching problem

1382 assigning individuals to objects in the frame is, in the worst case, exactly that: an unbalanced assignment problem – potentially with $r \neq s$. During development, we found that we can achieve better average-complexity by combining an approach commonly used to solve *NP-hard* problems. This is a class problems for which it is (probably) not possible to find a polynomial-time solution. In order to motivate our usage of a less stable algorithm than for example the Hungarian method, let us first introduce a more general algorithm, following along with remarks for adapting it to our special case. The next subsection concludes with considerations regarding its complexity in comparison to the more stable Hungarian method.

*Branch and Bound* (or BnB, *Land and Doig, 2010*, formalized in *Little et al., 1963*) is a very general approach to traversing the large search spaces of *NP-hard* problems, traditionally represented by a tree. Branching and bounding gives optimal solutions by traversing the entire search space if necessary, but stopping along the way to evaluate its options, always trying to choose better branches of the tree to explore next or skip unnecessary ones. BnB always consists of three main ingredients:

1. Branching: The division of our problem into smaller, partial problems
2. Bounding: Estimate the upper/lower limits of the probability/cost gain to be expected by traversing a given edge
3. Selection: Determining the next node to be processed.

Finding good strategies is essential and can have a big impact on overall computation time. Strategies can only be worked out with insight into the specific problem, but *bounding* is generally the dominating factor here – in that choosing good selection and branching techniques cannot make up for a bad bounding function (*Clausen, 1999*). A bounding function estimates an upper (or lower) limit for the quality of results that can be achieved within a given sub-problem (current branch of the tree).

The 'problem' is the entire assignment problem located at the root node of the tree. The further down we go in the tree, the smaller the partial problems become until we reach a leaf. Any graph can be represented as a tree by duplicating nodes when necessary (*Weixiong, 1996*, 'Graph vs. tree'). So even if the bipartite assignment graph (an example sketched in *Appendix 4—figure 1*) is a more 'traditional' representation of the assignment problem, we can translate it into a rooted, directed out-tree $T = (U, V, E, F)$ with weighted edges. Here, $U$ are individuals and $V$ are objects in the current frame that are potentially assigned to identities in $U$. $E$ are edges mapping from $U \rightarrow V$, while $F : V \rightarrow U$. It is quite visible from *Appendix 4—figure 1*, that the representation as a tree (b) is much more verbose than a bipartite graph (a). However, its structure is very simple:

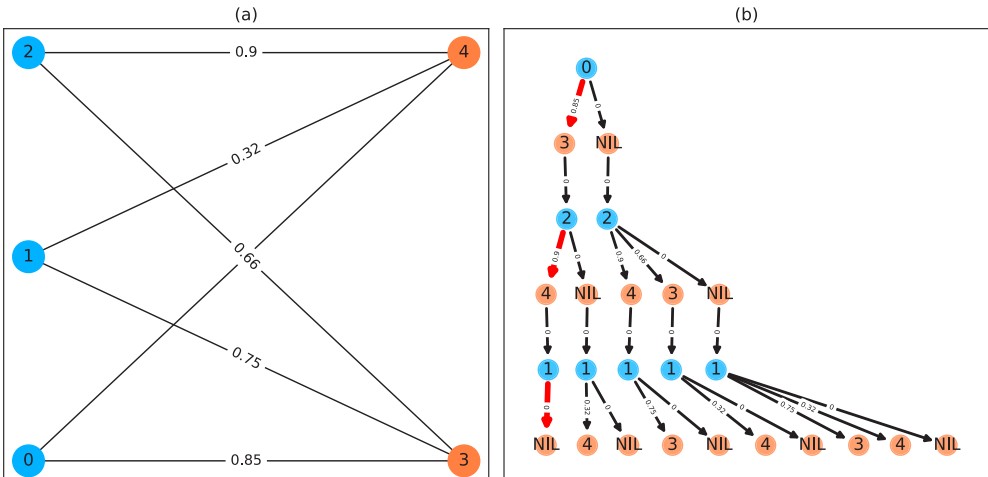

**Appendix 4—figure 1.** A bipartite graph (a) and its equivalent tree-representation (b). It is *bipartite* since nodes can be sorted into two disjoint and independent sets ($\{0, 1, 2\}$ and $\{3, 4\}$), where no nodes have edges to other nodes within the same set. (a) is a straight-forward way of depicting an
*Appendix 4—figure 1 continued on next page*

*Appendix 4—figure 1 continued*

assignment problem, with the identities on the left side and objects being assigned to the identities on the right side. Edge weights are, in TRex and this example, probabilities for a given identity to be the object in question. This graph is also an example for an unbalanced assignment problem, since there are fewer objects (orange) available than individuals (blue). The optimal solution in this case, using weight-maximization, is to assign $0 \rightarrow 3; 2 \rightarrow 4$ and leave one unassigned. Invalid edges have been pruned from the tree in (b), enforcing the rule that objects can only appear once in each path. The optimal assignments have been highlighted in red.

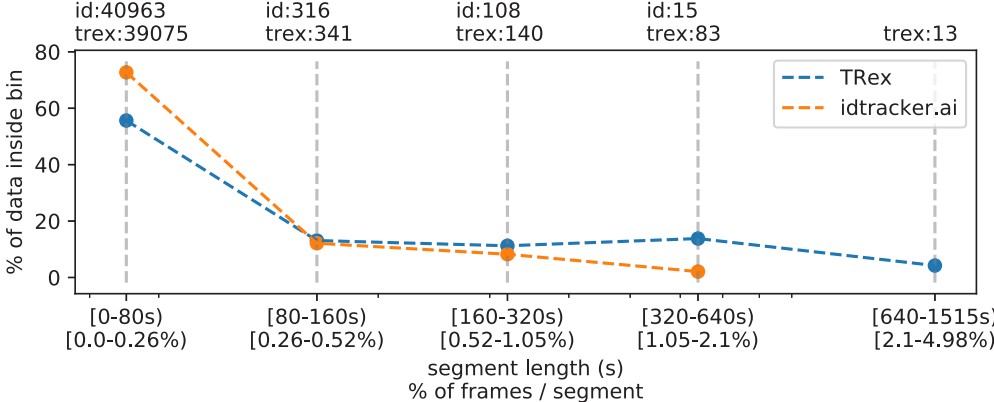

**Appendix 4—figure 2.** The same set of videos as in *Table 5* pooled together, we evaluate the efficiency of our crossings solver. *Consecutive frame segments* are sequences of frames without gaps, for example due to crossings or visibility issues. We find these *consecutive frame segments* in data exported by TRex, and compare the distribution of segment-lengths to `idtracker.ai`'s results (as a reference for an algorithm without a way to resolve crossings). In `idtracker.ai`'s case, we segmented the non-interpolated tracks by missing frames, assuming tracks to be correct in between. The Y-axis shows the percentage of $\sum_{k \in [1,V]} \text{video\_length}_k * \#\text{individuals}_k$ in $V$ videos that one column makes up for – the overall coverage for TRex was 98%, while `idtracker.ai` was slightly worse with 95.17%. Overall, the data distribution suggests that, probably due to it attempting to resolve crossings, TRex seems to produce longer consecutive segments.

The online version of this article includes the following source data is available for figure 2:

**Appendix 4—figure 2—source data 1.** A list of all consecutive frame segments used in *Appendix 4—figure 2*.

**Appendix 4—figure 2—source data 2.** The raw data-points as plotted in *Appendix 4—figure 2*.

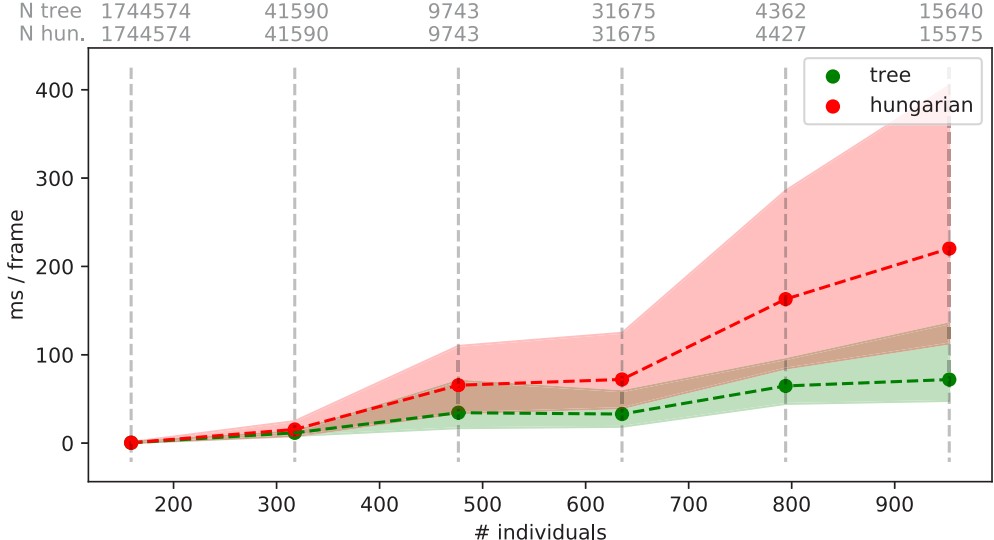

**Appendix 4—figure 3.** Mean values of processing-times and 5 %/95 % percentiles for video frames of all videos in the *speed dataset* (*Table 1*), comparing two different matching algorithms. Parameters were kept identical, except for the matching mode, and posture was turned off to eliminate its effects on performance. Our tree-based algorithm is shown in green and the Hungarian method in red. Grey numbers above the graphs show the number of samples within each bin, per method. Differences between the algorithms increase very quickly, proportional to the number of individuals. Especially the Hungarian method quickly becomes very computationally intensive, while our tree-based algorithm shows a much shallower curve. Some frames could not be solved in reasonable time by the tree-based algorithm alone, at which point it falls back to the Hungarian algorithm. Data-points belonging to these frames ($N = 79$) have been excluded from the results for both algorithms. One main advantage of the Hungarian method is that, with its bounded worst-case complexity (see Appendix D Matching an object to an object in the next frame), no such combinatorical explosions can happen. However, even given this advantage the Hungarian method still leads to significantly lower processing speed overall (see also *Appendix 4—table 3*).

The online version of this article includes the following source data is available for figure 3:

**Appendix 4—figure 3—source data 1.** Raw data for producing this figure and *Appendix 4—table 3*.

Looking at the tree in *Appendix 4—figure 1(b)*, individuals (blue) are found along the y-axis/ deeper into the tree while objects in the frame (orange) are listed along on the x-axis. This includes a 'null' case per individual, representing the possibility that it is *not* assigned to any object – ensuring that every individual has at least one edge.

Tree is never generated in its entirety (except in extreme cases), but it represents all *possible* combinations of individuals and objects. Overall, the set $Q$ of every complete and valid path from top to bottom would be exactly the same as the set of every valid permutation of pairings between objects (plus null) and individuals. Edge weights in $E$ are equal to the probability $P_i t, \tau_i | B_j$ (see *Equation 7*), abbreviated to $P_i(B_j)$ here since we are only ever looking at one time-step. $B_j$ is an object and $i$ is an individual, so we can rewrite it in the current context as $P_u(v)$, with $u \in U; v \in V$.

We are maximizing the objective function

$$o(\rho) = \sum_{uv \in \rho} P_u(v),$$

where $\rho \in Q$ is an element of all valid paths within $T$.

The simplest approach would be to traverse every edge in the graph and accumulate a sum of probabilities along each path, guaranteeing to find the optimal solution eventually. Since the number of possible combinations $|U|^{|E|}$ grows rapidly with the number of edges, this is not realistic – even with few individuals. Thus, at least the *typical* number of visited edges has to be minimized. While we do not know the exact solution to our problem before traversing the graph, we can make

very good guesses. For example, we may order nodes in such a way that branching (visiting a node leads to $gt_1$ new edges to be visited) is reduced in most cases. To do that, we first need to calculate the *degree* of each individual. The degree $C_u$ of individual $u$, which is exactly equivalent to the maximum number of edges going out from that individual, we define as

$$C_u \in \mathbb{N} := \sum_{u \in U} \begin{cases} 1 & \text{if} P_u(v) > P\text{min} \\ 0 & \text{otherwise} \end{cases}.$$

The maximally probable edge per individual also has to be computed beforehand, defined as

$$\overline{P_u} = \max_{v \in V}\{P_u(v)\}.$$

Nodes are sorted first by their degree (ascending) and secondly by $\overline{P_u}$ (descending). We call this ordered set *S*. Sorting by degree ensures that the nodes with the fewest outgoing edges are visited *first*, causing severe branching to only happen in the lower regions of the tree. This is preferable, because a new branch in the bottom layer merely results in a few more options. If this happens at the top, the tree is essentially duplicated $C_u$ times – in one step drastically increasing the overall number of items to be kept in memory. This process is, fittingly, called *node sorting* (**Weixiong, 1996**). Sorting by $\overline{P_u}$ is only applied whenever nodes of the same degree have to be considered.

We always follow the most promising paths first (the one with the highest accumulated probability), which is called 'best-first search' (BFS) – our selection strategy for (1.) in D.2.1. BFS is implemented using a queue maintaining the list of all currently expanded nodes.

Regarding (2.) in D.2.1, we utilize $\overline{P_u}$ as an approximation for the upper bound to the achievable probability in each vertex. For each layer with vertices of $U$, we calculate an accumulative sum $\text{upper\_limit}(i) = \sum_{j > i \in U} \overline{P_j}$, with $j, i$ being indices into our ordered set $S$ of individuals and $i$ being the current depth in the graph (only counting vertices of $U$). This hierarchical upper limit for the expected value does not consider whether the respective edges are still *viable*, so they could have been eliminated already by assigning the object of $V$ to another vertex of $U$ above the current one. Any edge with $P_\text{current} + \text{upper\_limit}(i) < P_\text{best}$ is skipped since it can not improve upon our previous best value $P_\text{best}$. If we do find an edge with a better value, we replace $P_\text{best}$ with the new value and continue.

As an example, let us traverse the tree in *Appendix 4—figure 1b*:

- We first calculate $\overline{P_u}$ for every $u \in U$ ($\overline{P_0} = 0.85; \overline{P_2} = 0.9; \overline{P_1} = 0.75$), as well as the hierarchical probability table $\text{upper\_limit}(i)$ for each index $0 \leq i < N$ ($0.9 + 0.75; 0.75; 0$). $P_\text{best} := 0$.
- Individual 0 (the root) is expanded, which has one edge with probability $0.85 + \text{upper\_limit}(0) \geq P_\text{best}$ to object 3 (plus the null case) and is the only node with a degree of 1. We know that our now expanded node is the best, since it has the largest probability due to sorting, plus also is the deepest. In fact, this is true for all expanded nodes exactly in the order they are expanded (depth-first search $==$ best-first search for our case). We set $P_\text{best} := 0.85$. The edge to NIL is added to our queue.
- Objects in $V$ are only virtual and always have zero-probability connections to the next individual in an ordered set ($f \in F$), so they do not add to the overall probability sum. We skip to the next node.
- Individual 2 branches off into one or two different edges, depending on which edges have been chosen previously.
- We first explore the edge towards object four with a probability of $0.9 + \text{upper\_limit}(1) = 1.65 \geq P_\text{best}$ and add it to $P_\text{best}$.
- Only one possibility is left and we arrive at a leaf with an accumulated probability of $0.85 + 0.9 + 0 = 1.75$.
- We now perform backtracking, meaning we look at every expanded node in our queue, each time observing $\overline{P_u} + \text{upper\_limit}(i)$.
  - NIL (from node 2) would be added to the front of our queue, however its probability $0.85 + 0 + \text{upper\_limit}(1) = 1.6 < 1.75 = P_\text{best}$, so it is discarded.
  - NIL (from node 0) would be added now, but its probability of $0 + \text{upper\_limit}(0) = 1.65 < P_\text{best}$, so it is also discarded.

We can see that with increased depth, we have to keep track of more and more possibilities. Since our nodes and edges are pre-sorted, our path through the tree is optimal after exactly $N = |U|$ node expansions (not counting $v \in V$ expansions since they are only 'virtual').

## Complexity

Utilizing these techniques, we can achieve very good average-case complexity. Of course having a good worst-case complexity is important (such as the Hungarian method), but the impact of a good average-case complexity can be significant as well. This is illustrated nicely by the timings measured in Table *Appendix 4—table 3*, where our method consistently surpasses the Hungarian method in terms of performance – especially for very large groups of animals – despite having worse worst-case complexity. Usually, even in situations with over 1000 individuals present, the average number of leaves visited was approximately 1.112 (see Table *Appendix 4—table 5*) and each visit was a global improvement (not shown). The number of nodes visited per frame were around 2844 to $19,804,880$ in the same video, which, given the maximal number of possible combinations $N^M$ for $M$ edges and $N$ individuals (*Thomas, 2016*), is quite moderate. Especially considering the number of calculations that the Hungarian method has to perform in every step, which, according to its complexity, will be in the range of $N^3 \approx 1\mathrm{e}9$ for $N = 1024$ individuals.

The average complexity of a solution using best-first-search BnB is given by *Weixiong, 1996*. It depends on the probability of encountering a 'zero-cost edge' $p_0$, as well as the mean branching factor $b$ of the tree:

1. $\Theta(\beta^N)$ when $bp_0 < 1$, with $\beta \leq b$ and $N$ is the depth of the tree
2. $\Theta(N^2)$ when $bp_0 = 1$
3. $\Theta(N)$ when $bp_0 > 1 \Leftrightarrow b > 1/p_0$
4. as $N \to \infty$.

In our case, the depth of the tree is exactly the number of individuals $N$, which we have already substituted here. This is the number of nodes that have to be visited in the best case. A 'zero-cost edge' is an edge that does not add any cost to the current path. We are maximizing (not minimizing) so in our case this would be 'an edge with a probability of 1'. While reaching exactly one is improbable, it is (in our case) equivalent to 'having only one viable edge arriving at an object'. $p_0$ depends very much on the settings, specifically the maximum movement speed allowed, and behavior of individuals, which is why in scenarios with $gt_{100}$ individuals the maximum speed should always be adjusted first. To put it another way: If there are only few branching options available for the algorithm to explore per individual, which seems to be the case even in large groups, we can assume our graph to have a probability $p_0$ within $0 \ll p_0 \leq 1$. The mean branching factor $b$ is given by the mean number of edges arriving at an object (not an individual). Averaging at around $b \approx k + 1$, with $k \geq 1$ being the average number of assignable blobs per individual (roughly 1.005 in Video 0) and one the null-case, we can assume $bp_0$ to be $gt_1$ on average. An average complexity of $O(N^2)$, as long as $b > 1/p_0$, is even better than the complexity of the Hungarian method (which is also $O(N^3)$ in the average-case, *Bertsekas, 1981*), giving a possible explanation for the good results achieved using tree-based matching in TRex on average (Table *Appendix 4—table 3*).

Further optimizations could be implemented, for example using impact-based heuristics (as an example of dynamic variable ordering) instead of the static and coarse maximum probability estimate used here. Such heuristics first choose the vertex 'triggering the largest search space reduction' (*Pesant et al., 2012*). In our case, assigning an individual first if, for example, it has edges to many objects that each only one other individual is connected to.

**Appendix 4—table 1.** Showing quantiles for frame timings for videos of the *speed dataset* (without posture enabled).

Video 15, 16, and 14 each contain a short sequence of taking out the fish, causing a lot of big objects and noise in the frame. This leads to relatively high spikes in these segments of the video, resulting in high peak processing timings here. Generally, processing time is influenced by a lot of factors involving not only TRex, but also the operating system as well as other programs. While we did try to control for these, there is no way to make sure. However, having sporadic spikes in the timings per frame does not significantly influence overall processing time, since it can be compensated for by later

frames. We can see that videos of all quantities ≤256 individuals can be processed faster than they could be recorded. Videos that can not be processed faster than real-time are underlaid in gray.

| Video characteristics | | | Ms / frame (processing) | | | | | Processing time |
|---|---|---|---|---|---|---|---|---|
| Video | # ind. | Ms / frame | 5% | Mean | 95 % | Max | > real-time | % video length |
| 0 | 1024 | 25.0 | 46.93 | 62.96 | 119.54 | 849.16 | 100.0% | 358.12 |
| 1 | 512 | 20.0 | 19.09 | 29.26 | 88.57 | 913.52 | 92.11% | 259.92 |
| 2 | 512 | 16.67 | 17.51 | 26.53 | 36.72 | 442.12 | 97.26% | 235.39 |
| 3 | 256 | 20.0 | 8.35 | 11.28 | 13.25 | 402.54 | 1.03% | 77.18 |
| 4 | 256 | 16.67 | 8.04 | 11.62 | 13.48 | 394.75 | 1.13% | 94.77 |
| 5 | 128 | 16.67 | 3.54 | 5.14 | 5.97 | 367.92 | 0.41% | 40.1 |
| 6 | 128 | 16.67 | 3.91 | 5.64 | 6.89 | 381.51 | 0.51% | 44.38 |
| 7 | 100 | 31.25 | 2.5 | 3.57 | 5.19 | 316.75 | 0.1% | 28.35 |
| 8 | 59 | 19.61 | 1.43 | 2.29 | 3.93 | 2108.77 | 0.19% | 16.33 |
| 9 | 15 | 40.0 | 0.4 | 0.52 | 1.67 | 4688.5 | 0.01% | 2.96 |
| 10 | 10 | 10.0 | 0.28 | 0.33 | 0.57 | 283.7 | 0.07% | 8.08 |
| 11 | 10 | 31.25 | 0.21 | 0.25 | 0.65 | 233.7 | 0.01% | 3.48 |
| 12 | 10 | 31.25 | 0.23 | 0.27 | 0.75 | 225.63 | 0.02% | 2.82 |
| 13 | 10 | 31.25 | 0.22 | 0.25 | 0.54 | 237.32 | 0.02% | 2.64 |
| 14 | 8 | 33.33 | 0.24 | 0.29 | 0.66 | 172.8 | 0.02% | 1.8 |
| 15 | 8 | 40.0 | 0.22 | 0.26 | 0.88 | 244.88 | 0.01% | 1.5 |
| 16 | 8 | 28.57 | 0.18 | 0.21 | 0.51 | 1667.14 | 0.02% | 1.38 |
| 17 | 1 | 7.14 | 0.03 | 0.04 | 0.06 | 220.81 | 0.01% | 1.56 |

The online version of this article includes the following source data for Table Appendix 4—table 1.:
**Appendix 4—table 1—Source data 1.** Raw samples for this table and *Appendix 4—table 5*.

**Appendix 4—table 2.** A quality assessment of assignment decisions made by the general purpose tracking system without the aid of visual recognition – comparing results of two accurate tracking algorithms with the assignments made by an approximate method.

Here, *decisions* are reassignments of an individual after it has been lost, or the tracker was too 'unsure' about an assignment. Decisions can be either correct or wrong, which is determined by comparing to reference data generated using automatic visual recognition: Every segment of frames between decisions is associated with a corresponding 'baseline-truth' identity from the reference data. If this association changes after a decision, then that decision is counted as wrong. Analysing a decision may fail if no good match can be found in the reference data (which is not interpolated). Failed decisions are ignored. Comparative values for the Hungarian algorithm (*Kuhn, 1955*) are always exactly the same as for our tree-based algorithm, and are therefore not listed separately. Left-aligned *total*, *excluded* and *wrong* counts in each column are results achieved by an accurate algorithm, numbers to their right are the corresponding results using an approximate method. Raw data of trial runs using the hungarian and tree-based matching algorithms, as well as baseline data from manually or automatically corrected trials used in this table is available for download from *Walter et al., 2020* (in A4T2_source_data.zip).

| Video | # ind. | Length | Total | | Excluded | | Wrong | |
|---|---|---|---|---|---|---|---|---|
| 7 | 100 | 1 min | 717 | 755 | 22 | 22 | 45 (6.47%) | 65 (8.87%) |
| 8 | 59 | 10 min | 279 | 312 | 146 | 100 | 55 (41.35%) | 32 (16.09%) |
| 9 | 15 | 1h0min | 838 | 972 | 70 | 111 | 100 (13.02%) | 240 (27.87%) |
| 13 | 10 | 10min3s | 331 | 337 | 22 | 22 | 36 (11.65%) | 54 (17.14%) |

*Continued on next page*

*Appendix 4—table 2 continued*

| Video | # ind. | Length | Total | | Excluded | | Wrong | |
|---|---|---|---|---|---|---|---|---|
| 12 | 10 | 10min3s | 382 | 404 | 42 | 43 | 83 (24.41%) | 130 (36.01%) |
| 11 | 10 | 10min10s | 1067 | 1085 | 50 | 52 | 73 (7.18%) | 92 (8.91%) |
| 14 | 8 | 3h15min22s | 7424 | 7644 | 1428 | 1481 | 1174 (19.58%) | 1481 (24.03%) |
| 15 | 8 | 1h12min | 3538 | 3714 | 427 | 517 | 651 (20.93%) | 962 (30.09%) |
| 16 | 8 | 3h18min13s | 2376 | 3305 | 136 | 206 | 594 (26.52%) | 1318 (42.53%) |
| sum | | | 16952 | 16754 | -2343 | -2554 | 2811 (19.24%) | 4374 (27.38%) |

**Appendix 4—table 3.** Comparing computation speeds of the tree-based tracking algorithm with the widely established Hungarian algorithm *Kuhn, 1955*, as well as an approximate version optimized for large quantities of individuals.

Posture estimation has been disabled, focusing purely on the assignment problem in our timing measurements. The tree-based algorithm is programmed to fall back on the Hungarian method whenever the current problem 'explodes' computationally – these frames were excluded. Listed are relevant video metrics on the left and mean computation speeds on the right side for three different algorithms: (1) The tree-based and (2) the approximate algorithm presented in this paper, and (3) the Hungarian algorithm. Speeds listed here are percentages of real-time (the videos' fps), demonstrating usability in closed-loop applications and overall performance. Results show that increasing the number of individuals both increases the time-cost, as well as producing much larger relative standard deviation values. (1) is almost always fast than (3), while becoming slower than (2) with increasing individual numbers. In our implementation, all algorithms produce faster than real-time speeds with 256 or fewer individuals (see also appendix Table *Appendix 4—table 1*), with (1) and (2) even getting close for 512 individuals.

| Video metrics | | | | % real-time | | |
|---|---|---|---|---|---|---|
| Video | # ind. | Fps (Hz) | Size (px$^2$) | Tree | Approximate | Hungarian |
| 0 | 1024 | 40 | 3866 × 4048 | 35.49 ± 65.94 | 38.69 ± 65.39 | 12.05 ± 18.72 |
| 1 | 512 | 50 | 3866 × 4140 | 51.18 ± 180.08 | 75.02 ± 193.0 | 28.92 ± 29.12 |
| 2 | 512 | 60 | 3866 × 4048 | 59.66 ± 121.4 | 65.58 ± 175.51 | 23.18 ± 26.83 |
| 3 | 256 | 50 | 3866 × 4140 | 174.02 ± 793.12 | 190.62 ± 743.54 | 127.86 ± 9841.21 |
| 4 | 256 | 60 | 3866 × 4048 | 140.73 ± 988.15 | 155.9 ± 760.05 | 108.48 ± 2501.06 |
| 5 | 128 | 60 | 3866 × 4048 | 318.6 ± 347.8 | 353.58 ± 291.63 | 312.05 ± 337.71 |
| 6 | 128 | 60 | 3866 × 4048 | 286.13 ± 330.08 | 314.91 ± 303.53 | 232.33 ± 395.21 |
| 7 | 100 | 32 | 3584 × 3500 | 572.46 ± 98.21 | 611.5 ± 96.46 | 637.87 ± 97.03 |
| 8 | 59 | 51 | 2306 × 2306 | 744.98 ± 364.43 | 839.45 ± 257.56 | 864.01 ± 223.47 |
| 9 | 15 | 25 | 1880 × 1881 | 4626 ± 424.8 | 4585.08 ± 378.64 | 4508.08 ± 404.56 |
| 10 | 10 | 100 | 1920 × 1080 | 2370.35 ± 303.94 | 2408.27 ± 297.83 | 2362.42 ± 296.99 |
| 11 | 10 | 32 | 3712 × 3712 | 6489.12 ± 322.59 | 6571.28 ± 306.34 | 6472.0 ± 322.03 |
| 12 | 10 | 32 | 3712 × 3712 | 6011.59 ± 318.12 | 6106.12 ± 305.96 | 55.49.25 ± 318.21 |
| 13 | 10 | 32 | 3712 × 3712 | 6717.12 ± 325.37 | 6980.12 ± 316.59 | 6726.46 ± 316.87 |
| 14 | 8 | 30 | 3008 × 3008 | 8752.2 ± 2141.03 | 8814.63 ± 2140.4 | 8630.73 ± 2177.16 |
| 15 | 8 | 25 | 3008 × 3008 | 9786.68 ± 1438.08 | 10118.04 ± 1380.2 | 9593.44 ± 1439.28 |
| 16 | 8 | 35 | 3008 × 3008 | 6861.42 ± 1424.91 | 10268.82 ± 1339.8 | 9680.68 ± 1387.14 |
| 17 | 1 | 140 | 1312 × 1312 | 15323.05 ± 637.17 | 15250.39 ± 639.2 | 15680.93 ± 640.99 |

**Appendix 4—table 4.** Comparing the time-cost for tracking and converting videos in two steps with doing both of those tasks at the same time.

The columns *prepare* and *tracking* show timings for the tasks when executed separately, while *live* shows the time when both of them are performed at the same time using the live-tracking feature of TGrabs. The column *win* shows the time 'won' by combining tracking and preprocessing as the percentage (prepare + tracking − live)/(prepare + tracking). The process is more complicated than simply adding up timings of the tasks. Memory and the interplay of work-loads have a huge effect here. Posture is enabled in all variants.

| Video metrics | | | | Minutes | | | |
|---|---|---|---|---|---|---|---|
| Video | # ind. | Length | Fps (Hz) | Prepare | Tracking | Live | Win (%) |
| 0 | 1024 | 8.33min | 40 | 10.96 ± 0.3 | 41.11 ± 0.34 | 65.72 ± 1.35 | -26.23 |
| 1 | 512 | 6.67min | 50 | 11.09 ± 0.24 | 24.43 ± 0.2 | 33.67 ± 0.58 | 5.24 |
| 2 | 512 | 5.98min | 60 | 11.72 ± 0.2 | 20.86 ± 0.47 | 31.1 ± 0.62 | 4.55 |
| 3 | 256 | 6.67min | 50 | 11.09 ± 0.21 | 7.99 ± 0.17 | 12.32 ± 0.17 | 35.26 |
| 4 | 256 | 5.98min | 60 | 11.76 ± 0.26 | 9.04 ± 0.26 | 15.08 ± 0.13 | 27.46 |
| 6 | 128 | 5.98min | 60 | 11.77 ± 0.29 | 4.74 ± 0.13 | 12.13 ± 0.32 | 26.49 |
| 5 | 128 | 6.0min | 60 | 11.74 ± 0.26 | 4.54 ± 0.1 | 12.08 ± 0.25 | 25.79 |
| 7 | 100 | 1.0min | 32 | 1.92 ± 0.02 | 0.47 ± 0.01 | 2.03 ± 0.02 | 14.88 |
| 8 | 59 | 10.0min | 51 | 6.11 ± 0.07 | 7.68 ± 0.12 | 9.28 ± 0.08 | 32.7 |
| 9 | 15 | 60.0min | 25 | 12.59 ± 0.18 | 5.32 ± 0.07 | 13.17 ± 0.12 | 26.47 |
| 11 | 10 | 10.17min | 32 | 8.58 ± 0.04 | 0.74 ± 0.01 | 8.8 ± 0.12 | 5.66 |
| 12 | 10 | 10.05min | 32 | 8.68 ± 0.04 | 0.75 ± 0.01 | 8.65 ± 0.07 | 8.3 |
| 13 | 10 | 10.05min | 32 | 8.67 ± 0.03 | 0.71 ± 0.01 | 8.65 ± 0.07 | 7.76 |
| 102 | 10 | 10.08min | 100 | 4.17 ± 0.06 | 2.02 ± 0.02 | 4.43 ± 0.05 | 28.3 |
| 14 | 8 | 195.37min | 30 | 110.51 ± 2.32 | 8.99 ± 0.22 | 109.97 ± 2.05 | 7.98 |
| 15 | 8 | 72.0min | 25 | 31.84 ± 0.53 | 3.26 ± 0.07 | 32.1 ± 0.42 | 8.55 |
| 16 | 8 | 198.22min | 35 | 133.45 ± 2.22 | 11.38 ± 0.28 | 1.33 ± 2.28 | 8.1 |
| mean | | | | | | | 14.55 % |

**Appendix 4—table 5.** Statistics for running the tree-based matching algorithm with the videos of the speed dataset.

We achieve low leaf and node visits across the board – this is especially interesting in videos with high numbers of individuals. High values for '# nodes visited' are only impactful if they make up a large portion of the assignments. These are the result of too many choices for assignments – the weak point of the tree-based algorithm – and lead to combinatorial 'explosions' (the method will take a really long time to finish). If such an event is detected, TRex automatically switches to a more computationally bounded algorithm like the Hungarian method.

| Video characteristics | | Matching stats | | |
|---|---|---|---|---|
| Video | # ind. | # nodes visited (5,50,95,100%) | # leafs visited | # improvements |
| 0 | 1024 | [1535; 2858; 83243; 18576918] | 1.113 ± 0.37 | 1.113 |
| 1 | 512 | [1060; 8156; 999137; 19811558] | 1.247 ± 0.61 | 1.247 |
| 2 | 512 | [989; 2209; 56061; 8692547] | 1.159 ± 0.47 | 1.159 |
| 3 | 256 | [452; 479; 969; 205761] | 1.064 ± 0.29 | 1.064 |
| 4 | 256 | [475; 496; 584; 608994] | 1.028 ± 0.18 | 1.028 |
| 5 | 128 | [233; 245; 258; 7149] | 1.012 ± 0.12 | 1.012 |

*Continued on next page*

*Appendix 4—table 5 continued*

| Video characteristics | | Matching stats | | |
|---|---|---|---|---|
| Video | # ind. | # nodes visited (5,50,95,100%) | # leafs visited | # improvements |
| 6 | 128 | [237; 259; 510; 681702] | 1046 ± 0.25 | 1.046 |
| 7 | 100 | [195; 199; 199; 13585] | 1.014 ± 0.14 | 1.014 |
| 8 | 59 | [117; 117; 117; 16430] | 1.014 ± 0.2 | 1.014 |
| 9 | 15 | [24; 29; 29; 635] | 1.027 ± 0.22 | 1.027 |
| 10 | 10 | [17; 19; 19; 56] | 1.001 ± 0.02 | 1.001 |
| 11 | 10 | [19; 19; 19; 129] | 1.006 ± 0.1 | 1.006 |
| 12 | 10 | [19; 19; 19; 1060] | 1.023 ± 0.23 | 1.023 |
| 13 | 10 | [19; 19; 19; 106] | 1.001 ± 0.04 | 1.001 |
| 14 | 8 | [11; 15; 15; 893] | 1.003 ± 0.08 | 1.003 |
| 15 | 8 | [13; 15; 15; 597] | 1.024 ± 0.23 | 1.024 |
| 16 | 8 | [15; 15; 15; 2151] | 1.009 ± 0.17 | 1.009 |
| 17 | 1 | [1; 1; 1; 1] | 1.0 ± 0.02 | 1.0 |

## Appendix 5

### Posture

Estimating an animals orientation and body pose in space is a diverse topic, where angle and pose can mean many different things. We are not estimating the individual positions of many legs and antennae in TRex, we simply want to know where the front- and the back-end of the animal are. Ultimately, the goal here is to be able to align animals using an arbitrary axis with their head extending in one direction and their tail roughly in the opposite direction. In order to achieve this, we are required to follow a series of steps to acquire all the necessary information:

1. Locate objects in the image
2. Detect the edge of objects
3. Find an ordered set of points (the outline), which in sequence approximate the outer edge of an object in the scene. This is done for each object (as well as for holes).
4. Calculate a center-line based on local curvature of the outline.
5. Calculate head and tail positions.

The first point is a given at this point (see Appendix C Connected components algorithm). We can utilize the format in which connected components are computed in TRex (an ordered array of horizontal line segments), which reduces redundancy by avoiding to look at every individual pixel. These line segments also contain information about edges since every start and end has to be an edge-pixel, too.

Even though we already have a list of edge-pixels, retrieving an *ordered* set of points is crucial and requires much more effort. Without information about a pixels connectivity, we cannot differentiate between inner and outer shapes (holes vs. outlines) and we cannot calculate local curvature.

### Connecting pixels to form an outline

We implemented an algorithm based on horizontal line segments, which only ever retains three consecutive rows of pixels (*p* previous, *c* current, and *n* next). These horizontal line segments always stem from a 'blob' (or connected component). Rows contain (i) their y-value in pixels, (ii) $x_0, x_1$ values describing the first and last 'on'-pixel that has been found in it, (iii) a set of detected border pixels (identified by their x-coordinate). A row is valid, whenever the *y* coordinate is not -1 – all three rows are initialized to an invalid $y = -1$ is the previous row. Using $p, c$ or $n$ as a function $c(x)$ returns one for on-pixels at that x-coordinate, and 0 for off-pixels.

For each line, *l* in the sorted list of horizontal line segments, we detect border pixels:

1. Subtract the blobs position (minimum of all $l_{x0}$ and $l_y$ separately) from *l*
2. If $n_y \neq l_y$, a row has ended and a new one starts: call finalize
   else if $l_{x0} - l'_{x1} \geq 1 \wedge l_{x0} \geq c_{x0}$, we either skipped a few pixels in *n* or *l* starts before *c* even had valid pixels. This means that all pixels *x* between $\max\{l'_{x1} + 1; c_{x0}\} \leq x < \min\{l_{x0}; c_{x1} + 1\}$ are border pixels in *c*.
3. If $l_{x1} < c_{x0}$, or *c* is invalid, then line *l* ends before the previous row (*c*) even has any 'on'-pixels. All pixels *x* between $l_{x0} \leq x \leq l_{x1}$ are border pixels in *n*.
   else
   (a) $s l_{x0}$
   (b) if $s < c_{x0}$, then lines are overlapping in *c* and *n* (line *l*). We can fill *n* up with border while $x < c_{x0}$ and $x \leq l_{x1}$. Set $s := \min\{c_{x0} - 1; l_{x1}\}$. else if $s = 0$ or $s > 0 \wedge n(s - 1) = 0$, then *l* starts at the image border (which is an automatic border pixel) or there is a gap before *l*. Set $s := s + 1$.
   (c) All pixels at x-coordinates $s \leq x \leq l_{x1}$ are border in *n*, if they are either (i) beyond *c*'s bounds ($x \geq c_{x1}$), or (ii) $c(x) = 0$.
4. Set $n_{x1} := l_{x1}$.

After iterating through all lines, we need two additional calls to finalize to populate the lines currently in *c* and *n* through.

A graph is updated each time a row is finalized. This graph stores all border 'nodes', as well as all a maximum of two edges per node (since this is the maximum number of neighbors for a line vertex). More on that below. The following procedure (finalize) prepares a row (*c*) to be integrated into the

graph, using two parameters: A triplet of rows $(p, c, n)$ and the first line $l$, which started the new row to be added.

1. If $n$ is invalid, continue to the next operation.
   else if $l_y > n_y + 1$, then we skipped at least one row between $n$ and the new row – making all on-pixels in $n$ border pixels.
   else we have consecutive rows where $l_y = n_y + 1$. All on-pixels $x$ in $n$ between $n_{x0} \leq x \leq l_{x0} - 1$ are border pixels.
2. Now the current row ($c$) is certainly finished, as it will in the following become the previous row ($p$), which is read-only at that point. We can add every border-pixel of $c$ to our graph (see below).
3. It then discards $p$ and moves $c \to p$ and $n \to c$, as well as reading a new row to assign to $n$, setting $n_{x0} = l_{x0}, n_{x1} = l_{x1}, n_y = l_y$.

The graph consists of nodes (border pixels), indexed by their x and y coordinates (integers) and containing a list of all on-pixels around them (8-neighborhood with top-left, top, left, bottom-left, etc.). This information is available when finalize is called, since the middle row ($c$) is fully defined at that point (its entire neighborhood has been cached).

After all rows have been processed, an additional step is needed to connect all nodes and produce a connected, clockwise ordered outline. We already marked all pixels that have at least one border. We can also already mark TOP, RIGHT, BOTTOM, and LEFT borders per node if no neighboring pixel is present in that direction, since these major directions will definitely get a 'line' in the end. So all we have left to do now, is check the diagonals. The points that will be returned, are located half-way along the outer edges of pixels. In the end, each pixel can potentially have four border lines (if it is a singular pixel without connections to other pixels, see yellow 'hole' in **Appendix 5—figure 1b**). The half-edge-points for each node are generated as follows:

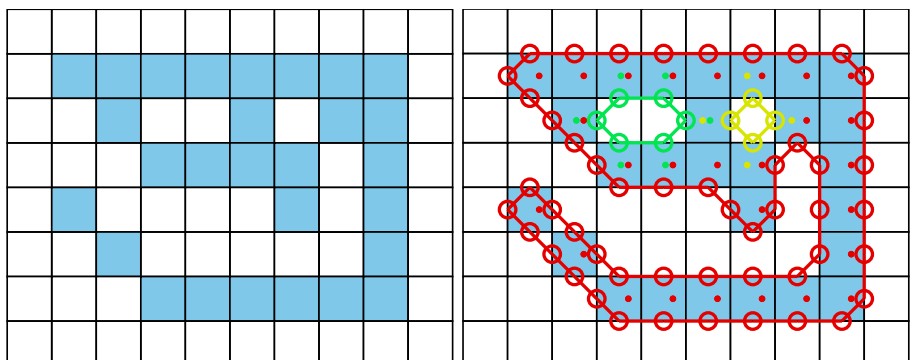

**Appendix 5—figure 1.** The original image is displayed on the left. Each square represents one pixel. The processed image on the right is overlaid with lines of different colors, each representing one connected component detected by our outline estimation algorithm. Dots in the centers of pixels are per-pixel-identities returned by OpenCVs findContours function (for reference) coded in the same colors as ours. Contours calculated by OpenCVs algorithm can not be used to estimate the one-pixel-wide 'tail' of the 9-like shape seen here, since it becomes a 1D line without sub-pixel accuracy. Our algorithm also detects diagonal lines of pixels, which would otherwise be an aliased line when scaled up.

1. A nodes list of border pixels is a sparse, ordered list of directions (top, top-right, . . ., top-left). Each major direction of these (TOP, RIGHT, BOTTOM, LEFT), if present, check the face of their square to the left of them (own direction - 1, or $- 45°$). For example, TOP would check top-left.
2. if the checked neighbor is on, we add an edge between our face (e.g. TOP) and its 90° rotated face (e.g. own direction + 2 = RIGHT).
   else check the face an additional 45° to the left (e.g. LEFT).
   (a) if it there is an on-pixel attached to this face, add an edge between the two faces (of the focal and its left pixel) in the same direction (e.g. TOP $\to$ TOP).

(b) else we do not seem to have a neighbor to either side, so this must be a corner pixel. Add an edge from the focal face (e.g. TOP) to the side 90° to the left of itself (e.g. LEFT).

Each time an edge is added, more and more of the half-edges are becoming fully connected (meaning they have two of the allowed two edges). To generate the final result, all we have to do is to start somewhere in the graph and walk strictly in clockwise direction. 'Walking' is done using a queue and edges are followed using depth-first search (see Appendix D Matching an object to an object in the next frame): Each time a node is visited, all its yet unexplored edges are added to the front of the queue (in clockwise order). Already visited edges are marked (or pruned) and will not be traversed again – their floating-point positions (somewhere on an edge of its parent pixel) are added to an array.

After a path ended, meaning that no more edges can be reached from our current node, the collected floating-point positions are pushed to another array and a different, yet unvisited, starting node is sought. This way, we can accumulate all available outlines in a given image one-by-one – including holes.

These outlines will usually be further processed using an Elliptical Fourier Transform (or EFT, *Kuhl and Giardina, 1982*), as mentioned in the main-text. Outlines can also be smoothed using a weighted average of the $N$ points around a given point, or resampled to either reduce or (virtually) increase resolution.

## Finding the tail

Given an ordered outline, curvature can be calculated locally (per index $i$):

$$C(i) = 4 * \text{triangle\_area}(p_{i-r}, p_i, p_{i+r})/(p_i - p_{i-r} * p_i - p_{i+r} * p_{i-r} - p_{i+r})$$

where $1 \leq r \in \mathbb{N}$ is a parameter, which effectively leads to more smoothing when increased. Triangle area can be calculated as follows:

$$\text{triangle\_area}(\mathbf{a}, \mathbf{b}, \mathbf{c}) = (\mathbf{b}_x - \mathbf{a}_x)(\mathbf{c}_y - \mathbf{a}_y) - (\mathbf{b}_y - \mathbf{a}_y)(\mathbf{c}_x - \mathbf{a}_x).$$

To find the 'tail', or the pointy end of the shape, we employ a method closely related to scipys find_peaks function: We find local maxima using discrete curve differentiation and then generate a hierarchy of these extrema. The only major difference to normal differentiation is that we assume periodicity to achieve our results – values wrap around in both directions, since we are dealing with an outline here. We then find the peak with the largest integral, meaning we detect both very wide and very high peaks (just not very slim ones). The center of this peak is the 'tail'.

To find the head as well, we now have to search for the peak that has the largest (index-) distance to the tail-peak. This is a periodic distance, too, meaning that $N$ is one of the closest neighbors of 0.

The entire outline array is then rotated, so that the head is always the first point in it. Both indexes are saved.

## Calculating the center-line

A center-line, for a given outline, can be calculated by starting out at the head and walking in both directions from there – always trying to find a pair of points with minimal distance to each other on both sides. Two indices are used: $l, r$ for left and right. We also allow some 'wiggle-room' for the algorithm to find the best-matching points on each side. This is limited by a maximum offset of $\omega$ points which is set to $0.025 * N$ by default, where $N$ is the number of points in the outline. $\mathbf{f}(i)$ gives the point on in outline at position $i$.

Starting from $l := -1, r := 1$ we continue while $r < l + N$:

1. Find $m := \text{argmin}_i\{\mathbf{f}(r + i) - \mathbf{f}(l); \forall i \leq \omega \wedge r + i < N\}$. If no valid $m$ can be found, abort. Otherwise set $rm$.
2. Find $k := \text{argmin}_i\{\mathbf{f}(l - i + N) - \mathbf{f}(r); \forall i \leq \omega \wedge l - i \leq -N\}$. If no valid $k$ can be found, abort. Otherwise set $lk$.
3. Our segment now consists of points $\mathbf{f}(m)$ and $\mathbf{f}(k)$, with a center vector of $(\mathbf{f}(k) - \mathbf{f}(m)) * 0.5 + \mathbf{f}(m)$. Push it to the center-line array. We can also calculate the width of the body at that point using $\mathbf{f}(k) - \mathbf{f}(m)$.
4. Set $l := l - 1$.

5. Set $r := r + 1$.

Head and tail positions can be switched now, for example for animals where the wider part is the head. We may also want to start at the slimmest peak first, which ever that is, since there we have not as much space for floating-point errors regarding where *exactly* the peak was. These options depend on the specific settings used in each video.

The angle of the center-line is calculated using $\mathrm{atan2}$ for a vector between the first point and one point at an offset from it. The specific offset is determined by a midline stiffness parameter, which offers some additional stability – despite for example potentially noisy peak detection.

## Appendix 6

### Visual field estimation

Visual fields are calculated by casting rays and intersecting them with other individuals and the focal individual (for self-occlusion). An example of this can be seen in *Figure 6*. The following procedure requires posture for all individuals in a frame. In case an individual does not have a valid posture in the given frame, its most recent posture and position are used as an approximation. The field is internally represented as a discretized vector of multi-dimensional pixel values. Depending on the resolution parameter ($F_{\text{res}}$), which sets the number of pixels, each index in the array represents step-sizes of $(F_{\text{max}} - F_{\text{min}})/F_{\text{res}}$ radians. The $F$ values are constants setting the minimum and maximum field of view ($-130°$ to $130°$ by default, which gives a range of $260°$). Each pixel consists of multiple data-streams: The distance to the other individual, the identity of the other individual and the body-part that the ray intersected with.

Eyes are simulated to be located on the outline of the focal individual, near the head. The distance to the head can be set by the user as a percentage of midline-length. To find the exact eye position, the program calculates intersections between vectors going left/right from that midline point, perpendicular to the midline, and the individual's outline. In order to be able to simulate different types of binocular and monocular sight, a parameter for eye separation $E_{\text{sep}}$ (radians) controls the offset from the head angle $H_\alpha$ per eye. Left and right eye are looking in directions $H_\alpha - E_{\text{sep}}$ and $H_\alpha + E_{\text{sep}}$, respectively.

We iterate through all available postures in a given frame and use a procedure which is very similar to depth-maps (*Williams, 1978*) in for example OpenGL. In the case of 2D visual fields, this depth-map is 1D. Each pixel holds a floating-point value (initialized to $\infty$) which is continuously compared to new samples for the same position – if the new sample is closer to the 'camera' than the reference value, the reference value is replaced. This way, after all samples have been evaluated, we generate a map of the objects closest to the 'camera' (in this case the eye of the focal individual). For that to work we also have to keep the identity in each of these discrete slots maintained. So each time a depth value is replaced, the same goes for all the other data-streams (such as identity and head-position). When an existing value is replaced, values in deeper layers of occlusion are pushed downwards alongside the old value for the first layer.

Position of the intersecting object's top-left corner is located at $\hat{P}$. Let $E_e$ be the position of each eye, relative to $\hat{P}$. For each point $P_j$ (coordinates relative to $\hat{P}$) of the outline, check the distance between $E_e$ and the outline segments $(P_j - P_{j-1})$. For each eye $E_e$:

1. Project angles ranging from $\left[\text{atan2}(P_{j-1} + E_e), \text{atan2}(P_j + E_e)\right]$, where $\alpha_e$ is the eye orientation, using:

$$\Gamma_e(\beta) = \text{angle\_normalize}(\beta - \alpha_e - F_{\text{min}})/(F_{\text{max}} - F_{\text{min}}) * F_{\text{res}}$$

$\text{angle\_normalize}(\beta)$ normalizes beta to be between $[-\pi, \pi]$.

2. If $\max(R)$ or $\min(R)$ is inside the visual field ($0 \leq \Gamma_e(\beta) \leq 1$):

(a) We call the first angle satisfying the condition β.

(b) Then the search range becomes $R := [\lfloor \max\{\beta - 0.5; 0\}\rfloor, \lfloor \beta + 0.5\rfloor]$, where the elements in $R$ are integers.

(c) Let $\delta_{j,e} = P_{j-1} - E_e$, the distance between outline point at $j - 1$ and the eye (interpolation could be done here).

(d) Let index $k \in \mathbb{N}, k \in R$ be our index into the first layer of the depth-map *depth* $_0$:

(e) if $\text{depth}_0(k) > \delta_{j,e}$: Calculate all properties $D_0(k) := \{\text{head\_distance}, \ldots \in \text{data\_streams}\}^T$, and push values at $k$ in layer 0 to layer 1.

(f) Otherwise, if $\text{depth}_1(k) > \delta_{j,e}$, calculate properties for layer one instead and move data from layer one further down, etc.

The data-streams are calculated individually with the following equations:

- Distance: Given already in $\text{depth}_i(k)$. In practice, values are cut off at the maximum distance (size of the video squared) and normalized to $[0, 255]$.
- Identity: Is assigned alongside $\text{depth}_i(k)$ for each element that successfully replacing another in the map.

- Body-part: Let $T_i$ = tail index, $L_{l/r}$ = number of points in left/right side of the outline (given by tail- and head-indexes):

  (a) if $i > T_i$: $\text{head\_distance} = 1 - |i - T_i|/L_l$

  (b) else: $\text{head\_distance} = 1 - |i - T_i|/L_r$.

## Appendix 7

### The PV file format

Since we are using a custom file format to save videos recorded using TGrabs (MP4 video can be saved alongside PV for a limited time or frame-rate), the following is a short overview of PV6 contents and structure. This description is purely technical and concise. It is mainly intended for users who wish to implement a loader for the file format (e.g. in Python) or are curious.

### Structure

Generally, the file is built as a header (containing meta information on the video) followed by a long data section and an index table plus a settings string at the end. The header at the start of the file can be read as follows:

1. version (string): 'PV6'
2. channels (uint8): Hard-coded to 1
3. width and height (uint16): Video size
4. crop offsets (4x uint16): Offsets from original image
5. size of HorizontalLine struct (uchar)
6. # frames (uint32)
7. index offset (uint64): Byte offset pointing to the index table for
8. timestamp (uint64): time since 1970 in microseconds of recording (or conversion time if unavailable)
9. empty string
10. background image (byte*): An array of uint8 values of size width * height * channels.
11. mask image size (uint64): 0 if no mask image was used, otherwise size in bytes followed by a byte* array of that size.

Followed by the data section, where information is saved per frame. This information can either be in a zip-compressed format, or raw (determined by size), see below:

1. compression flag (uint8): one if compression was used, 0 otherwise
2. if compressed:
   (a) original size (uint32)
   (b) compressed size (uint32)
   (c) lzo1x compressed data (byte*) in the format of the uncompressed variant (below)
3. if uncompressed:
   (a) timestamp since start time in header (uint32)
   (b) number of images in frame (uint16)
   (c) for each image in frame:
   i.    number of HorizontalLines (uint16)
   ii.   data of HorizontalLine (byte*)
   iii.  pixel data for each pixel in the previous array (byte*).

Files are concluded by the index table, which gives a byte offset for each video frame in the file, and a settings string. This index is used for quick frame skipping in TRex as well as random access. It consists of exactly one uint64 index per video frame (as determined by the number of video frames read earlier). After that map ends, a string follows, which contains a JSON style string of all metadata associated by the user (or program) with the video (such as species or size of the tank).

## Appendix 8

### Automatic visual identification

Network layout and training procedure

Network layout is sketched in *1* c. Using version 2.2.4 of Keras See keras.io documentation for default arguments, weights of densely connected layers as well as convolutional layers are initialized using Xavier-initialization (*Glorot and Bengio, 2010*). Biases are used and initialized to 0. The default image size in TRex is $80 \times 80$, but can be changed to any size in order to retain more detail or improve computation speed.

During training, we use the Adam optimizer (*Bengio et al., 2015*) to traverse the loss landscape, which is generated by categorical focal loss. *Categorical* focal loss is an adaptation of the original *binary* focal loss (*Lin et al., 2020*) for multiple classes:

$$\mathrm{cFL}(j) = \sum_{c=1}^{N} -\alpha\left(1 - \mathbf{P}_{jc}\right)^{\gamma} \mathbf{V}_{jc} \log\left(\mathbf{P}_{jc}\right),$$

where $\mathbf{P}_{jc}$ is the prediction vector component returned by the network for class $c$ in image $j$. $\mathbf{V}$ is a set of validation images, which remains the same throughout the training process. It comprises 25 % of the images available per individual. Images are marked *globally* when becoming part of the validation dataset and are not used for training in the current or any of the following steps.

After each epoch, predictions are generated by performing a forward-pass through the network layers. Returned are the softmax-activations $\mathbf{P}_{jc}$ of the last layer for each image $j$ in the validation dataset. Simply calculating the mean of

$$\overline{A} = \frac{1}{M} \sum_{j \in [0,M]} \begin{cases} 1 & \text{if } \mathbf{P}j = \mathbf{V}j, \\ 0 & \text{otherwise} \end{cases}$$

gives the mean accuracy of the network. $M$ is the number of images in the validation dataset, where $\mathbf{V}_j$ are the expected probability vectors per image $j$. However, much more informative is the per-class (per-identity) accuracy of the network among the set of images $i$ belonging to class $c$, which is

$$I_c = \left\{ j; \text{where } \mathbf{V}_{jc} = 1, j \in [0,M] \right\},$$

given that all vectors in $V$ are one-hot vectors – meaning the vector has length $N$ with $\mathbf{V}_{j\phi} = 0 \, \forall \phi \, /= c$ and $\mathbf{V}_{jc} = 1$.

$$A_c = \frac{1}{|I_c|} \sum_{j \in I_c} \begin{cases} 1 & \text{if } \mathbf{P}j = \mathbf{V}j \\ 0 & \text{otherwise} \end{cases}$$

Another constant, across training units – not just across epochs, is the set of images used to calculate mean uniqueness $\bar{U}$ (see Box 1, as well as Guiding the Training Process). Values generated in each epoch $t$ of every training unit are kept in memory and used to calculate their derivative $\bar{U}'(t)$.

### Stopping-criteria

A training unit can be interrupted if one of the following conditions becomes true:
 1. Training commenced for at least $t = 5$ epochs, but uniqueness value $\bar{U}$ was never above

$$\bar{U}_{\mathrm{best}}^2 > \bar{U}(t) \, \forall t$$

where $\bar{U}_{\mathrm{best}}$ is the best mean uniqueness currently achieved by any training unit (initialized with zero). This prevents to train on faulty segments after a first successful epoch.
 2. The worst accuracy value per class has been 'good enough' in the last three epochs:

$$\min_{c\in[0,N]}\{A_c\} \geq 0.97$$

3. The global uniqueness value has been plateauing for more than 10 epochs.

$$\sum_{k\in[t-10,t]} \bar{U}'(k) \leq 0.01$$

4. Overfitting: Change in loss is very low on average after more than five epochs. Mean loss is calculated as follows:

$$\mathrm{cFL}_\mu(t) = \frac{1}{5}\sum_{k\in[t-6,t-1]} \mathrm{cFL}(k)$$

Now if the difference between the current loss and the previous loss is below a threshold:

$$\lambda(t) = \lfloor \ln(\mathrm{cFL}(t))\rfloor - 1$$

$$\frac{1}{5}\sum_{k\in[t-5,t]} \max\{\epsilon; |\mathrm{cFL}(k) - \mathrm{cFL}_\mu(k)|\} < 0.05 * 10^{\lambda(t)}$$

5. Maximum number of epochs has been reached. User-defined option limiting the amount of time that training can take per unit. By default this limit is set to 150 epochs.

6. Loss is zero. No further improvements are possible within the current training unit, so we terminate and continue with the next.

A high per-class accuracy over multiple consecutive epochs is usually an indication that everything that can be learned from the given data has already been learned. No further improvements should be expected from this point, unless the training data is extended by adding samples from a different part of the video. The same applies to scenarios with consistently zero or very low change in loss. Even if improvements are still possible, they are more likely to happen during the final (overfitting) step where all of the data is combined.

## Appendix 9

### Data used in this paper and reproducibility

All of the data, as well as the figures and tables showing the data, have been generated automatically. We provide the scripts that have been used, as well as the videos if requested. 'Data' refers to converted video-files, as well as log- and NPZ-files. Analysis has been done in Python notebooks, using mostly matplotlib and pandas, as well as numpy to load the data. Since TRex and TGrabs, as well as `idtracker.ai` have been run on a Linux system, we were able to run everything from two separate bash files:

1. run.bash
2. run_idtracker.bash

where (1) encompasses all trials run using TRex and TGrabs, both for the speed- and recognition-datasets. (2) runs `idtracker.ai` in its own dedicated Python environment, using only the recognition-dataset. The parameters we picked for `idtracker.ai` vary between videos and are hand-crafted, saved in individual .json files (see Table *Appendix 9—table 1* for a list of settings used). We ran multiple trials for each combination of tools and data with $N = 5$ where necessary:

- 3x TGrabs [speed-dataset]
- 5x TRex + recognition [recognition-dataset]
- 3x `idtracker.ai` [recognition-dataset]
- TRex without recognition enabled [speed-dataset]:
    - 3x for testing the tree-based, approximate and Hungarian methods (4.2 Tracking), without posture enabled – testing raw speeds (see Table *Appendix 4—table 3*)
    - 3x testing accuracy of basic tracking (see Table *Appendix 4—table 2*), with posture enabled.

A Python script used for *Figure 5*, which is run only once. It generates a series of results for the same video (Video 7 with 100 individuals) with different sample-sizes. It uses a single set of training samples and then – after equalizing the numbers of images per individual – generates multiple virtual subsets with fewer images. They span 15 different sample-sizes per individual, saving a history of accuracies for each run. We repeated the same procedure with for the different normalization methods (no normalization, moments and posture), each repeated five times.

As described in the main text, we recorded memory usage with an external tool (syrupy) and used it to measure both software solutions. This tool saves a log-file for each run, which is appropriately renamed and stored alongside the other files of that trial.

All runs of TRex are preceded by running a series of TGrabs commands first, in order to convert the videos in the datasets. We chose to keep these trials separately and load whenever possible, to avoid data-duplication. Since subsequent results of TGrabs are always identical (with the exception of timings), we only keep one version of the PV files (Appendix G The PV file format) as well as only one version of the results files generated using live-tracking. However, multiple runs of TGrabs were recorded in the form of log-files to get a measure of variance between runs in terms of speed and memory.

### Human validation

To ensure that results from the automatic evaluation (in Visual identification: accuracy) are plausible, we manually reviewed part of the data. Specifically, the table in *Table 3* shows an overview of the individual events reviewed and percentages of wrongly assigned frames. Due to the length of videos and the numbers of individuals inside the videos, we did not review all videos in their entirety, as shown in the table. Using the reviewing tools integrated in TRex, we focused on crossings that were automatically detected. These tools allow the user to jump directly to points in the video that it deems problematic. Detecting problematic situations is equivalent to detecting the end of individual segments (see Automatic visual identification based on machine learning). While iterating through these situations, we corrected individuals that have been assigned to the wrong object, generating a clean and corrected baseline dataset. We assumed that an assignment is correct, as long as the individual is at least part of the object that the identity has been assigned to. Misassignments were

typically fixed after a few frames. Identities always returned to the correct individuals afterward (thus not causing a chain of follow-up errors).

## Comparison between trajectories from different softwares, or multiple runs of the same software

In our tests, the same individuals may have been given different IDs (or 'names') by each software (and in each run of each software for the same video), so, as a first step in every test where this was relevant, we had to determine the optimal pairing between identities of two datasets we wished to compare. This was done using a square distance matrix containing overall euclidean distances between identities is calculated by summing their per-frame distances. Optimally, this number would be zero for one and greater than zero for every other pairing, but temporary tracking mistakes and differences in the calculation of centroids may introduce noise. Thus, we solved the matching problem (see Appendix D Matching an object to an object in the next frame) for identities between each two datasets and paired individuals with the smallest accumulative distance between them. This was done for all results presented, where a direct comparison between two datasets was required.

**Appendix 9—table 1.** Settings used for `idtracker.ai` trials, as saved inside the json files used for tracking.

The minimum intensity was always set to 0 and background subtraction was always enabled. An ROI is an area of interest in the form of an array of 2D vectors, typically a convex polygon containing the area of the tank (e.g. for fish or locusts). Since this format is quite lengthy, we only indicate here whether we limited the area of interest or not.

| Video | length (# frames) | Nblobs | Area | Max. intensity | Roi |
|---|---|---|---|---|---|
| 7 | 1921 | 100 | [165, 1500] | 170 | Yes |
| 8 | 30,626 | 59 | [100, 2500] | 160 | Yes |
| 11 | 19,539 | 10 | [200, 1500] | 10 | Yes |
| 13 | 19,317 | 10 | [200, 1500] | 10 | Yes |
| 12 | 19,309 | 10 | [200, 1500] | 10 | Yes |
| 9 | 90,001 | 8 | [190, 4000] | 147 | Yes |
| 16 | 416,259 | 8 | [200, 2500] | 50 | No |
| 14 | 351,677 | 8 | [200, 2500] | 50 | No |
| 15 | 108,000 | 8 | [250, 2500] | 10 | No |

## Appendix 10

### Matching probabilities

One of the most important steps, when matching objects in one frame with objects in the next frame, is to calculate a numerical landscape that can then be traversed by a maximization algorithm to find the optimal combination. This landscape, which can be expressed as an $m \times n$ matrix $\mathbf{P}(t)$, contains the probability values between $[0, 1]$ for each assignment between individuals $i$ and objects $B_j$.

Below are definitions used in the following text:

- $T_\Delta$ is the typical time between frames (s), which depends on the video
- $\tau_i < t$ is most recent frame assigned to individual $i$ previous to the current frame $t$
- $P_{\min}$ is the minimally allowed probability for the matching algorithm, underneath which the probabilities are assumed to be zero (and respective combination of object and individual is ignored). This value is set to 0.1 by default.
- $F(t \in \mathbb{R}) \to \mathbb{N}$ is the frame number associated with the time $t$ (s)
- $\mathcal{T}(f \in \mathbb{N}) \to \mathbb{R}$ is the time in seconds of frame $f$, with $F(\mathcal{T}(f)) = f$
- $\mathbf{x}$ indicates that $x$ is a vector
- $\mathbf{U}(\mathbf{x}) = \mathbf{x}/\mathbf{x}$.

Some values necessary for the following calculations are independent of the objects in the current frame and merely depend on data from previous frames. They can be re-used per frame and individual in the spirit of dynamic programming, reducing computational complexity in later steps:

$$\mathbf{v}_i(t) = \mathbf{p}'_i(t) = \frac{\delta}{\delta t}\mathbf{p}_i(t)$$

$$\hat{\mathbf{v}}_i(t) = \mathbf{v}_i(t) * \begin{cases} 1 & \text{if } \mathbf{v}i(t) \leq D\max \\ D_{\max}/\mathbf{v}_i(t) & \text{otherwise} \end{cases}$$

$$\mathbf{a}_i(t) = \frac{\delta}{\delta t}\hat{\mathbf{v}}_i(t)$$

Velocity $\mathbf{v}_i(t)$ and acceleration $\mathbf{a}_i(t)$ are simply the first and second derivatives of the individuals position at time $t$. $\hat{\mathbf{v}}_i(t)$ is almost the same as the raw velocity, but its length is limited to the maximally allowed travel distance per second ($D_{\max}$, parameter track_max_speed).

These are then further processed, combining and smoothing across values of multiple previous frames (the last five valid ones). Here, $\bar{f}(x)$ indicates that the resulting value uses data from multiple frames.

$$\bar{s}_i(t) = \underset{k \in [F(\tau)-5, F(t)]}{\mathrm{median}} \hat{\mathbf{v}}_i(\mathcal{T}(k))$$

is the speed at which the individual has travelled at recently. The mean direction of movement is expressed as

$$\bar{\mathbf{d}}_i(t) = \frac{1}{F(t) - F(\tau) + 5} \sum_{k \in [F(\tau)-5, F(t)]} \hat{\mathbf{v}}_i(\mathcal{T}(k))$$

with the corresponding direction of acceleration

$$\bar{\mathbf{a}}_i(t) = \mathbf{U}\left(\frac{1}{F(t) - F(\tau) + 5} \sum_{k \in [F(\tau)-5, F(t)]} \mathbf{a}_i(\mathcal{T}(k))\right).$$

The predicted position for individual $i$ at time $t$ is calculated as follows:

$$\dot{\mathbf{p}}_i(t) = s_i(t) \sum_{k \in [F(\tau_i), F(t)-1]} w(k)(\bar{\mathbf{d}}_i(t) + \mathcal{T}'(k) * \bar{\mathbf{a}}_i(t)),$$

with weights for each considered time-step of

$$w(f) = \frac{1+\lambda^4}{1+\lambda^4 \max\{1, f - F(\tau_i) + 1\}},$$

where $\lambda \in [0,1]$ is a decay rate (parameter track_speed_decay) at which the impact of previous positions on the predicted position decreases with distance in time. With its value approaching 1, the resulting curve becomes steeper – giving less weight previous positions the farther away they are from the focal frame.

In order to locate an individual $i$ in the current frame $F(t)$, a probability is calculated for each object $B_j$ found in the current frame resulting in the matrix:

$$\mathbf{P}(t) = \begin{bmatrix} P_0(t|B_0) & \dots & P_n(t|B_0) \\ \vdots & \ddots & \vdots \\ P_0(t|B_m) & \dots & P_n(t|B_m) \end{bmatrix}. \tag{3}$$

Probabilities $P_i(t|B_j)$ for all potential connections between blobs $B_j$ and identities $i$ at time $t$ are calculated by first predicting the expected position $\dot{\mathbf{p}}_i(t)$ for each individual in the current frame $F(t)$. This allows the program to focus on a small region of where the individual is expected to be located, instead of having to search the whole arena each time.

Based on the individual's recent speed $\bar{s}_i(t)$, direction $\bar{\mathbf{d}}_i(t)$, acceleration $\bar{\mathbf{a}}_i(t)$ and angular momentum $\bar{\alpha}'_i(t)$, the individual's projected position $\dot{\mathbf{p}}_i(t)$ is usually not far away from its last seen location for small time-steps. Only when $\Delta t$ increases, if the individual has been lost for more than one frame or frame-rates are low, does it really play a role.

The actual probability values in $\mathbf{P}(t)$ are then calculated by combining three metrics - each describing different aspects of potential concatenation of object $b$ at time $t$ to the already existing track for individual $i$:

The time metric $T_i(t)$, which does not depend on the blob the individual is trying to be matched to. It merely reflects the recency of the individuals last occurence in a way that recently seen individual will always be preferred over individuals that have been lost for longer.

$$F_{\min} = \min\left\{\frac{1}{T_\Delta}, 5\right\}$$

$$R_i(t) = \left\{\mathcal{T}(k) | F(t) - T_\Delta^{-1} \leq k \leq t \wedge \mathcal{T}(k) - \mathcal{T}(k-1) \leq T_{\max}\right\}$$

$$T_i(t) = \left(1 - \min\left\{1, \frac{\max\{0, \tau_i - t - T_\Delta\}}{T_{\max}}\right\}\right) * \begin{cases} \min\left\{1, \frac{R_i(\tau_i)-1}{F_{\min}} + P_{\min}\right\} & F(\tau i) \geq F(t0) + F\min \\ 1 & \text{otherwise} \end{cases} \tag{4}$$

$S_i(t|B_j)$ is the speed that it would take to travel from the individuals position to the blobs position in the given time (which might be longer than one frame), inverted and normalized to a value between 0 and 1.

$$S_i(t|B_j) = \left(1 + \frac{\left(\mathbf{p}_{B_j}(t) - \dot{\mathbf{p}}_i(t)\right)/(\tau_i - t)}{D_{\max}}\right)^{-2} \tag{5}$$

and the angular difference metric $A_i t, \tau_i|B_j$, describing how close in angle the resulting vector of connecting blob and individual to a track would be to the previous direction vector:

$$\mathbf{a} = \dot{\mathbf{p}}_i(t) - \mathbf{p}_i(\tau_i)$$

$$\mathbf{b} = \mathbf{p}_{B_j}(t) - \mathbf{p}_i(\tau_i)$$

$$A_i(t, \tau_i|B_j) = \begin{cases} 1 - \frac{1}{\pi}|\text{atan2}\{\mathbf{a} \times \mathbf{b}, \mathbf{a} \cdot \mathbf{b}\}| & \text{if } \mathbf{a} > 1 \wedge \mathbf{b} > 1 \\ 1 & \text{otherwise} \end{cases} \tag{6}$$

The conditional ensures that the individual travelled a long enough distance, as the $\mathrm{atan2}$ function used to determine angular difference here lacks numerical precision for very small magnitudes. This is, however, an unproblematic case in this situation as the positions are in pixel-coordinates and anything below a movement of one pixel is likely to be due to noise anyway.

Combining (4 , 5) and (6) into a weighted probability product yields:

$$P_i(t, \tau_i|B_j) = S_i(t|B_j) * \left(1 - \omega_1\left(1 + A_i(t, \tau_i|B_j)\right)\right) * (1 - \omega_2(1 + T_i(t, \tau_i))) \tag{7}$$

Results from *equation (7)* can now easily be used in a matching algorithm, in order to determine the best combination of objects and individuals as in Appendix D Matching an object to an object in the next frame. $\omega_1$ is usually set to 0.1, $\omega_2$ is set to 0.25 by default.

## Appendix 11

## Algorithm for splitting touching individuals

Algorithm 2 The algorithm used whenever two individuals touch, which is detected by a history-based method. This history-based method also provides $N_e$, the number of expected objects within the current (big) object. $T_e$ is the starting threshold constant parameter, as set by the user.

**Data**: image of a blob, $N_e$ number of expected blobs

**Result**:$N \geq N_e$ smaller image-segments, or error

**while** threshold<255**do**

 blobs = apply_threshold.image; threshold;

 **if** blobs = 0 **then**

 **break**;

 **end**

 **if** blobs $\geq N_e$ **then**

 sort blobs by size in decreasing fashion;

 loop through all blobs $i$ up to $i \leq N_e$ and detect whether the size-ratio between them is roughly even. until then, we keep iterating.;

 **if** $\min\{\text{ratio}_i \; \forall i \in [0, N_e]\}$<0.3 **then**

 threshold = threshold + 1;

 **continue** ;

 **else**

 **return** *blobs*;

 **end**

 **else**

 threshold = threshold + 1;

 **end**

**end**

**return** fail;

## Appendix 12

## Posture and visual identification of highly deformable bodies

To evaluate further whether TRex's posture and visual identification algorithms are broadly applicable, such as to mammals (e.g. rodents) – which have highly deformable bodies and thus increased variance per individual, we conducted additional analyses on videos of groups of four freely behaving mice (four C57BL/6 mice provided by D. Mink and M. Groettrup, and four 'black mice' from *Romero-Ferrero et al., 2019* provided to us by G.G. de Polavieja, and now linked under idtrackerai. readthedocs.io).

Both videos, listed in *Appendix 12—table 1* and previewed in *Appendix 12—figure 1*, were analyzed using the same scripts used to generate *Table 3*, although each video has only been automatically tracked once (since accuracy of tracking is very high, as detailed below). We manually generated verified trajectories for both videos in full, following the same procedure described in I.1 Human validation, and compared them to the automatically generated trajectories. As can be seen in *Appendix 12—table 1*, TRex provides highly accurate results for both videos ($\geq 99.6\%$).

**Appendix 12—table 1.** Analogous to our analysis in *Table 3*, we compared automatically generated trajectories for two videos with manually verified ones.
Unlike the table in the main text, the sample size per video is only one here, which is why the standard deviation is zero in both cases. Results show very high accuracy for both videos, but relatively high numbers of interpolated frames compared to *Table 3*, where only the results for Video 9 showed more than 8 % interpolation and all others remained below 1 %.

| Video | # ind. | Reviewed (%) | Interpolated (%) | TRex |
|---|---|---|---|---|
| (V1) *Romero-Ferrero et al., 2019* | 4 | 100.0 | 6.41 | 99.6 ± 0.0 |
| (V2) D. Mink, M. Groettrup | 4 | 100.0 | 1.74 | 99.82 ± 0.0 |

Tracking, in theory and in practice as per our results here, is not generally impacted by the shape of individuals. However, individuals of some species tend to stay close/on top of con-specifics, which may render them impossible to track during periods where traditional image processing methods are unable to separate them. This explains the $\sim 6\%$ interpolated frames in V1 (see *Appendix 12—table 1*), and also gives a reason why there is similarity between Video 9 and V1 in that respect – the locusts in Video 9 also spend much time either on top of others, or in places where they are harder to see.

Very short segments of mistaken identities (with a maximum length of less than 200 ms) occurred whenever individuals 'œappear' only for a short moment and the segment does not contain enough data to be properly matched with a learned identity. Correct identities were reassigned in all cases after the individuals could be visually separated from each other again, and such events only make up <1% of the tracked data.

Furthermore, we found that our method for posture estimation works well despite the more deformable bodies and complex 3D-postures of mice. Head and tail may switch occasionally, especially when animals shrink to 'œa circle' from the viewpoint of the camera. Overall, however, by far most samples are normalized correctly – as can be seen in *Appendix 12—figure 2* and *Appendix 12—figure 3*.

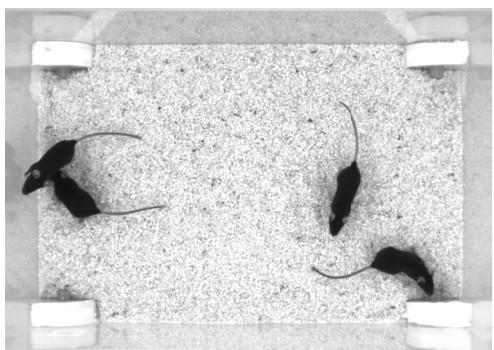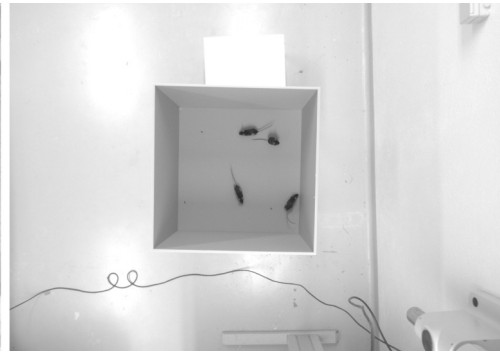

**Appendix 12—figure 1.** Screenshots from videos V1 and V2 listed in *Appendix 12—table 1*. Left (V1), video of four 'black mice' (17 min, 1272 × 909 px resolution) from *Romero-Ferrero et al., 2019*. Right (V2), four C57BL/6 mice (1: 08 min, 1280 × 960 px resolution) by M. Groettrup, D. Mink.

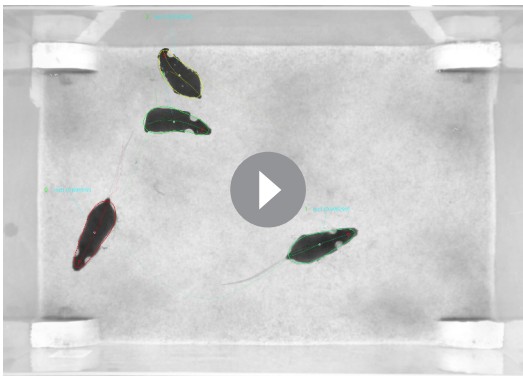

**Appendix 12—figure 1—video 1.** A clip of the tracking results from V1, played back at normal speed.     Although it succumbs to noise in some frames (e.g. around 13 s), posture estimation remains remarkably robust to it throughout the video – sometimes even through periods where individuals overlap (e.g. at 27 s). Identity assignments are near perfect here, confirming our results in *Appendix 12—table 1*. https://youtu.be/UnqRNKrYiR4.

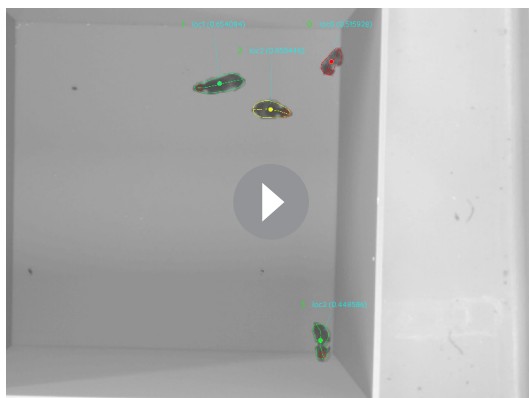

**Appendix 12—figure 1—video 2.** Tracking results from V2, played back at two times normal speed.     Since resolution per animal in V2 is lower than V1, and contrast is lower, posture estimation in V2 is also slightly worse than in V1. Importantly, however, identity assignment is very stable and accurate. https://youtu.be/OTP4dVSc7Es.

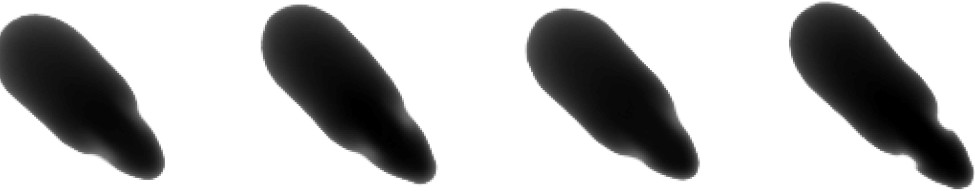

**Appendix 12—figure 2.** Median of all normalized images (N = 7161, 7040, 7153, 7076) for each of the four individuals from V1 in *Appendix 12—table 1*. Posture information was used to normalize each image sample, which was stable enough â€" also for TRex â€" to tell where the head is, and even to make out the ears on each side (brighter spots).

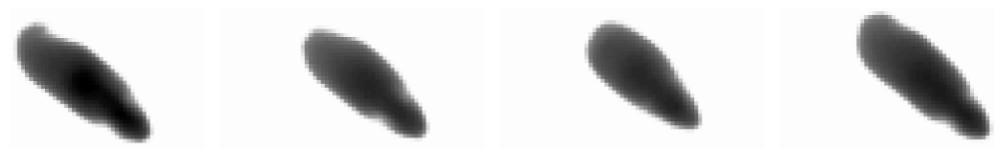

**Appendix 12—figure 3.** Median of all normalized images (N = 1593, 1586, 1620, 1538) for each of the four individuals from V2 in *Appendix 12—table 1*. Resolution per animal is lower than in V1, but ears are still clearly visible.

