## [Decision Letter]

**Acceptance summary:**

The paper by Walter and Couzin describes new open-source software to track individual animals, such as fish and insects, moving and interacting within groups in a quasi-two-dimensional plane. The method assumes controlled lighting conditions and a stationary camera, which facilitates ease of use and fast data analysis. The software is useful for animal behaviour, neurobiology and comparative biomechanics research. The authors report assessments of accuracy, run-time, and memory consumption, which illustrate this open-source solution is state-of-the-art. The invaluable utility of this open-source tool is enabled by how many standard and robust algorithms are combined in a single functional package. A particular strength of the present framework is the unusually large number of individuals that can be tracked simultaneously in real-time, enabling new virtual-reality manipulative studies of collective behaviour.

**Decision letter after peer review:**

Thank you for submitting your article "TRex, a fast multi-animal tracking system with markerless identification, and 2D estimation of posture and visual fields" for consideration by *eLife*. Your article has been reviewed by two peer reviewers, and the evaluation has been overseen by a Reviewing Editor and Christian Rutz as the Senior Editor. The following individual involved in the review of your submission has agreed to reveal their identity: Sergi Pujades (Reviewer #2).

The reviewers have discussed their reviews with one another, and the Reviewing Editor has drafted this decision letter to help you prepare a revised submission. Please address the comments and suggestions to the best of your ability, mark all changes in the revised manuscript using a blue font, and provide a point-by-point response to the issues raised. This will significantly facilitate the Reviewing Editor's evaluation of your revision.

Summary:

The paper by Walter and Couzin describes new open-source software to track individual animals such as fish and insects moving and interacting within groups in a quasi-two-dimensional plane. The method assumes controlled lighting conditions and a stationary camera, which facilitates ease of use and fast data analysis. The software is useful for animal behaviour, neurobiology and comparative biomechanics research. The authors demonstrate this utility based on assessments of accuracy, runtime, and memory consumption, which illustrate this open-source solution is state-of-the-art. While the authors rightfully point out that their system is applicable to essentially any animal species, their tests were mainly on fish and insects, which have relatively limited deformations and appearance changes compared to many mammals and other organisms with dynamically morphing body shapes. The invaluable utility of this open-source tool is enabled by how many standard and robust algorithms are combined in a single functional package. A particular strength of the present framework is the unusually large number of individuals that can be tracked simultaneously in real-time, enabling new virtual-reality manipulative studies of collective behaviour.

Essential revisions:

Abstract: Please mention that background subtraction is the key default segmentation approach. Clarify that limb position of highly deformable bodies is not tracked.

To put the contribution of the software into perspective, it would be helpful to add in the conclusions the assumptions made on the shape of the animals (subsection “Posture Analysis”), and please provide examples of animals which do not fall into this category, to help the general *eLife* readership comprehend both the promise and limits of the new method.

Subsection “Realtime Tracking Option for Closed-Loop Experiments”: Please clarify what "very outdated " or "low-end" is in this context. Specs are given in the Results. This could be rephrased in the line of: "Problems of using a lower hardware as the recommended one (ref) lead to.… frame-rates, fragmented data and bad identity assignments." Please address.

"QR codes" are more correctly called "fiducial markers" (and mostly QR codes themselves are not used).

Subsection “Automatic Visual Identification Based on Machine Learning”: To our understanding, James Crall and Stacey Combes performed a thorough study documenting the effects of these fiducial markers ("QR codes") for tracking and quantifying bee behaviour. It would be useful to cite this work, because it shows such documentation can be done although it is not trivial.

The final training:

Is a validation set kept to avoid overfitting? The procedure to check on "uniqueness improvement" is not clearly described. This could be improved.

Figure 1B: The text in the figure seems to imply that Trex would do real-time online tracking, but isn't this the case for TGrabs only? Please double-check the implication of the text in this figure to make sure it is what you wish to communicate to the reader.

Figure 2: Exports: Does TRex export to.csv? It seems .npz only, correct? Also in the Introduction.

---

## [Author Response]

Summary:The paper by Walter and Couzin describes new open-source software to track individual animals such as fish and insects moving and interacting within groups in a quasi-two-dimensional plane. The method assumes controlled lighting conditions and a stationary camera, which facilitates ease of use and fast data analysis. The software is useful for animal behaviour, neurobiology and comparative biomechanics research. The authors demonstrate this utility based on assessments of accuracy, runtime, and memory consumption, which illustrate this open-source solution is state-of-the-art. While the authors rightfully point out that their system is applicable to essentially any animal species, their tests were mainly on fish and insects, which have relatively limited deformations and appearance changes compared to many mammals and other organisms with dynamically morphing body shapes. The invaluable utility of this open-source tool is enabled by how many standard and robust algorithms are combined in a single functional package. A particular strength of the present framework is the unusually large number of individuals that can be tracked simultaneously in real-time, enabling new virtual-reality manipulative studies of collective behaviour.

Thank you for your detailed review of our method, and for your helpful suggestions. We agree that we should have been more clear regarding the ability of our software to accurately track, and maintain identities of animals that have highly deformable bodies, such as rodents. We have now added an additional Appendix (please see Appendix 12) where we show TRex can readily be used to track mice which, as can be seen from the included videos, have highly-deformable body shapes.

Essential revisions:Abstract: Please mention that background subtraction is the key default segmentation approach. Clarify that limb position of highly deformable bodies is not tracked.To put the contribution of the software into perspective, it would be helpful to add in the conclusions the assumptions made on the shape of the animals (subsection “Posture Analysis”), and please provide examples of animals which do not fall into this category, to help the general eLife readership comprehend both the promise and limits of the new method.

We have now made it clear in the Abstract that the default is “using background-subtraction”. We have also replaced “postures” with “visual-fields, outlines, and head/rear of bilateral animals” to clarify what the specific assumptions/limits of our method are – as per our main text, where we write *“*TRex does not track individual body parts apart from the head and tail of the animal”. While it is possible, however, to derive limb information from the outline generated by TRex, we also added a sentence in the main-text to make clear that output from TRex can be utilised in complimentary markerless tracking software such as our DeepPoseKit (Graving et al., 2020), and DeepLabCut (Mathis et al., 2018). The paragraph now reads as follows:

“When detailed tracking of all extremities is required, TRex offers an option that allows it to interface with third-party software like DeepPoseKit (Graving et al., 2019), SLEAP (Pereira et al., 2020), or DeepLabCut (Mathis et al., 2018). […] Normalisation, for example, can make it easier for machine-learning algorithms in these tools to learn where body-parts are likely to be (see Figure 5) and may even reduce the number of clicks required during annotation.”

Subsection “Realtime Tracking Option for Closed-Loop Experiments”: Please clarify what "very outdated " or "low-end" is in this context. Specs are given in the Results. This could be rephrased in the line of: "Problems of using a lower hardware as the recommended one (ref) lead to.… frame-rates, fragmented data and bad identity assignments." Please address.

We have rephrased the sentence to:

“Running the program on hardware with specifications below our recommendations (see Results), however, may affect frame-rates as described below”,

and added clarifications to the paragraph as follows:

“If the script (or any other part of the recording process) takes too long to execute in one frame, consecutive frames may be dropped until a stable frame-rate can be achieved. […] Alternatively, if live-tracking is enabled but closed-loop feedback is disabled, the program maintains detected objects in memory and tracks them in an asynchronous thread (potentially introducing wait time after the recording stops).”

"QR codes" are more correctly called "fiducial markers" (and mostly QR codes themselves are not used).

We changed this line, and also added citations for recent papers:

“Attaching fiducial markers (such as QR codes) to animals allows for a very large number (thousands) of individuals to be uniquely identified at the same time (see Gernat et al., 2018, Wild et al., 2020, Mersch et al., 2013, Crall et al., 2015) – and over a much greater distance than RFID tags.”

Subsection “Automatic Visual Identification Based on Machine Learning”: To our understanding, James Crall and Stacey Combes performed a thorough study documenting the effects of these fiducial markers ("QR codes") for tracking and quantifying bee behaviour. It would be useful to cite this work, because it shows such documentation can be done although it is not trivial.

This is a good point. We searched extensively and could not find a paper by these authors that had thoroughly evaluated the effects of fiducial markers (we are certainly keen to add it and are sorry if we have missed it). We have already cited another paper by Crall and Combes (Crall et al., 2015) elsewhere, but this does not study how tags impacted the animals' behavior. However, we have now cited a related paper, “Switzer and Combes, 2016”, which is a comparative study on the effects of using RFID tags vs. paint to mark animals. It shows that the effects of different kinds of tagging can, with considerable effort, be documented. The paragraph now reads:

“While physical tagging is often an effective method by which to identify individuals, it requires animals to be caught and manipulated, which can be difficult (Mersch et al., 2013) and is subject to the physical limitations of the respective system. […] In addition, for some animals, like fish and termites, attachment of tags that are effective for discriminating among a large number of individuals can be problematic, or impossible.”

The final training:Is a validation set kept to avoid overfitting? The procedure to check on "uniqueness improvement" is not clearly described. This could be improved.

Yes. In addition to using Dropout layers in our network layout, a validation set is kept to avoid overfitting in the early training units. This is necessary since 1. overfitting is much more likely to happen in smaller datasets, and 2. at early stages we still care about how well our network generalises (especially to segments far away from the current segment). This restriction is much less important in the final training unit (except maybe for a use-case involving transfer learning, where it can be disabled) simply because it is the end of the procedure and we aim to (1) include all parts of the video, and (2) get as close to 1/0 probability predictions as we can (anecdotally speaking, however, we do not observe such overfitting effects in our results – likely thanks to Dropout layers – and extensive, but as-yet unpublished, work by A. Albi from our lab involves transfer learning and works equally well with/without this last step).

We have changed the text to reflect what we have described here:

“After the accumulation phase, one last training step is performed. […] The reason being that this is the first time when all of the training data from all segments is considered at once (instead of mostly the current segment plus fewer samples from previously accepted segments), and samples from all parts of the video have an equal likelihood of being used in training after possible reduction due to memory-constraints.”

Figure 1B: The text in the figure seems to imply that Trex would do real-time online tracking, but isn't this the case for TGrabs only? Please double-check the implication of the text in this figure to make sure it is what you wish to communicate to the reader.

You are absolutely correct. We kept the introductory bullet-points from the Materials and methods when moving the Results to the front of the document, but changed the beginning of that section to:

“Our software package consists of two task-specific tools, TGrabs and TRex, with different specializations. […] Typically, such a sequence can be summarized in four stages (see also Figure 2 for a flow diagram)”.

To better reflect this we have also changed Figure 1B/Tracking section.

Figure 2: Exports: Does TRex export to.csv? It seems .npz only, correct? Also in the Introduction.

Yes, TRex also exports to CSV. In the main text we now write:

“Results can be exported to independent data-containers (NPZ, or CSV for plain-text type data) for further analyses in software of the user’s choosing.”

The user can select the output format, which depends on a settings parameter (output_format), which accepts the values “npz” or “csv” (see https://trex.run/docs/parameters_trex.html#output_format). This can be changed from within the GUI, or via the command-line (as all parameters can). Some specific data cannot be exported as plain-text-CSV because they are generally very large, such as visual field data (essentially a stream of floating-point images) and the full posture data – thus being able to export all data in both formats does not offer any additional benefit to users. We have added “for plain-text type data” to reflect this minor restriction, excluding larger binary outputs.

We also changed the text under Figure 2 to:

“Thanks to its integration with parts of the TRex code, TGrabs can also perform online tracking for limited numbers of individuals, and save results to a .results file (that can be opened by TRex) along with individual tracking data saved to numpy data-containers (.npz) or standard CSV files, which can be used for analysis in third-party applications.”